# PhyloLM: Inferring the Phylogeny of Large Language Models and Predicting their Performances in Benchmarks

**Nicolas Yax**
LNC2, INSERM, Paris, France
DEC, ENS, Paris, France
Inria, France
nicolas.yax@ens.psl.eu

**Pierre-Yves Oudeyer***
Inria and University of Bordeaux, France

**Stefano Palminteri***
LNC2, INSERM, Paris, France
DEC, ENS, Paris, France

∗ equal contribution

## Abstract

This paper introduces PhyloLM, a method adapting phylogenetic algorithms to Large Language Models (LLMs) to explore whether and how they relate to each other and to predict their performance characteristics. Our method calculates a phylogenetic distance metric based on the similarity of LLMs' output. The resulting metric is then used to construct dendrograms, which satisfactorily capture known relationships across a set of 111 open-source and 45 closed models. Furthermore, our phylogenetic distance predicts performance in standard benchmarks, thus demonstrating its functional validity and paving the way for a time and cost-effective estimation of LLM capabilities. To sum up, by translating population genetic concepts to machine learning, we propose and validate a tool to evaluate LLM development, relationships and capabilities, even in the absence of transparent training information.

## 1 Introduction

The Large Language Models (LLMs) landscape is vast and rapidly expanding, comprising both private and open-access models. Each day a few hundreds of new language models are created on the huggingface hub among which most will not be benchmarked, and a small minority are transparent about the training details. Evaluating these models presents challenges due to the sheer volume and the complexity of assessing their true capabilities. The evaluation methods used today mostly rely on a multitude of benchmarks, each focused on specific domains like reasoning or question-answering (Chollet, 2019; Hendrycks et al., 2021; Srivastava et al., 2023). However, tracking LLMs evolution and progress using benchmarks presents inherent limitations, including the fact that they are rather domain-specific, meaning that to get a full picture of a model's capabilities one has to run multiple costly tests that are prone to contamination (Chang et al., 2023; Deng et al., 2023; Liang et al., 2023). Moreover, the opacity of algorithmic and training data specifications in many models, adds further complexity and constraints to monitor progress in LLMs (Liao & Vaughan, 2023).

Our approach stems from the observation that most of the newly released models are not created ex-nihilo (from scratch). In fact, they rather inherit features from existing ones, such as training data or initial weights. We reasoned that we could therefore think about LLMs development as

---

code : https://github.com/hrl-team/PhyloLM
notebook : https://colab.research.google.com/drive/1agNE52eUevgdJ3KL3ytv5Y9JBbfJRYqd?usp=copy

an "evolutionary" process and therefore study their relationships and functional properties with conceptual and quantitative tools borrowed from genetics.

In the field of Phylogeny, algorithms have been developed that reconstruct evolutionary trees to understand evolutionary relationships among species (Takezaki & Nei, 1996). The idea of applying these methods, initially developed for biology, to cultural artefacts is not new. Previous studies yielded particularly useful insights into the evolution of popular tales, languages, or craft assemblages (Atkinson et al., 2008; Dawkins, 1976; d'Huy, 2013; Gray et al., 2010; Tehrani & d'Huy, 2017; Tehrani & Collard, 2009). We hypothesize here that LLMs, which are a new kind of cultural artefact (in the sense that they are productions of humans that convey information about the culture of their creators and users), may also be studied using similar tools.

Thus, we here apply a conceptually similar approach to LLMs and, by doing so, we make several contributions. In a **first contribution**, we introduce an algorithm, **PhyloLM**, inspired by a simplified phylogenetic model, but specifically tailored for Large Language Models (LLMs), which core idea is to consider that generated tokens are to contexts what alleles are to genes in genetics. This analogy makes it possible to apply algorithms from the genetics framework to LLMs and to generate distance matrices and dendrograms. In addition to presenting the underlying theory, we also explore the hyperparameters of our algorithm to strike a balance between precision and computational efficiency.

In our **second contribution**, we analyze the resulting phylogenetic trees ("dendrograms") and confirm that **PhyloLM** is capable of correctly retrieving known relationships between LLMs and overall correctly capturing models families and sub-families. Our analysis primarily focuses on open-access model families (Llama (Touvron et al., 2023a;b), Mistral (Jiang et al., 2023), Bloom (BigScienceWorkshop et al., 2023), Pythia (Biderman et al., 2023), Falcon (Almazrouei et al., 2023), OPT (Zhang et al., 2022), Qwen (Bai et al., 2023) and Gemma (Team et al., 2024b) families), where ground truth information is available, but also provides insights into fine-tuning relationships for proprietary models (GPT-3 (Brown et al., 2020), 3.5 (Ouyang et al., 2022), 4 (OpenAI et al., 2023), Claude , Palm (Chowdhery et al., 2022) and Gemini models (Team et al., 2024a)). Finally, in our **third contribution**, we examine whether phylogenetic distance can also be used to predict performance in several benchmarks, thus showing that the utility of **PhyloLM** extends to the assessment of functional properties of LLMs.

To sum up, our study illustrates the potential of leveraging methods from genetics to understand how models evolve, shedding light on their relationships and functional capabilities in a relatively cost-efficient manner, even in the absence of transparent training information and also without direct access to the model.

$$S(P_1, P_2) = \frac{\sum_{g \in G} \sum_{a \in A_g} P_1(a|g) P_2(a|g)}{\sqrt{(\sum_{g \in G} \sum_{a \in A_g} P_1(a|g)^2)(\sum_{g \in G} \sum_{a \in A_g} P_2(a|g)^2)}} \tag{1}$$

Equation 1: **Similarity computation** with $P_1$ and $P_2$ two populations seen as probability distribution of alleles $a$ given a gene $g$ estimated empirically in the selected populations. $G$ is the set of genes considered and $A_G$ the set of possible alleles for this gene and matrix $S$ is the similarity matrix (bounded in [0,1]). In genetics people tend to use a distance matrix $D$ to plot dendrograms derived from the similarity matrix with this formula $D(P_1, P_2) = -\log(S(P_1, P_2))$ (Takezaki & Nei, 1996). Seen from the autoregressive LLM framework, 'populations' are LLMs, 'genes' are contexts and 'alleles' are the different tokens in the vocabulary : $P(a|g) = LLM(t|c)$

## 2 METHODS

### 2.1 TRANSLATING PHYLOGENETIC ALGORITHMS TO LLMS

In the current landscape, LLMs predominantly operate on an autoregressive basis, wherein they learn the conditional probability denoted as $LLM(t|c)$. Here, $LLM$ represents the probability learned by the language model, $t$ signifies a token, and $c$ denotes the context in which to sample token $t$. Transposing genetic methods to LLMs involves establishing analogies for the elements of the phylogenetic analysis, namely genes, alleles, and populations. Drawing a parallel with the notation for

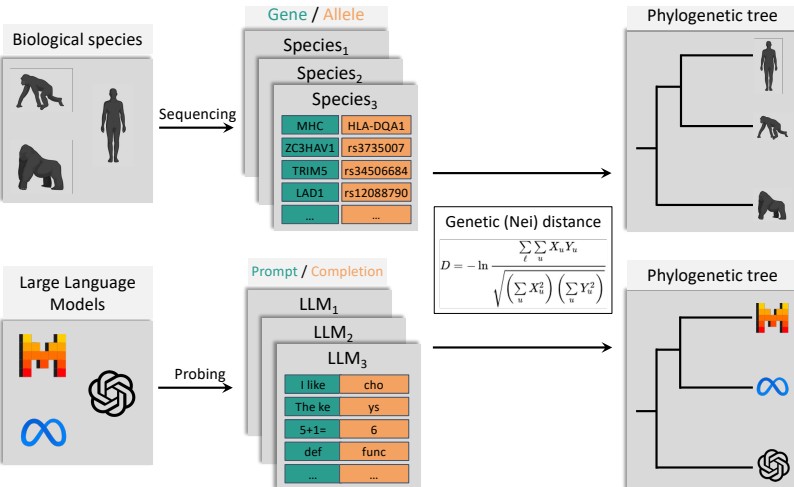

Figure 1: **Analogy between running human genetic studies and LLMs genetic studies.** The first stage consists in selecting genes (for both humans and LLMs). Then alleles are collected for each individual in the population and will be used to compare the populations (either populations of humans or LLMs seen as populations). Finally these data go through the Nei distance computation (Takezaki & Nei, 1996) that returns a distance matrix that can then be turned into dendrograms using the NJ algorithm (Saitou & Nei, 1987) in the same way for both humans and LLMs.

populations in the Nei genetic distance (see Equation 1) (Takezaki & Nei, 1996), $P_1(a|g)$ with $a$ an allele and $g$ a gene, we propose that LLMs play the role of populations (i.e., the set of the individuals belonging to a given population); contexts (or "prompts") are aligned to genes (i.e., portions of DNA); finally, tokens align with alleles (i.e., variants in the DNA sequence).

To substantiate this analogy, consider that, in the realm of genetics, populations are conceptualized as probability distributions of DNA, represented by $P(a|g)$, where $a$ stands for specific alleles at gene locations. Gene-specific alleles are then considered to be probabilistically drawn from the abstract statistical construct that is the population, akin to context-specific tokens are probabilistically generated from Large Language Models, expressed as $LLM(t|c)$ ($t$ being a token likely to follow text $c$). The generated text can therefore be seen as a thread of DNA, comprised of tokens (alleles) sampled in contexts (genes) according to a probability distribution defined by the LLM.

To elucidate this crucial point, consider a tokenized text sequence: 'I' '_like' '_choco' 'late'. This sequence can be analogous to a DNA thread represented as 'I_like_chocolate'. Breaking it down, the allele I corresponds to the gene $\epsilon$ (empty text), _like aligns with the gene I, _choco associates with the gene I_like, and late is linked to the gene I_like_choco. Now, consider another individual represented by 'I' '_prefer' '_ice' '_cream'. These two individuals share exactly two genes: $\epsilon$, for which they possess the same allele I, and the gene I, for which they have distinct alleles (_like and _prefer). They do not share any further genes, as their prefixes diverge beyond this point.

The algorithm is illustrated in Figure 1. The initial step involves collecting model outputs to contexts (genes). Given a set of LLMs, a set of 'genes', and the specified number of individuals in each population (i.e., the number of times the model is queried on each gene refered to as the number of probes) as $N$, the models are queried for a single token $N$ times. This process generates the matrix $P$, which serves as an approximation of $P(a|g)$, the proportion of the population with allele $a$ to gene $g$. Subsequently, based on this approximation, the similarity matrix $S$ is computed using the Nei genetic distance formula (Takezaki & Nei, 1996) depicted in Equation 1. The pseudo code of PhyloLM can be found in Algorithm 1 in Appendix C.

## 2.2 CHOICE OF THE SET OF GENES

The implementation of phylogenetic algorithms requires selecting specific genes that show enough evolutionary changes among the species studied to differentiate them, while still retaining enough

similarity to trace relationships between closely related species (Grünwald et al., 2017). If these genes mutate too quickly and are completely altered between similar species, they will not provide useful information about their evolution. Conversely, if they are too stable and show no changes across the species being considered, they are also not informative. These genes must strike a balance between stability and variation among the species studied.

That is why we need to carefully select genes (i.e., prompt contexts) that could show a moderate variance between LLMs. Recent LLM development focused a lot on instruction tuning, reasoning and coding (Brown et al., 2020; Chiang et al., 2023; OpenAI et al., 2023; Taori et al., 2023). Selecting contexts on these topics might show a relevant variance between generations of language models as well as finetuning refinements that improved these models on these specific topics.

Furthermore, contexts ('genes') which are very likely to belong to the training data of these LLMs can suffer from contamination issues and generate very low variance[1]. To obviate this issue, we used contexts (or a 'gene' set) taken from recent test benchmarks because, in principle, LLMs shouldn't be trained on this data. To further assess the robustness of our approach and study the impact of the choice of the set of 'genes', we took our contexts from two different test sets: open-web-math (Paster et al., 2023) and MBXP (Athiwaratkun et al., 2023). They address different capabilities of LLMs: reasoning and coding, respectively, which are very relevant in recent LLM-related research and are therefore likely to deliver useful results.

The exact selection of contexts from the benchmarks consisted of randomly and uniformly selecting lines from the solution column in the datasets and truncating the text to leave it open for LLMs to complete the sentence. To decide the length of the contexts we need once more to think about making 'genes' show a moderate completion variance. If the context is only a few tokens long it may not be informative enough for LLMs to understand the topic of the context (that is relevant for the recent evolution of language models as discussed above) but also to follow the logics of the text that would constrain the generation. On the other hand, making it hundreds of tokens long will induce additional costs without necessarily improving the variability balance. That is why we decided to truncate randomly and uniformly between the 20th and 100th characters in each text (5 to 30 tokens approximately). 'Gene' examples are shown in Appendix A. More details about the impact of the gene length can be found in appendix K.1.

## 2.3 SELECTION OF THE HYPER-PARAMETERS OF THE DISTANCE MATRICES

We devised two complementary analyses to estimate the right hyperparameters to run PhyloLM. The hyperparameters are the 'gene' set, the number of probes and sampling parameters from the LLM (see Appendix B). Testing the gene set is more difficult as testing thousands of different combinations of genes would come at a very expensive cost. Thus we limited ourselves at 2 parameters of the gene set : the topic (math and code in this paper) and the size of the gene set $G$. In this section we will investigate the impact of $G$ and $N$ in the math gene set, the results for the code gene set are in Appendix D.

First we investigate how $G$ and $N$ affect the variability of the distance matrix, namely how much the similarity matrix changes between different estimations. We focus on similarity matrices (the matrix $S$ in Equation 1) instead of distance matrices at this point as they are bounded in [0,1] making them a lot easier to plot and compare. Then, once the variance is controlled, what combination of $G$ and $N$ approximate reasonably well a very high $G'$ and $N'$ distance matrix.

To assess the impact of the number of contexts ('genes') $G$ and the number of probes/individuals $N$ for each dataset, the algorithm was executed across a range of gene set sizes $G$ (varying between 16 and 256 genes per run) and individuals $N$ (ranging from 8 to 128) building similarity matrices. This optimization process, aimed at testing the best values for the algorithm hyperparameters, is particularly computationally expensive. Therefore it was only run on the 5 smallest OPENAI models (ada,babbage,text-ada-001,text-babbage-001 and babbage-002), in order to minimize the costs. Thus similarity matrices in this section are $5 \times 5$ making it an estimate of what could be a larger distance matrix at a very low cost.

---

[1]To understand this point, imagine using "*May the force be with*" as context. All models will complete this sentence with "*you*", thus making impossible establishing distance matrices between them

To investigate the variability of PhyloLM for different combination of hyperparameters, we composed 8 sets of genes of size $G$, each with different genes. Each set of gene is probed $N$ times to build a similarity matrix $S_{G,N,i}$, $i \in [\![0,7]\!]$ representing the independant set of genes of size $G$ used to generate the matrix with $N$ probes. A variance computation over this set of matrices is finally performed yielding a matrix V containing the variance of each distance between 2 models : $V_{G,N}{}^2 = \frac{1}{8}\sum_i \left(S_{G,N,i} - \left(\frac{1}{8}\sum_j S_{G,N,j}\right)\right)^2$. The square operator is applied coefficient by coefficient. The final variability score is the mean value of the coefficients in the matrix $v_{G,N} = \mu\left(\sqrt{V_{G,N}{}^2}\right)$.

Then we investigated the impact of these hyperparameters when trying to approximate a high precision matrix. For this purpose, we compute the variance around a very expensive distance matrix $S_{G',N'}$ with $G' = 2048$ and $N' = 128$. The gene set for the high precision matrix is independent from the lower size set of genes used to estimate it. The formula to compute this variance around the high precision matrix is $V'_{G,N}{}^2 = \frac{1}{8}\sum_i \left(S_{G,N,i} - S_{G',N'}\right)^2$. The final metric is the mean value in the matrix $v'_{G,N} = \mu\left(\sqrt{V'_{G,N}{}^2}\right)$.

## 2.4 ALIGNMENT OF THE RESULTS ACROSS DIFFERENT TOKENIZATION

In situations where models do not share the same tokenizer, comparing only the first alleles generated can pose challenges. For instance, if the context is "*The president of the US is Joe*," and one model could complete with "*Biden*" in one token while another could complete with "*Bi*" "*den*" in two tokens they would be considered as different alleles while both LLM meant the same completion.

To mitigate this issue of tokenizer alignment, a proxy approach was employed by only using the first 4 characters of the generated text instead of the first token. Practically, each model was instructed to generate at least 4 tokens (tokens are at least 1 character long) and the comparison focused on the first 4 characters in the concatenation of these tokens. In the previous example, the word "*Biden*" generated in one token or in two ("*Bi*" and "*den*") would have been considered as the same response, because the first 4 characters ("*Bide*") constitute the same 'allele', despite having being tokenized differently. In other words, we are retokenizing the text using a tokenizer with a vocabulary of words that are 4 characters long, and then comparing the first token of the generated text with this new tokenization scheme. An example of the results of such a proxy approach is presented in Appendix A. Further details about why 4 characters is efficient are discussed in Appendix K.2.

## 2.5 VISUALIZATION OF THE RESULTS

From a distance matrix obtained by the phylogenetic algorithm it is usual to plot dendrograms representing a possible evolution between the entities in the distance matrix. For this purpose many different algorithms exist and we chose the Neighbour Joining (NJ) technique (Saitou & Nei, 1987) for its simplicity, efficiency and being a common choice in genetics. We plotted unrooted trees as they are easier to make figures that fit in a paper and are more adapted to LLM evolution than rooted ones. The analysis of the resulting dendrograms also allowed us to validate the capability of our algorithm to predict actual relationship between LLMs in cases where the ground truth is known.

## 2.6 PREDICT BENCHMARK SCORES FROM GENETIC DISTANCE

We explored whether genetic distance can predict model performance by using logistic regression to estimate benchmark scores of large language models based on their similarity to other models. Due to the high dimensionality of the similarity matrix, we reduced the input dimensions to 15 using Independent Component Analysis (ICA), resulting in 15 parameters to learn from approximately 100 data points per fit. We then applied a sigmoid function to the output to scale the predictions between 0 and 1, corresponding to benchmark scores ranging from 0% to 100%. Since benchmark scores can be highly correlated within a family of models, we employed a leave-one-family-out method (see Figure 5 (a)). This involved training the regressor on all families but one and testing it on the excluded family. A Mean Squared Error loss was used with an Adam optimizer (learning rate of $10^{-3}$).

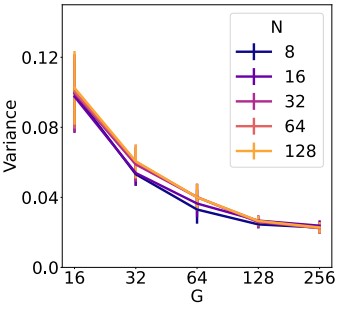 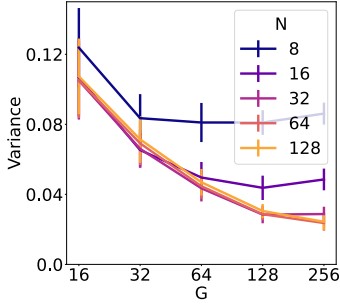

(a) Variability of distance matrices        (b) Distance to the high precision matrix

Figure 2: **Hyperparameters impact on distance matrices in the math set of genes** (a) shows the variability of distance matrices for different number of genes G and number of probes N in the math benchmark. Each set of genes of specified size contains different and independent genes from the other matrices for a total of 8 distance matrix for each data point in the figure. (b) shows the distance to the high precision matrix made of 2048 genes and N=128 in the math benchmark. Errorbars represent the standard error of the mean.

We tested the benchmarks available on the hugging face open llm leaderboard which includes MMLU, ARC, Hellaswag, TruthfulQA, Winogrande and GSM8k (HuggingFaceH4) and only included open access models for which the scores are available on the leaderboard. Thus we didn't include proprietary models in this study as, as explained in later sections, distance computation is slightly biased for these models and benchmark scores are not obtained in the same conditions as in the leaderboard (number of shots, CoT, ...). The benchmarks used for the 'gene' set were distinct from these benchmarks to avoid any type of contamination between the 'alleles' used to generate genetic distances and the performance of the models in the considered benchmark tasks.

# 3 EXPERIMENTS AND RESULTS

## 3.1 WHAT IS THE IMPACT OF HYPERPARAMETERS ON THE DISTANCE MATRIX?

We first ran the hyperparameters' optimization process explained in Methods2.3 and plotted the results in figure 2a left side. This graph shows a clear decrease in the variability as the number of 'genes', $G$ grows with almost no effect from $N$. This is interesting : it seems that having different sets of 'genes' doesn't appear to change the similarity matrix as long as there are enough of them (at least in the open-web-math and mbxp dataset - see Appendix D for the results on the code set of 'genes').

However this method doesn't make it possible to find a good $N$, indeed, the probability for two models to generate the same token in the same context in only one try is quite low. Therefore, a very low $N$ will make all models appear particularly different making the similarity matrix look like the identity matrix yielding unsatisfactory results despite having a low variance. Thus having a $N$ high enough is required to get a useful similarity matrix and we need to find a better metric but how to choose it ?

We have just seen that $G$ monitors the variability of the matrix (variability parameter), thus a similarity matrix with a very high $G$ should be particularly stable across different sets of genes. We then compared modestly parametrized similarity matrices to study how hyperparameters $G$ and $N$ influence the difference between a lower precision matrices to a high precision matrix on average (see Methods 2.3 for the computational details). This new metric should penalize having a low $N$ leading to similarity matrices close to the identity matrix and may yield more satisfying results.

As shown in Figure 2b, while increasing the number of genes still seems to approximate better high precision matrix, this time, the number of probes is also very important. Indeed, for each value of $N$, the performance saturates from some $G$ value making less and less improvement when $G$ increases. Thus, this figure gives an optimal $G$ for a given $N$ in order to approximate the high precision matrix efficiently with a low cost. The total cost of the algorithm in tokens being proportional to $G \times N$, we found a good tradeoff between variance and precision around $G = 128$ and $N = 32$.

The estimated cost to run the algorithm per model is therefore $128\,\text{genes} \times 32\,\text{probes} = 4096$ queries of $\approx 20$ tokens. As a point of reference, conducting the MMLU benchmark requires around 14,000 queries on significantly longer prompts ($\approx 70$ tokens each), making PhyloLM approximately 10 times less expensive in terms of the number of tokens required.

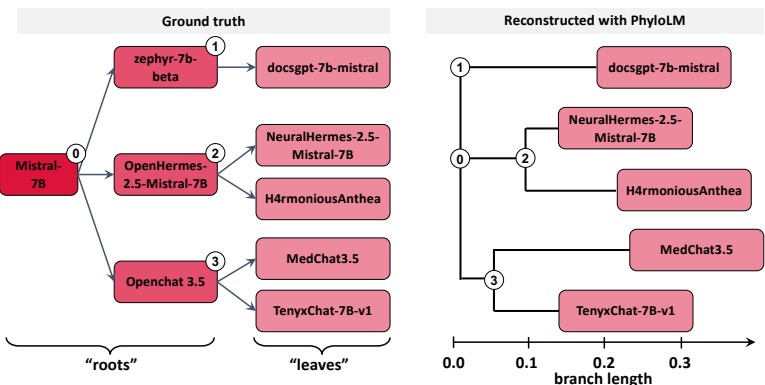

Figure 3: **Phylogenetic tree reconstruction**. On the left it is shown the ground truth concerning the relation of some LLMs of the Mistral family. Right is the reconstruction from the phylogenetic algorithm on the 'math' set of genes for the five latest models of this family ("leaves" of the phylogenetic tree) on which we run PhyloLM. On the right, it is shown the reconstructed phylogenetic tree PhyloLM on the 5 "leafs" models. The numerical labels (0:3) map the true common ancestors (on the right, "ground truth") to the inferred ones (on the left, "reconstructed"). It can be seen that the true and the reconstructed trees are topologically equivalent

## 3.2 CAN WE TRACE BACK THE GENEALOGY OF LLMs USING TOOLS FROM GENETICS?

We first examine the results of PhyloLM by analyzing the resulting phylogenic trees (materialized as dendrograms). However, before dwelling into the results, an important point to understand is that, in genetics, branches in the tree show probable speciation events that occured in the past, when from an extinct common ancestor, two (or more) current species (leaves of the tree) emerged. When looking at LLMs, 'common ancestors' are not extinct, but rather among the studied 'populations'. Take for instance Mistral 7B that is the common ancestor of OpenChat3.5 and Zephyr 7B Alpha, but still included in our analysis. Oblivious of this difference, the dendrogram plotting method will put all models at the 'leaves' of the tree, while, in fact, some of them (such as Mistral 7B) should be at a speciation node. As such, without additional information about which model is at a node, it is difficult to interpret them in the same way as in genetics. Without this important phylogenetic assumption, one has to bear in mind that what matters (and should be compared with the ground truth) is their relative distance and position when evaluating the dendrograms resulting from the phylogenetic analysis of LLMs. Indeed the distance between two models is represented by the distance from their respective leaves in the dendrogram.

To investigate the capabilities of PhyloLM, let's first start by respecting this assumption by looking at a set of 9 models from the Mistral family whose relationships are known because transparently disclosed by their creators. Out of these 9 models, 5 are leaves in the ground truth dendrogram (Arc53, 2023; mlabonne, 2023; Tenyx, 2024; Ullah, 2024; Vallego, 2024). Running PhyloLM on these 5 models getting the distance matrix between them and finally plotting the NJTree we perfectly get back the ground truth phylogenetic tree (see Fig 3) validating the method. These rooted trees are not necessarily very stable as the NJ algorithm makes an unrooted tree of the evolution but then has to choose the root. In Appendix D we show that, on the code genome, the root has been mistakenly attributed to model 3 while the structure of the tree is right. That is why we prefer to plot unrooted trees in the rest of this paper.

### 3.2.1 GLOBAL DENDROGRAM

**LLMs: open-source vs private, completion vs chat**    Now let's drop the assumption of not having 'common ancestors' in the set of LLMs. The LLMs we are investigating here include 111 open

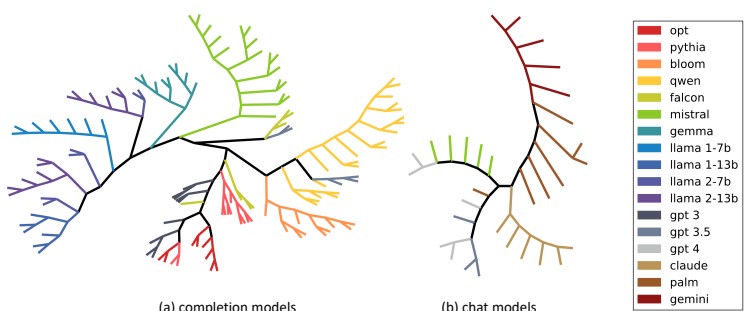

| | |
|---|---|
| opt | pythia |
| bloom | qwen |
| falcon | mistral |
| gemma | llama 1-7b |
| llama 1-13b | llama 2-7b |
| llama 2-13b | gpt 3 |
| gpt 3.5 | gpt 4 |
| claude | palm |
| gemini | |

(a) completion models          (b) chat models

Figure 4: **Inferred phylogenetic tree of LLMs on the 'math' set of genes**. (a) completion models inlcude all open source models included in our study and the 14 openai completion models (b) chat models include additional proprietary models. Completion and chat models were separated because they are not comparable due to additional prompting from the API. Llama models have been split by version of the pretrained model and the number of parameters.

access models spanning from 70M to 176B parameters and 45 closed LLMs. Most modern LLMs are only accessible through a chat API which naturally adds new tokens to the prompt such as chat messages markers biasing the completion of the given 'gene'. This can strongly influence PhyloLM as the algorithm will compare 'alleles' that do not correspond to the same 'gene'. As such we call *completion models* LLMs that were accessed in a way that can generate a completion to a very specific sequence of tokens without adding more tokens. All the 111 open access models we included in this study were accessed in this completion setting but among the 45 proprieraty models we only considered 14 of them to be completion models (see Appendix B for more details). That is why we split the LLMs and investigated them in 2 groups: completion models (to show the capabilities of PhyloLM when run in good conditions) and the others on which we suspect additional prompting manipulation. In both classes of models we found that our algorithm was largely capable of clustering LLMs into their original families, with only a few specificities discussed below. Dendrograms for both model classes are in Figure 4.

In the completion group of models we notice very clear Llama clusters separating the family from other families but also on a more fine grained level, subfamilies of llama linked to the version of the models and their respective sizes. Similarly clear cluster appear for Mistral, Qwen and Bloom. The other families such as Falcon, OPT, Pythia and GPT 3 are more mixed with each other and indeed we know that OPT, Pythia and Falcon-RW-1B (the one the closest to OPT in the tree) were trained each on their own version of the Common Crawl dataset and thus share a similar training set. Lastly, some GPT-3 models (ada, babbage and curie) appear to be close to this OPT,Pythia and Falcon-RW cluster showing they may have been trained on a version of the CommonCrawl as well. On the other hand, GPT-3.5 completion models including text-davinci-002 and text-davinci-003 seem to share more with Falcon than other models while davinci-002, babbage-002 and gpt-3.5-turbo-instruct look more related to Qwen and more precisely its CausalLM finetuning. It is important to understand that dendrograms in LLMs are just a visualisation tool, much more details can be found in the similarity matrix shown in Figure 9 Appendix I shows dendrograms with model names of the models (see Figure 23).

In the chat models group, we also find a lot of structure : Palm and Gemini models are on the same branch, Gemini seems to be a further improvement on Palm as it is further on the branch (and indeed they are both from Google showing maybe a sharing of their training data) while claude has its own branch and Mistral / GPT-3.5 and GPT-4 models show some similarities. Dendrograms with model names are provided in Figure 23 in Appendix I.

Additional figure are available and discussed in Appendix: similarity matrices are in Figure 9 (Appendix E). Code results are in Appendix D, with the dendrogram in Figure 8 and the similarity matrix in Figure 10. Additional mixed class figures are in Appendix G: Figure 18 (math), Figure 19 (code), and global similarity matrices in Figures 16 and 17.

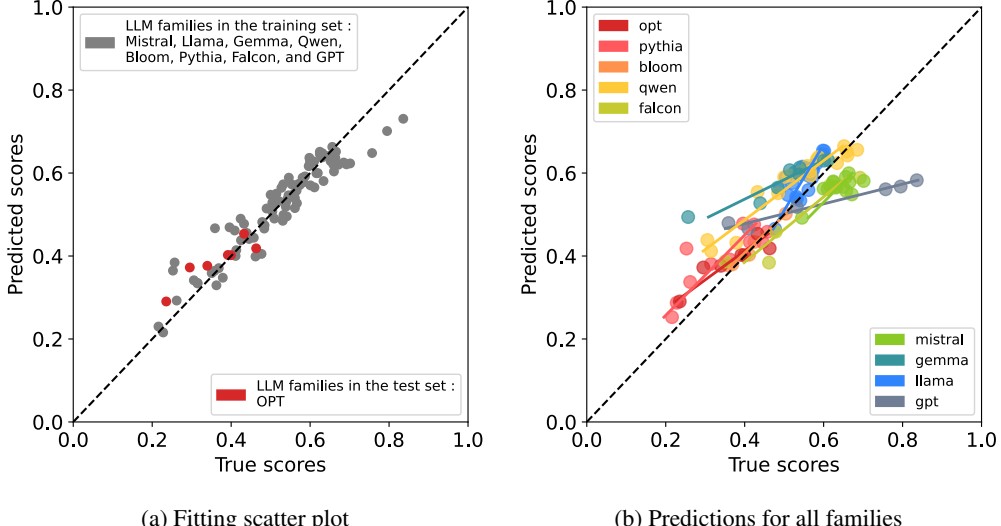

(a) Fitting scatter plot          (b) Predictions for all families

Figure 5: **Predictions from the logistic regression compared to ground truth for every model (leave one family out method) on ARC benchmark**. (a) Scatter plot showing the fitting of the logistic regression on all models but the OPT family (in grey) and the prediction of OPT performance by the regression (in red). (b) Predictions from the logistic regression for each family. To predict a family, the regressor fits on all the other families to finally predict the score of the models from the remaining family (leave one family out method - see (a)).

### 3.3 CAN WE INFER MODEL CAPABILITIES FROM THE GENETIC DISTANCE?

We then investigated whether the genetic distance metric can be used to predict the abilities of language models. As such we used the benchmark scores from the Huggingface open LLM leaderboard. The results indicate that the prediction correlates with the true score of the models (Figures 5 (b) and 15 (a)). Indeed, we found that the Pearson's correlation coefficients (r) of the correlation between the true scores and the predicted ones was positive and significant for all benchmarks and regardless of the set of "genes" used to make the prediction (mean$\pm$sem: $0.68\pm0.04$; Student's t-test again zero: $t(11)=16.0$, $p<0.001$; see Figure 15 (b) in Appendix F). In other terms, within benchmarks and across families, the phylogenetic distance metric allowed us to predict on average $48.2\pm0.03\%$ of the variance of the between-model benchmark performance. In a control analysis, we also verified that significant correlation was also achieved within families, thus eliminating the possibility that significant prediction in the previous analysis was driven by our metric simply capturing the fact that different families have different levels of performance on average. To do so, we calculated the Pearson correlation between the true and the predicted scores per benchmark and within each family separately. The results indicate that, even though for some combinations of families and benchmarks, we obtained small or negative correlation coefficients (which is unsurprising, since these correlations were sometimes calculated across very few data points), also in this case, the results were in average positive and significant difference from zero ($0.64\pm0.05$; $t(107)= 20.7$, $p<0.001$; see Figure 15 (c) in Appendix F). Within families, the variance explained by our method amounted to $52.2\pm0.03\%$ on average, thus indicating that our metric achieved good predictive power even when drastically increasing the level of granularity. Individual plots for each benchmark are shown in Appendix F

## 4 DISCUSSION

Here we show that an algorithm, inspired by those used in phylogeny, is successful in reconstructing important aspects of the genesis of LLMs, based solely on their outputs to diverse short queries. By leveraging the genetic distance matrix, it becomes feasible to robustly trace the relationships and evolution of models over time. This is particularly evident in the constructed dendrograms, where clear clusters align with distinct families of LLMs, offering a visual representation of their evolutionary trajectories or at least their training similarity. It is important to also emphasize the

applicability of these methods to proprietary models. Understanding the fine-tuning relationships and performance characteristics of private models is often challenging due to limited access to training details and data. PhyloLM offers a valuable tool for gaining insights into these aspects, by providing to the research community a more transparent image of how proprietary models evolve.

We also show that the utility of the "genetic" distance, derived from our algorithm, was not limited to capturing the training relationships, but could be used to infer the performances of models on various benchmarks. The observation that a logistic regression trained on the genetic distance matrix can accurately predict benchmark accuracy has the potential to accelerate the evaluation of new LLMs capabilities in a very computationally efficient manner. Overall, our method provides a robust and insightful analysis of the history, relationships, and performance of Large Language Models, even in cases where detailed training information is not publicly available.

Despite these promising results, it is important to acknowledge the inherent limitations of applying the genetic metaphor to LLMs. Phylogenetic algorithms, traditionally designed for biological analysis where common ancestors are not included among the tested species, face challenges when applied to LLMs, where common ancestors are present among the studied models. Furthermore, chat interfaces complicate the acquisition of reliable genetic material. Nonetheless, this work lays the foundation for further studies aimed at refining these algorithms to better fit the LLM framework and chat models. Our study did not explore the effect of temperature, and while our results were consistent across two sets of genes (and more in Appendix J), examining an even broader range of genes could provide additional insights. Additionally, while the predictive results for benchmark scores are promising (roughly 50% of the variance explained) and could be practically applied to estimate the capabilities of new models, it remains room for improvement (a possible venue being using multiple sets of genes in the evaluation).

Lastly, similarity matrices serve as versatile tools with numerous applications in the study and optimization of large language models (LLMs). For instance, in our investigation of model quantization, we discovered that as the size of the model increases, the quantized version more closely approximates the original model (see Appendix H). Additional fields in which PhyloLM could provide very good insights could also include model merging (Goddard et al., 2024) and scaling laws but we leave it for further research.

## 5    ACKNOWLEDGMENTS

This work was granted access to the HPC/IA resources of [IDRIS HPE Jean Zay A100] under the allocation 2023- [AD011013693R1] made by GENCI. SP is supported by the European Research Council under the European Union's Horizon 2020 research and innovation program (ERC) (RaReMem: 101043804), the Agence National de la Recherche (CogFinAgent: ANR-21-CE23-0002-02; RELATIVE: ANR-21-CE37-0008- 01; RANGE: ANR-21-CE28-0024-01), the Alexander Von Humboldt foundation and a Google unrestricted gift. PYO is supported by ANR AI individual chair ANR-19-CHIA-0004.

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

APPENDIX

This Appendix provides additional materials for PhyloLM :

- Section A presents examples of 'genes' and 'alleles' for 3 LLMs.
- Section B shows the list of models included in this study with finetuning relationships, sizes and benchmark scores.
- Section C gives PhyloLM pseudo code.
- Section D outlines the results on the code set of genes.
- Section E discusses more in depth similarity matrices results, differences between models and proposes potential explanations for such observations.
- Section F provides results about benchmark prediction on each benchmark for each family of models.
- Section G represents dendrograms including both sets of genes together.
- Section H exhibits results on PhyloLM and model quantization.
- Section I gives an overview of dendrograms with the names of the models.

## A  EXAMPLES OF GENES AND ALLELES

Genes were extracted from the Open-Web-Math (Paster et al., 2023) and MBXP (Athiwaratkun et al., 2023) datasets. The Open-Web-Math dataset comprises 31,577 rows, from which we selected the first 20 to 100 characters from the 'text' column (refer to Section 2.2 for detailed extraction methods). The MBXP dataset contains 6,814 rows, from which genes were extracted from the 'canonical solution' column. Twenty of the shortest genes (to fit on the page) in the extracted gene sets are presented in Table 1.

Table 1: **Examples of 20 short genes extracted from both sets of genes and one allele sampled from 3 LLMs** : Llama 1 7B, Llama 2 7B and GPT4 turbo 03/14. ↵ stands for a newline and □ represents a space (except in genes where spaces are packed to only one space for simplicity). Only the shortest genes have been included to fit in the table. Longer genes are also present in the gene set.

(a) from open-web-math set of genes

| Genes | Llama1 7B | Llama2 7B | GPT-4 (0314) |
|---|---|---|---|
| # Ignatius and the P | igeo | ig↵↵ | arli |
| 1 $\begingroup$ Close | vote | vote | I'm□ |
| # Propositional Logic | Prop | /Mis | Prop |
| [texhax] environment | vari | 2↵De | An□e |
| ### Homes↵↵There are | 23□h | 219□ | seve |
| # Annual income of A a | pers | B□an | I□am |
| # Image Mosaicking¶↵↵# | Impl | Writ | Imag |
| # Physics (Version 8.4 | )↵↵P | )↵↵P | "Phy |
| # Solve the linear equatio | ...↵ | ##□S | To□s |
| [texhax] \mid Description↵↵ | [tex | I□am | The□ |
| # string.replace.regex↵↵Synt | akti | acti | ax:↵ |
| # Math Help - 2-norm of a ma | xtri | ##□P | To□f |
| # Math Help - matlab code hel | pe↵↵ | plin | I'd□ |
| Thank you for visiting nature | gift | tour | You' |
| size - Maple Help↵↵MTM↵↵ size↵ | The□ | \□beg | In□M |
| # In observing a Tetrahedron... | In□o | In□o | A□te |
| Previous issue · Next issue · | Volu | Arch | Arch |
| # All Questions↵↵1,524 questions↵ | All□ | ##□A | Unfo |
| ## [POJ2411]Mondriaan\'s Dream↵↵ 成 | 功↵↵□ | 绩排名： | 本题考察 |
| # How to prove that $C=\{x: Ax\le | 0\□}$ | 0\□}$ | b\□}$ |

(b) from MBXP set of genes

| Genes | Llama1 7B | Llama2 7B | GPT-4 (0314) |
|---|---|---|---|
| return 4 * a;↵} | temp | func | func |
| ↵ double s | =□1; | 1(do | I'm□ |
| ↵ res = []↵ for e | in□i | in□r | ach□ |
| ↵ // TODO: | □□□□ | 15.0 | Ther |
| return n > 0 ? n % 10 | 0□:□ | :□0; | +□Ma |
| return a + b + c;↵} | \□end |  | func |
| // Function to | hand | □□// | dete |
| if (monthnum3 == 6) | { ↵□ | {↵□□ | It□m |
| ↵ str = ''.join(tup1 | )↵□□ | if□t | If□y |
| ↵ res = tuple(list(te | .fie | .out | It□s |
| ↵ r = n%m↵ return (r) | prin | end↵ | def□ |
| ↵ seq_nums = [seq_nums[ | star | 1:]] | i]□f |
| double result = 0;↵ | if□( | doub | In□o |
| ↵ string result = | null | null | I□ca |
| $isSame = true;↵ fore | :↵□□ | ach□ | ach□ |
| return n * (2 * n - 1)↵} | //□T | /**↵ | func |
| let sortedArr = arr.sort | ((a, | {□a, | ((a, |
| $list1 = $list1 || [];↵ | $lis | $lis | if□( |
| return !a[0] == !a[n-1];↵} | bool | int□ | This |
| string result = "";↵ fo | obar | .ope | r□(i |

## B  MODELS INCLUDED IN THIS STUDY

Open access models were run on A100 80Gb GPUs (approximately one per 20B parameters) with default floating precision except for Falcon180B that was downsized from float32 to bfloat16 as it was too big to fit on 8 GPUs. OpenAI models were accessed through the Openai API. Claude models were acessed through the Anthropic AI API . Proprietary mistral models were accessed through the Mistral API. Proprietary Google models were accessed through the VertexAI API. Tokens were sampled with a temperature of 0.7 with no sampling restriction (topp=1).

The following table 2 includes all models tested in this study with their benchmark scores. Only the completion models have been included in the benchmark score prediction study. After this table is the table of chat models that were separated in dendrogram plots and not included in benchmark score prediction. They can be found in Table 3.

---

[1]From    HELM    (Liang    et    al.,    2023)    -    no    information    about    prompting    - https://crfm.stanford.edu/helm/classic/latest/#/models

[1]From GPT-Fathom (Zheng et al., 2023) - ARC(1 shot) - HellaSwag (1 shot) - MMLU (5 shot) - TruthfulQA (1 shot) - WinoGrande (1 shot) - GSM8K (8 shot CoT) - https://crfm.stanford.edu/helm/classic/latest/#/models

[1]From    HuggingFace    (HuggingFaceH4)    -    ARC(25    shot)    -    HellaSwag    (10    shot)    - MMLU (5    shot) - TruthfulQA (0    shot) - WinoGrande (5    shot) - GSM8K (5    shot) - https://huggingface.co/spaces/HuggingFaceH4/open_llm_leaderboard

Table 2: **List of completion LLMs included in the study**. They appear in the similarity matrices (see Fig 9 10) in the same order as presented in this table. For proprietary LLMs, names are taken from their respective API. For all the other models, they were downloaded from the huggingface hub with the same name as given here. B and M stand for billions and millions, $\varnothing$ for trained from scratch and ? means unknown or not officially communicated.

| Name | Family | Size | Parent | AR | HS | ML | TQ | WG | GS |
|---|---|---|---|---|---|---|---|---|---|
| llama-7b | llama 1-7 | 7B | $\varnothing$ | 50.9 | 77.8 | 35.6 | 34.3 | 71.4 | 8.03 |
| alpaca-7b | llama 1-7 | 7B | llama-7b | 52.0 | 76.9 | 41.4 | 37.5 | 69.4 | 1.44 |
| wizard-7b | llama 1-7 | 7B | llama-7b | ? | ? | ? | ? | ? | ? |
| vicuna-7b-v1.1 | llama 1-7 | 7B | llama-7b | ? | ? | ? | ? | ? | ? |
| vicuna-7b-v1.3 | llama 1-7 | 7B | llama-7b | 50.4 | 76.9 | 48.1 | 47.0 | 70.4 | 5.68 |
| baize-7b | llama 1-7 | 7B | llama-7b | 48.9 | 75.0 | 39.6 | 41.3 | 71.1 | 4.16 |
| chimera-inst-chat-7b | llama 1-7 | 7B | llama-7b | ? | ? | ? | ? | ? | ? |
| llama-13b | llama 1-13 | 13B | $\varnothing$ | 56.1 | 80.9 | 47.6 | 39.4 | 76.2 | 7.58 |
| vicuna-13b-v1.1 | llama 1-13 | 13B | llama-13b | 52.7 | 80.1 | 51.9 | 52.0 | 74.1 | 8.64 |
| vicuna-13b-v1.3 | llama 1-13 | 13B | llama-13b | 54.6 | 80.4 | 52.8 | 52.1 | 74.8 | 10.7 |
| openchat_v2 | llama 1-13 | 13B | llama-13b | 57.1 | 81.1 | 50.5 | 49.5 | 76.2 | 9.09 |
| openchat_v2_w | llama 1-13 | 13B | llama-13b | 57.3 | 81.2 | 50.1 | 50.6 | 75.9 | 8.41 |
| chimera-inst-chat-13b | llama 1-13 | 13B | llama-13b | 55.3 | 78.9 | 50.5 | 50.1 | 73.9 | 8.18 |
| llama-2-7b-hf | llama 2-7 | 7B | $\varnothing$ | 53.0 | 77.7 | 43.7 | 38.9 | 74.0 | 14.4 |
| Orca-2-7b | llama 2-7 | 7B | llama-2-7b | 54.0 | 76.1 | 56.3 | 52.4 | 73.4 | 14.7 |
| tigerbot-7b-base | llama 2-7 | 7B | llama-2-7b | 47.6 | 72.0 | 45.1 | 42.2 | 69.6 | 10.8 |
| tigerbot-7b-chat | llama 2-7 | 7B | tigerbot-7b-base | ? | ? | ? | ? | ? | ? |
| OpenHermes-7B | llama 2-7 | 7B | llama-2-7b | 56.1 | 78.3 | 48.6 | 44.9 | 74.5 | 5.00 |
| vicuna-7b-v1.5 | llama 2-7 | 7B | llama-2-7b | 53.2 | 77.3 | 50.8 | 50.3 | 72.1 | 8.18 |
| llama-2-13b-hf | llama 2-13 | 13B | $\varnothing$ | 58.1 | 80.9 | 54.3 | 34.1 | 76.6 | 22.8 |
| openchat_v3.1 | llama 2-13 | 13B | llama-2-13b | 59.8 | 82.8 | 56.7 | 44.4 | 76.2 | 13.7 |
| openchat_v3.2 | llama 2-13 | 13B | llama-2-13b | 59.6 | 82.6 | 56.6 | 44.4 | 76.9 | 13.6 |
| OpenHermes-13B | llama 2-13 | 13B | llama-2-13b | 60.1 | 82.1 | 56.1 | 45.9 | 75.4 | 11.5 |
| vicuna-13b-v1.5 | llama 2-13 | 13B | llama-2-13b | 57.0 | 81.2 | 56.6 | 51.5 | 74.6 | 11.2 |
| openchat_v3.2_super | llama 2-13 | 13B | llama-2-13b | 59.8 | 82.5 | 55.8 | 42.2 | 75.9 | 13.4 |
| tigerbot-13b-base-v1 | llama 2 13 | 13B | llama-2-13b | ? | ? | ? | ? | ? | ? |
| tigerbot-13b-base-v2 | llama 2 13 | 13B | llama-2-13b | ? | ? | ? | ? | ? | ? |
| tigerbot-13b-chat-v1 | llama 2 13 | 13B | tigerbot-13b-base-v1 | ? | ? | ? | ? | ? | ? |
| tigerbot-13b-chat-v2 | llama 2 13 | 13B | tigerbot-13b-base-v2 | ? | ? | ? | ? | ? | ? |
| tigerbot-13b-chat-v3 | llama 2 13 | 13B | tigerbot-13b-base-v2 | ? | ? | ? | ? | ? | ? |
| tigerbot-13b-chat-v4 | llama 2 13 | 13B | tigerbot-13b-base-v2 | ? | ? | ? | ? | ? | ? |
| bloom-3b | bloom | 3B | $\varnothing$ | 35.7 | 54.3 | 26.5 | 40.5 | 57.6 | 1.51 |
| bloom-7b | bloom | 7B | $\varnothing$ | 41.1 | 61.9 | 26.2 | 38.8 | 65.4 | 1.36 |
| bloomz-3b | bloom | 3B | bloom-3b | 36.8 | 54.9 | 32.9 | 40.3 | 57.1 | 0.0 |
| bloomz-7b | bloom | 7B | bloom-7b | 42.4 | 63.0 | 37.8 | 45.2 | 64.6 | 0.07 |
| tigerbot-7b-base-v1 | bloom | 7B | bloom-7b | ? | ? | ? | ? | ? | ? |
| tigerbot-7b-base-v2 | bloom | 7B | bloom-7b | ? | ? | ? | ? | ? | ? |
| tigerbot-7b-sft-v1 | bloom | 7B | tigerbot-7b-base-v1 | ? | ? | ? | ? | ? | ? |
| tigerbot-7b-sft-v2 | bloom | 7B | tigerbot-7b-base-v2 | ? | ? | ? | ? | ? | ? |
| phoenix-inst-chat-7b | bloom | 7B | bloom-7b | ? | ? | ? | ? | ? | ? |
| bloom-176b | bloom | 176B | $\varnothing$ | 50.4 | 76.4 | 30.8 | 39.7 | 72.0 | 6.89 |

| Name | Family | Size | Parent | AR | HS | ML | TQ | WG | GS |
|---|---|---|---|---|---|---|---|---|---|
| pythia-70m | pythia | 70M | ∅ | 21.5 | 27.2 | 25.9 | 47.0 | 51.4 | 0.30 |
| pythia-160m | pythia | 160M | ∅ | 22.7 | 30.3 | 24.9 | 44.2 | 51.5 | 0.22 |
| pythia-410m | pythia | 410M | ∅ | 26.1 | 40.8 | 27.2 | 41.2 | 53.1 | 0.68 |
| pythia-1.4b | pythia | 1.4B | ∅ | 31.4 | 52.8 | 25.8 | 38.8 | 58.0 | 1.51 |
| pythia-2.8b | pythia | 2.8B | ∅ | 36.2 | 60.6 | 26.7 | 35.5 | 60.2 | 0.83 |
| pythia-6.9b | pythia | 6.9B | ∅ | 41.2 | 67.0 | 26.4 | 35.1 | 64.0 | 1.66 |
| pythia-12b | pythia | 12B | ∅ | 39.5 | 68.8 | 26.7 | 31.8 | 64.1 | 1.74 |
| dolly-v2-3b | pythia | 3B | pythia-2.8b | 25.2 | 26.5 | 24.6 | ? | 59.4 | 1.06 |
| dolly-v2-7b | pythia | 7B | pythia-6.9b | 44.5 | 69.6 | 25.1 | 34.8 | 60.0 | 1.13 |
| dolly-v2-12b | pythia | 12B | pythia-12b | 42.4 | 72.5 | 25.9 | 33.8 | 60.8 | 1.21 |
| oasst-sft-4-pythia-12b-epoch-3.5 | pythia | 12B | pythia-12b | 45.7 | 68.5 | 26.8 | 37.8 | 65.9 | 3.03 |
| opt-125m | opt | 125M | ∅ | 22.8 | 31.4 | 26.0 | 42.8 | 51.6 | 0.07 |
| opt-350m | opt | 250M | ∅ | 23.5 | 36.7 | 26.0 | 40.8 | 52.6 | 0.30 |
| opt-1.3b | opt | 1B | ∅ | 29.5 | 54.5 | 24.9 | 38.7 | 59.7 | 0.15 |
| opt-2.7b | opt | 3B | ∅ | 33.9 | 61.4 | 25.4 | 37.4 | 61.9 | 0.22 |
| opt-6.7b | opt | 7B | ∅ | 39.1 | 68.6 | 24.5 | 35.1 | 65.9 | 0.98 |
| opt-13b | opt | 13B | ∅ | 39.9 | 71.2 | 24.8 | 34.0 | 68.5 | 1.74 |
| opt-30b | opt | 30B | ∅ | 43.2 | 74.0 | 26.6 | 35.1 | 70.6 | 2.19 |
| opt-66b | opt | 66B | ∅ | 46.3 | 76.2 | 26.9 | 35.4 | 70.0 | 1.66 |
| Mistral-7B-v0.1 | mistral | 7B | ∅ | 59.9 | 83.3 | 64.1 | 42.1 | 78.6 | 37.0 |
| Mistral-7B-Instruct-v0.1 | mistral | 7B | Mistral-7B-v0.1 | 54.5 | 75.6 | 55.3 | 56.2 | 73.7 | 14.2 |
| Mistral-7B-Instruct-v0.2 | mistral | 7B | Mistral-7B-v0.1 | 63.1 | 84.8 | 60.7 | 68.2 | 77.1 | 40.0 |
| zephyr-7b-alpha | mistral | 7B | Mistral-7B-v0.1 | 61.0 | 84.0 | 61.3 | 57.9 | 78.6 | 14.0 |
| zephyr-7b-beta | mistral | 7B | Mistral-7B-v0.1 | 62.4 | 84.3 | 60.7 | 57.8 | 77.1 | 29.0 |
| docsgpt-7b-mistral | mistral | 7B | zephyr-7b-beta | ? | ? | ? | ? | ? | ? |
| openchat_3.5 | mistral | 7B | Mistral-7B-v0.1 | 62.4 | 83.9 | 62.8 | 45.4 | 81.0 | 25.7 |
| TenyxChat-7B-v1 | mistral | 7B | openchat_3.5 | 65.6 | 85.5 | 64.8 | 51.2 | 80.5 | 63.0 |
| MedChat3.5 | mistral | 7B | openchat_3.5 | ? | ? | ? | ? | ? | ? |
| neural-chat-7b-v3 | mistral | | | 67.1 | 83.2 | 62.2 | 58.7 | 78.0 | 1.21 |
| neural-chat-7b-v3-1 | mistral | 7B | Mistral-7B-v0.1 | 65.6 | 83.5 | 62.1 | 59.4 | 78.6 | 20.0 |
| OpenHermes-2-Mistral-7B | mistral | 7B | Mistral-7B-v0.1 | 63.0 | 83.8 | 63.4 | 50.2 | ? | ? |
| OpenHermes-2.5-Mistral-7B | mistral | 7B | Mistral-7B-v0.1 | 64.9 | 84.1 | 63.6 | 52.2 | 78.0 | 26.0 |
| H4rmoniousAnthea | mistral | 7B | OpenHermes-2.5-Mistral-7B | ? | ? | ? | ? | ? | ? |
| NeuralHermes-2.5-Mistral-7B | mistral | 7B | OpenHermes-2.5-Mistral-7B | 66.5 | 84.9 | 63.3 | 54.9 | 78.2 | 61.3 |
| Mixtral-8x7B-v0.1 | mistral | 8x7B | ∅ | 66.3 | 86.4 | 71.8 | 46.8 | 81.6 | 57.6 |
| Mixtral-8x7B-Instruct-v0.1 | mistral | 8x7B | ∅ | 70.1 | 87.5 | 71.3 | 64.9 | 81.0 | 61.1 |
| Qwen-1_8B | qwen | 2B | ∅ | ? | ? | ? | ? | ? | ? |
| Qwen-7B | qwen | 7B | ∅ | 51.3 | 78.4 | 59.8 | 47.7 | 72.6 | 44.9 |
| Qwen-14B | qwen | 14B | ∅ | 58.2 | 83.9 | 67.7 | 49.4 | 76.7 | 58.9 |
| Qwen-72B | qwen | 72B | ∅ | 65.1 | 85.9 | 77.3 | 60.1 | 82.4 | 70.4 |
| causallm-7b | qwen | 7B | Qwen-7B | 50.0 | 74.5 | 61.7 | 50.1 | 69.6 | 22.9 |
| causallm-14b | qwen | 14B | Qwen-14B | 56.6 | 79.0 | 65.8 | 47.7 | 74.9 | 58.6 |
| Qwen1.5-0.5B | qwen | 0.5B | ∅ | 31.4 | 49.0 | 39.3 | 38.2 | 57.2 | 16.3 |
| Qwen1.5-1.8B | qwen | 1.8B | ∅ | 37.8 | 61.4 | 46.7 | 39.4 | 60.2 | 33.5 |
| Qwen1.5-4B | qwen | 4B | ∅ | 48.4 | 71.5 | 56.5 | 47.2 | 66.2 | 52.2 |
| Qwen1.5-7B | qwen | 7B | ∅ | 54.1 | 78.5 | 61.9 | 51.0 | 71.2 | 53.5 |
| Qwen1.5-14B | qwen | 14B | ∅ | 56.5 | 81.0 | 69.3 | 52.0 | 73.4 | 67.6 |
| Qwen1.5-32B | qwen | 32B | ∅ | 63.5 | 85.0 | 74.2 | 57.3 | 81.4 | 61.1 |
| Qwen1.5-72B | qwen | 72B | ∅ | 65.8 | 85.9 | 77.2 | 59.6 | 83.0 | 65.7 |
| Qwen1.5-0.5B-Chat | qwen | 0.5B | Qwen1.5-0.5B | 30.5 | 44.0 | 33.8 | 42.9 | 54.6 | 7.65 |
| Qwen1.5-1.8B-Chat | qwen | 1.8B | Qwen1.5-1.8B | ? | ? | ? | ? | ? | ? |
| Qwen1.5-4B-Chat | qwen | 4B | Qwen1.5-4B | 43.2 | 69.7 | 55.5 | 44.7 | 64.9 | 2.42 |
| Qwen1.5-7B-Chat | qwen | 7B | Qwen1.5-7B | 55.8 | 78.5 | 61.6 | 53.6 | 67.7 | 13.1 |
| Qwen1.5-14B-Chat | qwen | 14B | Qwen1.5-14B | 58.7 | 82.3 | 68.5 | 60.3 | 73.3 | 30.8 |
| Qwen1.5-32B-Chat | qwen | 32B | Qwen1.5-32B | 66.0 | 85.4 | 74.9 | 66.9 | 77.1 | 7.05 |
| Qwen1.5-72B-Chat | qwen | 72B | Qwen1.5-72B | 68.5 | 86.4 | 77.4 | 63.8 | 79.0 | 20.3 |

| Name | Family | Size | Parent | AR | HS | ML | TQ | WG | GS |
|------|--------|------|--------|----|----|----|----|----|----|
| falcon-rw-1b | falcon | 1B | ∅ | 35.0 | 63.5 | 25.2 | 35.9 | 62.0 | 0.53 |
| falcon-rw-7b | falcon | 7B | ∅ | ? | ? | ? | ? | ? | ? |
| falcon-7b | falcon | 7B | ∅ | 47.8 | 78.1 | 27.7 | 34.2 | 72.3 | 4.62 |
| falcon-7b-instruct | falcon | 7B | falcon-7b | 46.1 | 70.8 | 25.8 | 44.0 | 67.9 | 4.70 |
| falcon-40b | falcon | 40B | ∅ | 61.9 | 85.2 | 56.9 | 41.7 | 81.2 | 21.4 |
| falcon-40b-instruct | falcon | 40B | falcon-40b | 61.6 | 84.3 | 55.4 | 52.5 | 79.7 | 34.3 |
| falcon-180b | falcon | 180B | ∅ | 69.1 | 88.8 | 69.5 | 45.1 | 86.8 | 45.9 |
| gemma-2b | gemma | 2B | ∅ | 48.3 | 71.7 | 41.7 | 33.0 | 66.2 | 16.9 |
| gemma-2b-it | gemma | 2B | gemma-2b | 43.9 | 62.6 | 37.6 | 45.8 | 60.9 | 5.45 |
| gemma-7b | gemma | 7B | ∅ | 61.0 | 82.4 | 66.0 | 44.9 | 78.4 | 52.7 |
| gemma-7b-it | gemma | 7B | gemma-7b | 51.4 | 71.9 | 53.5 | 47.2 | 67.9 | 29.1 |
| gemma-1.1-2b-it | gemma | 2B | gemma-2b-it | ? | ? | ? | ? | ? | ? |
| gemma-1.1-7b-it | gemma | 7B | gemma-7b-it | 60.0 | 76.1 | 60.9 | 50.7 | 69.6 | 42.9 |
| codegemma-2b | gemma | 2B | gemma-2b | 25.6 | 39.6 | 28.3 | 41.2 | 53.6 | 4.62 |
| codegemma-7b | gemma | 7B | gemma-7b | 53.9 | 76.7 | 56.5 | 38.0 | 69.6 | 45.4 |
| codegemma-7b-it | gemma | 7B | codegemma-7b | 53.8 | 72.5 | 56.0 | 48.5 | 66.2 | 52.4 |
| ada | gpt 3 | ? | ∅ | ? | 43.5 | 24.3 | 21.5 | ? | 0.6 |
| babbage | gpt 3 | ? | ∅ | ? | 55.5 | 23.3 | 18.8 | ? | 0.7 |
| curie | gpt 3 | ? | ∅ | ? | 68.2 | 24.3 | 23.2 | ? | 1.6 |
| davinci | gpt 3 | 175B | ∅ | 35.9 | 22.8 | 34.3 | 21.4 | 48.0 | 12.1 |
| davinci-instruct-beta | gpt 3 | 175B | davinci | 40.9 | 18.9 | 39.9 | 5.4 | 49.6 | 10.8 |
| text-ada-001 | gpt 3 | ? | ada | ? | 42.9 | 23.8 | 23.2 | ? | 0.4 |
| text-babbage-001 | gpt 3 | ? | babbage | ? | 56.1 | 22.9 | 23.3 | ? | 0 |
| text-curie-001 | gpt 3 | ? | curie | ? | 67.6 | 23.7 | 25.7 | ? | 0.6 |
| text-davinci-001 | gpt 3 | 175B | ? | 53.2 | 34.6 | 46.7 | 21.7 | 54.6 | 15.6 |
| text-davinci-002 | gpt 3.5 | 175B | code-davinci-002 | 75.7 | 64.9 | 62.1 | 47.8 | 65.5 | 47.3 |
| text-davinci-003 | gpt 3.5 | 175B | ? | 79.5 | 60.4 | 63.7 | 52.2 | 70.6 | 59.4 |
| babbage-002 | gpt 3.5 | ? | ? | ? | ? | ? | ? | ? | ? |
| davinci-002 | gpt 3.5 | ? | ? | ? | ? | ? | ? | ? | ? |
| gpt-3.5-turbo-instruct-0914 | gpt 3.5 | ? | ? | 83.6 | 82.8 | 69.6 | 59.4 | 68.0 | 75.8 |

Table 3: **List of chat LLMs included in the study**. They appear in the similarity matrices (see Fig 9 10) in the same order as presented in this table. For proprietary LLMs, names are taken from their respective API. For all the other models, they were downloaded from the huggingface hub with the same name as given here. B and M stand for billions and millions, $\varnothing$ for trained from scratch and ? means unknown or not officially communicated.

| Name | Family | Size | Parent |
|---|---|---|---|
| gpt-3.5-turbo-instruct-0914 | gpt 3.5 | ? | ? |
| gpt-3.5-turbo-0301 | gpt 3.5 | ? | ? |
| gpt-3.5-turbo-0613 | gpt 3.5 | ? | ? |
| gpt-3.5-turbo-1106 | gpt 3.5 | ? | ? |
| gpt-4-0314 | gpt 4 | ? | ? |
| gpt-4-0613 | gpt 4 | ? | ? |
| gpt-4-vision-preview | gpt 4 | ? | ? |
| gpt-4-1106-preview | gpt 4 | ? | ? |
| gpt-4-0125-preview | gpt 4 | ? | ? |
| claude-2.0 | claude | ? | ? |
| claude-2.1 | claude | ? | ? |
| claude-3-haiku-20240307 | claude | ? | ? |
| claude-3-sonnet-20240229 | claude | ? | ? |
| claude-3-opus-20240229 | claude | ? | ? |
| mistral-tiny-2312 (Mistral-7B-v0.1) | mistral | 7B | $\varnothing$ |
| mistral-small-2312 (Mixtral-8x7B-v0.1) | mistral | 8x7B | $\varnothing$ |
| mistral-small-2402 | mistral | ? | ? |
| mistral-medium-2312 | mistral | ? | ? |
| mistral-large-2402 | mistral | ? | ? |
| text-bison@001 | palm | ? | ? |
| text-bison@002 | palm | ? | ? |
| text-unicorn@001 | palm | ? | ? |
| chat-bison@001 | palm | ? | ? |
| chat-bison@002 | palm | ? | ? |
| gemini-1.0-pro | gemini | ? | ? |
| gemini-1.0-pro-001 | gemini | ? | ? |
| gemini-1.0-pro-002 | gemini | ? | ? |
| gemini-1.0-pro-vision | gemini | ? | ? |
| gemini-1.0-pro-vision-001 | gemini | ? | ? |

## C    PHYLOLM ALGORITHM

**Data:** $genes, N, LLMs$

```
/* Estimate P                                                    */
```
$P \leftarrow array(LLMs, genes, vocab\_size)$    #Zero filled array of given shape
**for** *LLM in LLMs* **do**
  **for** *gene in genes* **do**
    **for** *i ⩽ N* **do**
      $allele \leftarrow generate\_4\_characters(LLM, gene)$    #Generate 'allele'
      $P[LLM, gene, allele] \leftarrow P[LLM, gene, allele] + \frac{1}{N}$        #Update P
    **end**
  **end**
**end**

```
/* Compute similarity matrix S from P                            */
```
$S \leftarrow array(LLMs, LLMs)$
**for** *LLM1 in LLMs* **do**
  **for** *LLM2 in LLMs* **do**

$$S[LLM1, LLM2] \leftarrow \frac{\sum_{g \in G} \sum_{a \in A_g} P[LLM1, g, a] P[LLM2, g, a]}{\sqrt{(\sum_{g \in G} \sum_{a \in A_g} P[LLM1, g, a]^2)(\sum_{g \in G} \sum_{a \in A_g} P[LLM2, g, a]^2)}}$$

  **end**
**end**

```
/* Compute distance matrix D from S                             */
```
$D \leftarrow -\log(S)$
**return** $S, D$

**Algorithm 1:** PhyloLM

# D  RESULTS REPLICATION ON THE CODE SET OF GENES

In the main text we discussed the results obtained on the math set of genes and analysed dendrograms (see Section 3.2.1). Here we discuss results obtained on the code set of genes about hyperparameter search for PhyloLM, ground truth tree reconstruction and dendrograms. The benchmark prediction score (Figure 5) already includes code benchmark. For more details about individual benchmarks and sets of genes, see Appendix F.

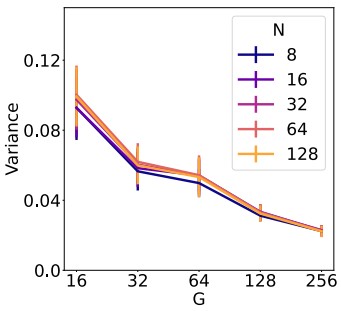
(a) Variability of distance matrices

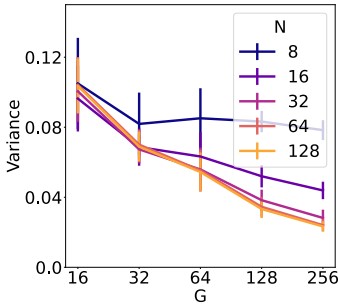
(b) Distance to the high precision matrix

Figure 6: **Hyperparameters impact on distance matrices in the code set of genes**. (a) shows the variability of distance matrices for different number of genes G and number of probes N in the math benchmark. Each set of genes of specified size contains different and independent genes from the other matrices for a total of 8 distance matrix for each data point in the figure. (b) shows the distance to the high precision matrix made of 2048 genes and N=128 in the math benchmark. Errorbars represent the standard error of the mean.

## D.1  HYPERPARAMETER INFLUENCE

We conducted the same experiments as described in Section 2.3. We first ran the variability analysis yielding the same results as in the math set of genes (see Figure 6 for code results and 2 for math set of genes): the variability decreases as $G$ increases but $N$ doesn't affect very much this value.

About the approximation of the high-precision matrix we get the same result as well : for each value of $N$, there is a value of $G$ from which increasing $G$ doesn't make the approximation very much better. One difference can be noted : with the math set of genes the approximation completely saturates while in the code set of genes it still improves a little bit. The same tradeoff of $G = 128$ and $N = 32$ can be found in this plot.

## D.2  PHYLOGENETIC TREE RECONSTRUCTION

When trying to reconstruct part of the Mistral Family we get similar results compared to the math set of genes (see Figure 7 for the code reconstruction and 3 for the math reconstruction). However, the root of the tree is not the right one. Indeed, the dendrogram algorithm, NJ (Saitou & Nei, 1987) constructs an unrooted tree that is then rooted for plotting. Here it failed to find the right root. Nonetheless, except the root choice, the structure of the tree is identical to the ground truth. That is also why we choose to plot unrooted trees in the paper as choosing a root for all the LLMs evolution is very artificial.

## D.3  DENDROGRAMS

Let's first look at the dendrogram (see Figure 8), one can notice very clear clusters for all completion families like in the math set of genes (see Figure 4). Indeed, all the llama branches are clustered together, Mistral, Gemma, Pythia, Qwen and Bloom models have very distinct branches in the phylogenetic tree. On the other hand, GPT-3, OPT and some Falcon models seem to share more than the other families like in the math similarity matrix. However, they still appear a little more different than in the math dendrogram. More details are given in the comparison of similarity matrices in Appendix E.

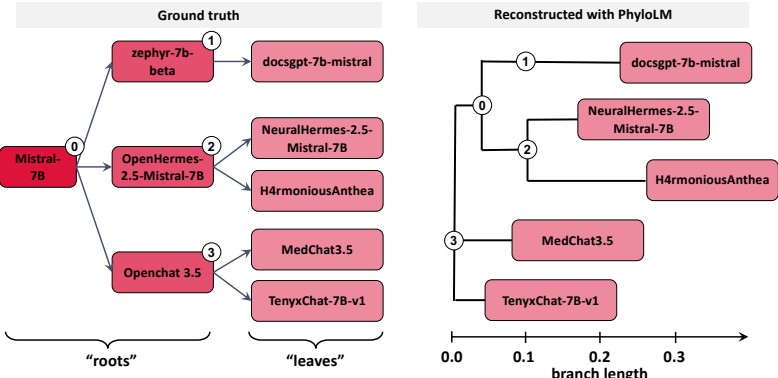

Figure 7: **Phylogenetic tree reconstruction on the code set of genes**. Left: ground truth concerning the relation of some LLMs of the Mistral family. Right is the reconstruction from the phylogenetic algorithm on the 'code' set of genes for the five latest models of this family ("leaves" of the phylogenetic tree) on which we run PhyloLM. Right: phylogenetic tree reconstructed by PhyloLM when given as input the 5 "leaves" models. The numerical labels (0:3) map the true common ancestors (on the right, "ground truth") to the inferred ones (on the left, "reconstructed"). It can be seen that the true and the reconstructed trees are very similar to the exception of the root that is 0 in the ground truth but 3 in the reconstruction. If we reroot the tree as having 0 as root we would get the exact ground truth like with the math set of genes.

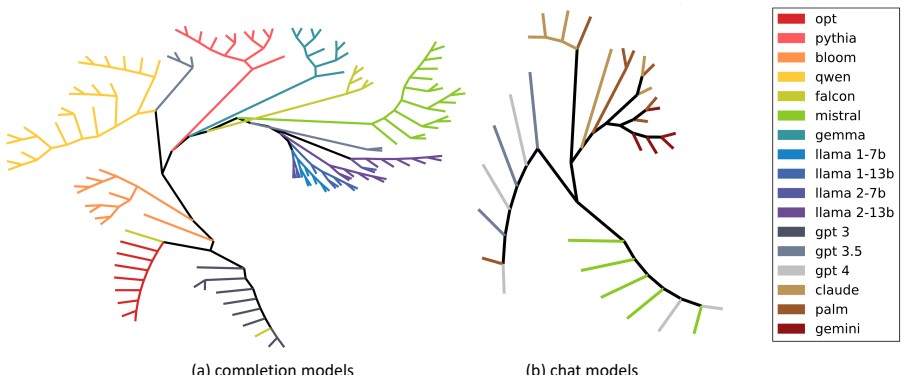

Figure 8: **Inferred phylogenetic tree of LLMs on the 'code' set of genes**. (a) completion models include 111 open source models included in our study and the 14 openai completion models (b) chat models include additional proprietary models. Completion and chat models were separated because they are not comparable due to additional prompting from the API. Leaves represent models and branches are colored according to model families. Llama models have been split by version of the pretrained model and the number of parameters. Length of lines from a model (leaf) to another represents the distance between the two models.

# E    SIMILARITY MATRICES COMPARISON BETWEEN SETS OF GENES

Let's now dive in the similarity matrices. It is important to understand that these matrices are a little noisy due to the value of $N$ and $G$ used (see Section 3.1). Estimating this noise is challenging as it may change from a LLM to another and is very difficult to represent in figures. By estimating variance on the smallest gpt models (ada, babbage, text-ada-001, text-babbage-001, babbage-002) we found the variance of the hyperparameters used in this study to be around 0.04 (see Figure 2a for the math set of genes and Figure 6 for the code set of genes). A difference of such a tiny value is barely visible with naked eye on a 0 to 1 scale (the scale in which similarity matrices are plotted) meaning that every visible difference of shade between models in the similarity matrix is significant. Of course, this is just an approximation, however, running such a statistical analysis on all models would be far too expensive in practice thus we stayed with this approximation in the paper.

Let's first dive into the completion models similarity matrices (see Figure 9 for the math set of genes and Figure Figure 10 for the code set of genes). First we notice that the code similarity matrix is a lot brighter than the math similarity matrix. This is likely due to the fact that code is a very formatted language meaning less options are available to complete a context making all models appear a little bit more similar when sampling alleles. Now, let's dive into the matrix one family at a time.

In the next paragraphs, we will present observations from the different distance matrices and contrast the observations between the two sets of genes. Then, we will hypothesise what are the reasons why we observe these things and sometimes validate these hypotheses if the models are transparent enough about the training details. For proprietary models we will only hypothesise drawing a parallel with validated hypothesis on the open access models that are transparent enough.

**Llama**    The same subfamily clusters can be found in both similarity matrices with, on one side, tigerbot-7b-base and tigerbot-7b-chat that appear closer to each other than the average llama models. Indeed, the second one is finetuned from the first one. Additionally, tigerbot-13b-base-v1, tigerbot-13b-base-v2, tigerbot-13b-chat-v1, tigerbot-13b-chat-v2, tigerbot-13b-chat-v3 and tigerbot-13b-chat-v4 form an additional cluster as they are based on each other (see Table 2 in Appendix B). We can also find openchat_v2 and openchat_v2_w in a same subcluster as they are trained on similar data as well as openchat_v3.1 and openchat_v3.2.

**Pythia**    Very clear finetuning relationships can be found in Pythia in both sets of genes as the dolly models are based on the respective pythia model with the same size and each of the dolly models shows an above average similarity with their repective pythia model.

**OPT**    This family is very interesting as in both sets of genes these models appear fairly close to each other. However, in the code similarity matrix, the family shows a lot of dissimilarity with all other models which is not the case with the math set of genes showing peculiar coding skills. While in the math matrix, OPT shares similarities with Pythia and GPT-3, in the code genome only the smaller gpt-3 family models including ada, babbage, curie, davinci, text-ada-001, text-babbage-001, text-curie-001 and davinci-instruct-beta (but not text-davinci-001 that is finetuned from davinci-instruct-beta !) seem to share an above average similarity and all the other models look very different from OPT. The only information we have is that OPT training data include ThePile but we don't know much about GPT-3 models.

**Falcon**    Falcon-RW-1 still shows similarities with the OPT family in both sets of genes but a little less in the code set of genes. Intriguingly, still in both matrices, the distance between Falcon-7b and Falcon-7b-instruct (finetuned from Falcon-7B) seems larger than the distance between Falcon-7b and Falcon-40b suggesting probably an intensive instruct finetuning on coding tasks as it is not found in the math similarity matrix. The same observation stands for Falcon-40b and Falcon-40b-instruct.

**Mistral**    In the Mistral family, in both sets of genes, we find back the subfamilies used in the tree reconstruction : zephyr-alpha-7b, zephyr-beta-7b and docsgpt-7b-mistral showing a lot of similarity while on another hand we have openchat_3.5 with TenyxChat-7B-v1 and MedChat3.5 and finally OpenHermes-2.5-Mistral-7B with H4rmoniousAnthea and NeuralHermes-2.5-Mistral-7B. Additional subclusters can be found such as Mixtral models and neural-chat models.

**Qwen**   Qwen, in both similarity matrices shows very intricate structure but with a lower contrast. We notice a first subcluster for the original Qwen models with their finetunings in CausalLM. Each CausalLM shows a better ressemblance to the Qwen model from which they were finetuned. Then we have the 1.5 family showing a lot of similarity with their chat finetuned counterparts. The smallest model Qwen1.5-0.5B-Chat shows very little similarity to all other models in the matrix indicating maybe catastrophic forgetting to some extent during finetuning in both sets of genes. However, in the code set of genes it shows even more dissimilarity with Qwen1.5-Chat-1_8B while it is the closest model in the math matrix. We are not entirely sure as to what could be the origin of this observation.

**Gemma**   In the Gemma family there is a lot of finetuning structure in both matrices. Indeed lots of white dots are far from the diagonal meaning several models show an unusual similarity with other models. Starting with gemma-2b and gemma-7b appearing very similar even more than their instruction finetuned counterparts (gemma-2b-it and gemma-7b-it). On the other hand, these finetuned versions appear to be very similar to the version 1.1 they were finetuned in. Lastly, codegemma-7b appear to be very close to gemma-2b and gemma-7b but codegemma-2b is less similar to them, probably because of its lower size that lead to forgetting to some extent.

**GPT-3**   Finally in the GPT-3 family, still in all matrices, we notice a first cluster corresponding to the pretrained models ada, babbage, curie and davinci. Intestingly, davinci-instruct-beta an SFT finetuned model from davinci shows a lot of similarities to davinci in both sets of genes meaning it might have been finetuned on a low size dataset. Then their finetuned counterparts, text-ada-001, text-babbage-001, text-curie-001 and text-davinci-001 appear a lot farther from the pretrained models but we can trace back the model they were finetuned from from the similarity they share with the GPT-3 pretrained models. Then, in the math set of genes, text-davinci-001, text-davinci-002 and text-davinci-003 look vaguely similar but on the code genome, the last two appear extremely close and a little far from text-davinci-001. This might be linked to the fact that text-davinci-002 and text-davinci-003 have been finetuned from code-davinci-002 (trained on code) and not from davinci indicating different coding skills between the 1st version and the last 2 while all of them are fairly different in reasoning tasks (which was likely the object of their respective different finetuning). Going back to the similarities with OPT, we notice that, in coding tasks, text-davinci-001 doesn't show much similarities with OPT models while davinci-instruct-beta appear close to OPT. What is truely interesting here is that text-davinci-001 is either finetuned from davinci or from davinci-instruct-beta and this finetuning does not seem to share similarities with OPT on coding skills showing a potential emphasis on coding skills during this finetuning. Finally, babbage-002, davinci-002 and gpt3.5-turbo-instruct-0914 appear close to each other. While babbage-002 and davinci-002 might be pretrained, if we had to find a parent for gpt-3.5-turbo-instruct it would likely be davinci-002 as it is the closest model to it (but we don't have enough information to be sure of this).

Now that we have seen the completion models similarity matrix let's switch to the chat models (see Figure 12). These models essentially include proprietary models with very little to no transparency at all about the training details. As such we will have to completely guess the reasons for the observations we are going to make.

**GPT-3.5/GPT-4**   We discern two clusters in the GPT family : before 11/06 and after and in both sets of genes. Indeed the newest versions of gpt-4 show a lot of similarity with each other but are extremely dissimilar to the previous versions that have a lot with each other. Something important must have happened at this point in the training methods at OPENAI.

**Claude**   In claude the two generations of models are clearly discernible : claude 1 and 2 on one side and claude 3 on another side. Once more a big change has occurred in the training methods in between. Interestingly, the 3rd generation share a lot with bison and gemini models especially in the code context. More specifically with code-bison@001. This is not the case with the math set of genes. This may show some similarities in code training sets or data generation using another model.

**Mistral**   In the Mistral API we notice a strong similarity between mistral-small-2402 and mistral-large-2402 that were probably trained on the same training set. On another hand most models appear close to the last two gpt models more specifically these two models released on 02/24. On the contrary, mistral-medium-2312 seems a lot closer to the previous version of gpt-4 and gpt-3.5 showing maybe a similar inspiration in the training set or data generation using a model.

**Palm**    In the Palm family, we notice that text-bison@002 and code-bison@002 are almost identical suggesting a finetuning from one to another like it is the case in the gemma family (also from google) or that it is the same model.

**Gemini**    Finally, models in the gemini family appear almost identical to each other except gemini-1.0-pro-002. Palm appear fairly close to Gemini in both context showing maybe a shared training set to some extent (both models are from google) but in the code context they appear very close to Claude 3 models (maybe a similar training set or data generation using a model).

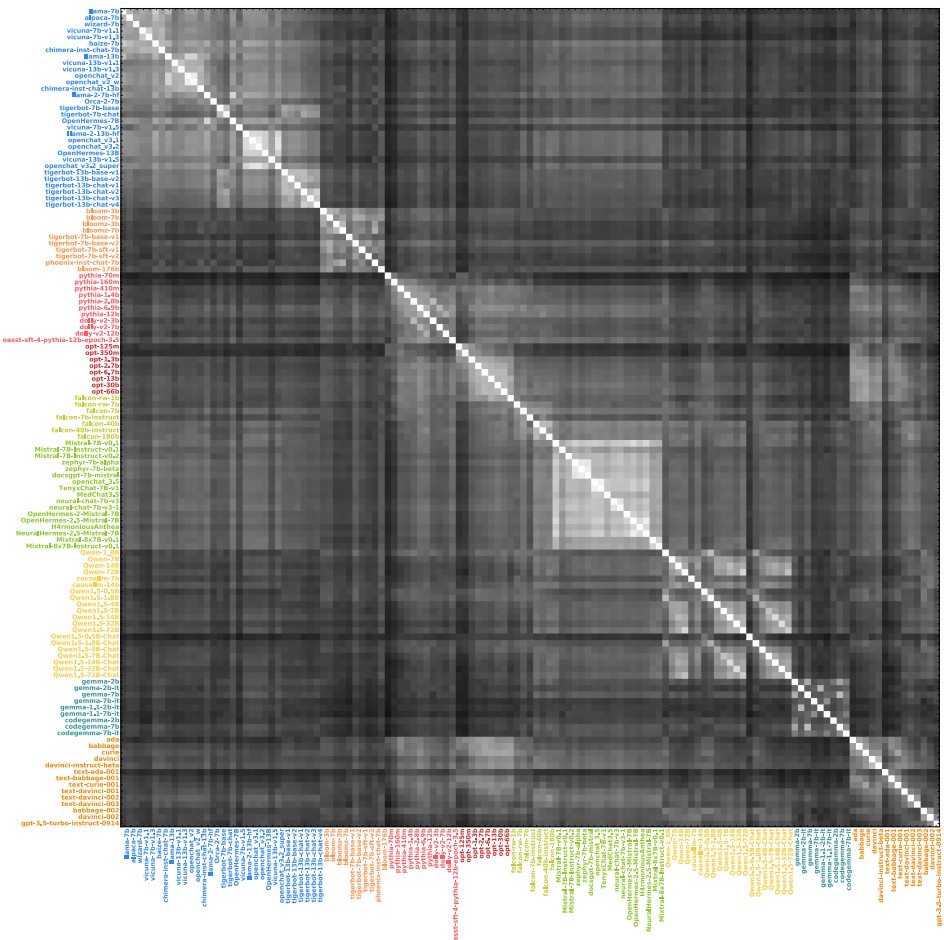

Figure 9: **Similarity matrix of completion LLMs on the math set of genes**

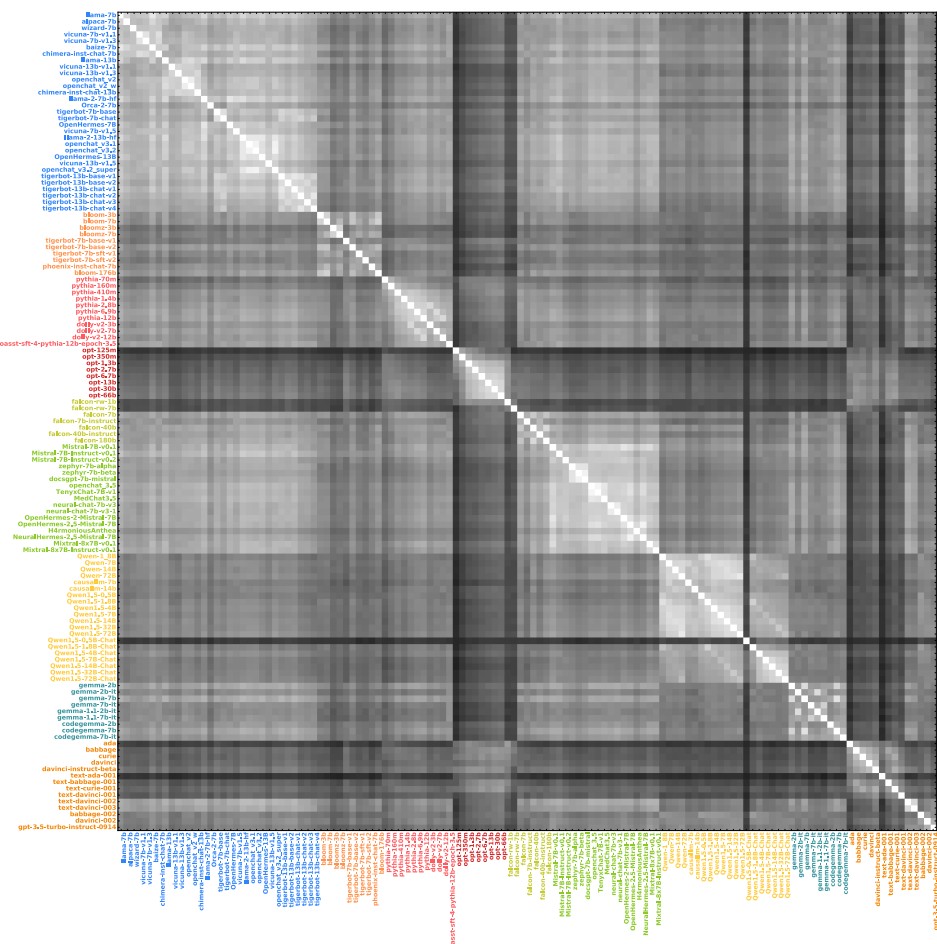

Figure 10: **Similarity matrix of completion LLMs on the code set of genes**

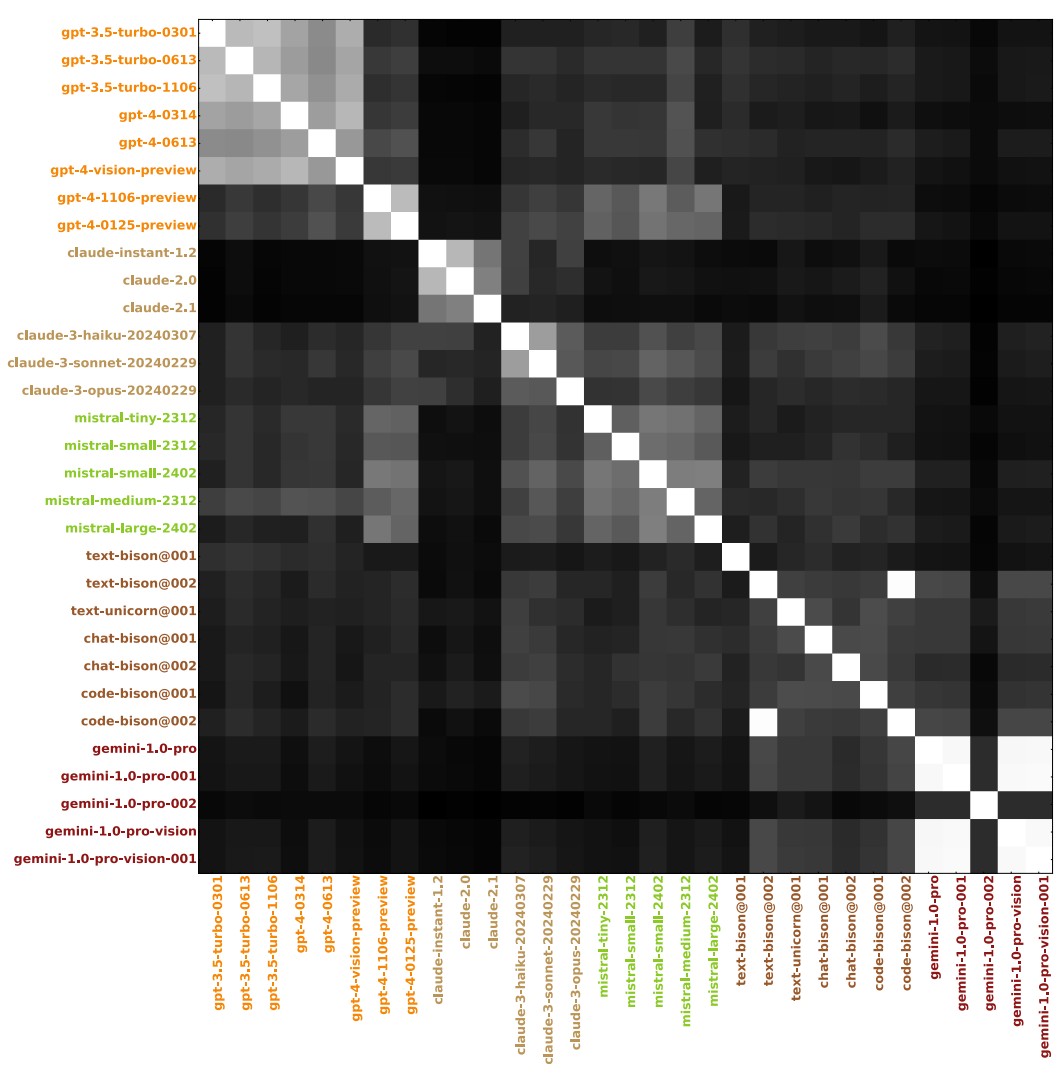

Figure 11: **Similarity matrix of chat LLMs on the math set of genes**

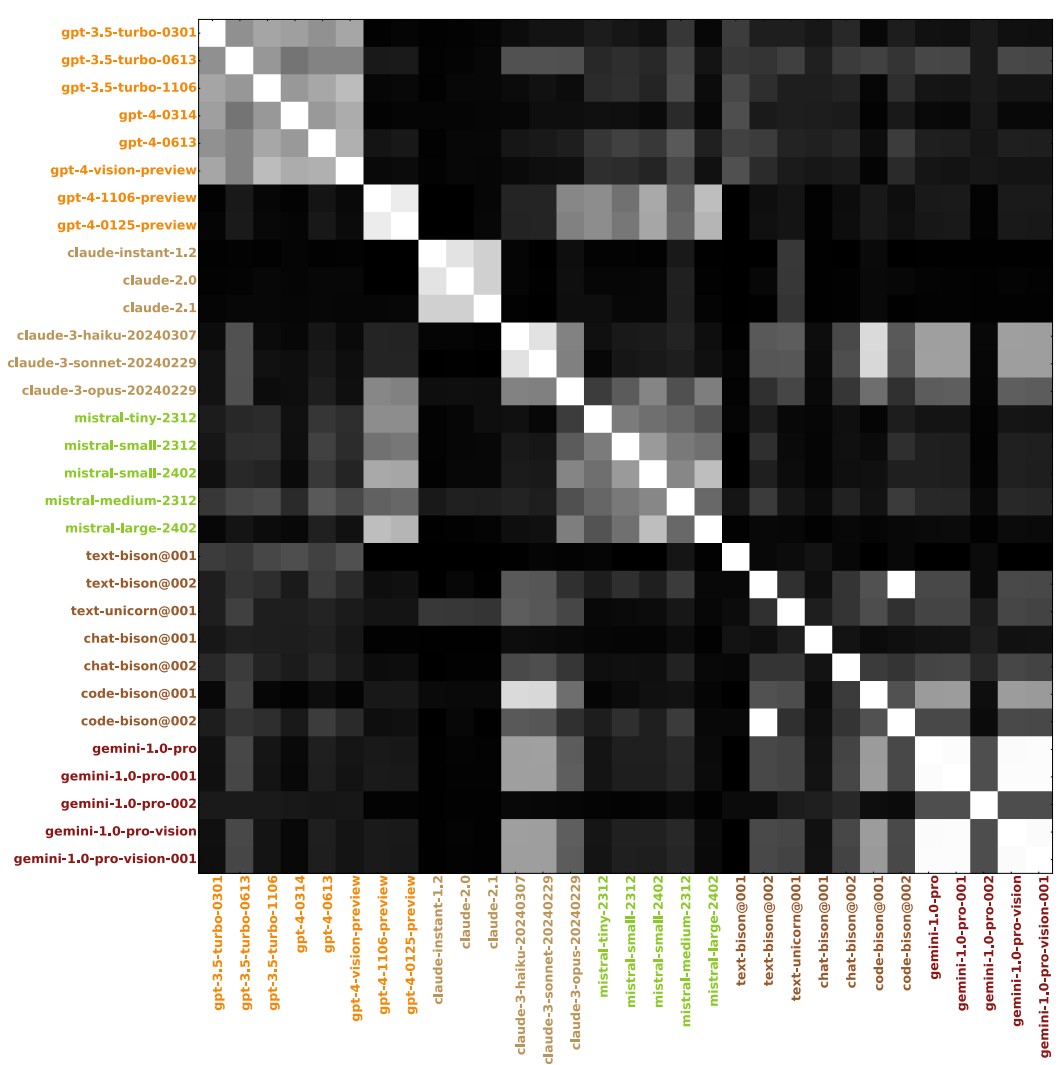

Figure 12: **Similarity matrix of chat LLMs on the code set of genes**

# F    BENCHMARK RESULTS

In the main paper only MMLU and ARC prediction results were presented. Here we show all the 6 benchmark predictions for both gene sets (Figure 5 for the math set of genes and Figure **??** for the code set of genes). Statistics are reported in Tables (Table 4 for the math set of genes and Table 5 for the code set of genes).

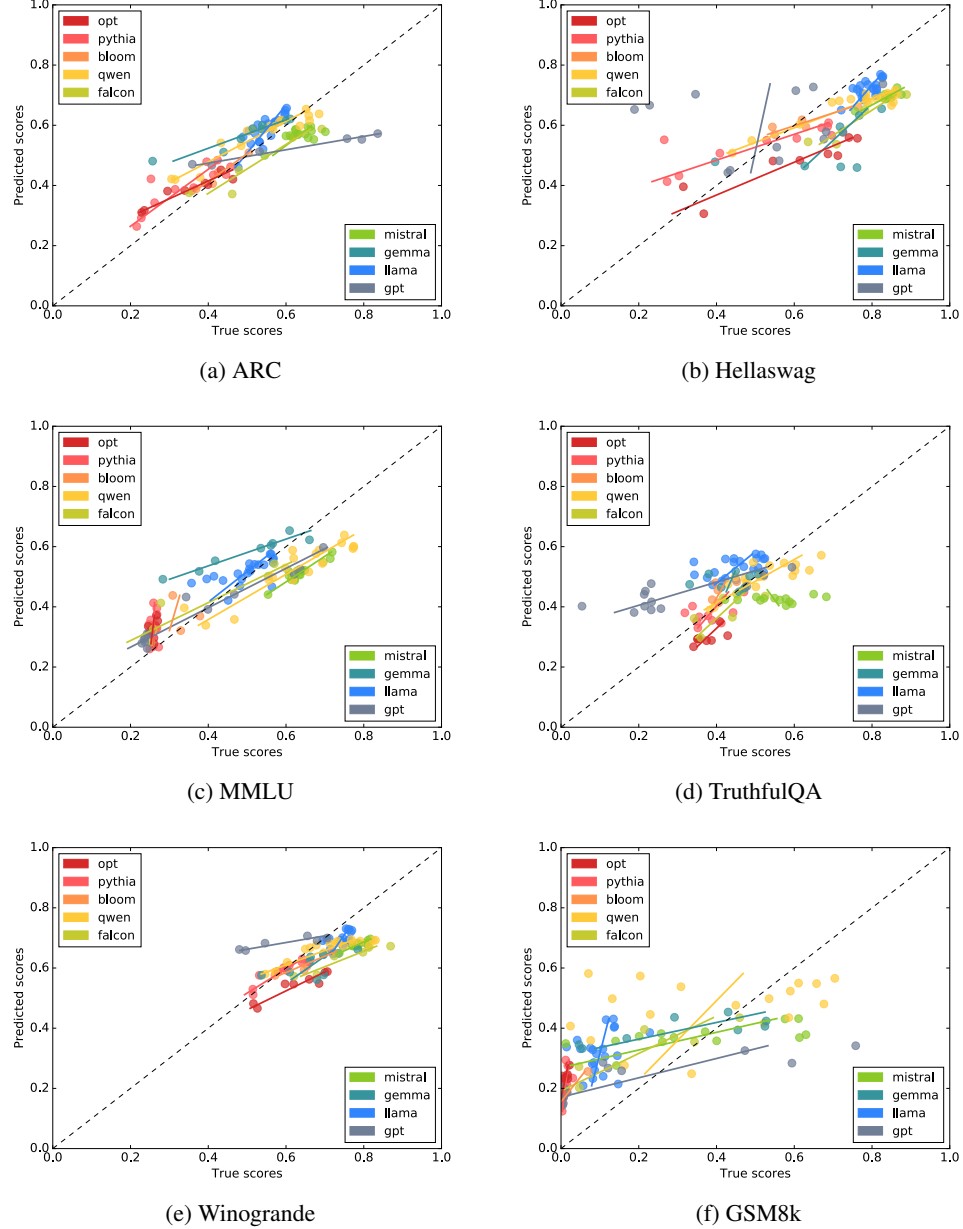

Figure 13: **Predictions of benchmark performances from the MLP compared to ground truth for every model(leave-one-out method) for the math set of genes** . Each point is the predicted score of the model when the regressor is trained on data from all other models but the given point. For the sake of clarity, names of models have been omitted and replace by the color of models from the dendrogram plot. Lines represent a linear regression between predicted and true labels within each family.

Table 4: **Statistics on benchmark prediction for the math set of 'genes'**. $\rho$ is the pearson correlation coefficient between the predicted labels and the true labels, p is the p-value for the student-test against 0 and N is the number of datapoint for this analysis.

| Benchmark | Family | $\rho$ | p | N | Benchmark | Family | $\rho$ | p | N |
|---|---|---|---|---|---|---|---|---|---|
|  | llama | 0.78 | <0.001 | 21 |  | llama | 0.66 | <0.001 | 21 |
|  | bloom | 0.98 | 0.002 | 5 |  | bloom | 0.94 | 0.015 | 5 |
|  | pythia | 0.84 | 0.001 | 11 |  | pythia | 0.81 | 0.002 | 11 |
|  | opt | 0.94 | <0.001 | 8 |  | opt | 0.89 | 0.002 | 8 |
| arc | falcon | 0.94 | 0.005 | 6 | hellaswag | falcon | 0.94 | 0.003 | 6 |
|  | mistral | 0.70 | 0.004 | 14 |  | mistral | 0.79 | <0.001 | 14 |
|  | qwen | 0.94 | <0.001 | 18 |  | qwen | 0.89 | <0.001 | 18 |
|  | gemma | 0.95 | <0.001 | 8 |  | gemma | 0.56 | 0.145 | 8 |
|  | gpt | 0.97 | <0.001 | 6 |  | gpt | 0.07 | 0.806 | 12 |
|  | all | 0.82 | <0.001 | 97 |  | all | 0.68 | <0.001 | 103 |
| Benchmark | Family | $\rho$ | p | N | Benchmark | Family | $\rho$ | p | N |
|  | llama | 0.61 | 0.002 | 21 |  | llama | 0.51 | 0.017 | 21 |
|  | bloom | 0.23 | 0.701 | 5 |  | bloom | 0.54 | 0.343 | 5 |
|  | pythia | 0.18 | 0.587 | 11 |  | pythia | 0.47 | 0.161 | 10 |
|  | opt | 0.31 | 0.450 | 8 |  | opt | 0.58 | 0.129 | 8 |
| mmlu | falcon | 0.93 | 0.005 | 6 | truthfulqa | falcon | 0.72 | 0.102 | 6 |
|  | mistral | 0.90 | <0.001 | 14 |  | mistral | 0.10 | 0.732 | 14 |
|  | qwen | 0.87 | <0.001 | 18 |  | qwen | 0.79 | <0.001 | 18 |
|  | gemma | 0.96 | <0.001 | 8 |  | gemma | 0.23 | 0.572 | 8 |
|  | gpt | 0.97 | <0.001 | 12 |  | gpt | 0.79 | 0.001 | 12 |
|  | all | 0.88 | <0.001 | 103 |  | all | 0.43 | <0.001 | 102 |
| Benchmark | Family | $\rho$ | p | N | Benchmark | Family | $\rho$ | p | N |
|  | llama | 0.60 | 0.003 | 21 |  | llama | 0.31 | 0.169 | 21 |
|  | bloom | 0.91 | 0.029 | 5 |  | bloom | 0.94 | 0.013 | 5 |
|  | pythia | 0.92 | <0.001 | 11 |  | pythia | 0.58 | 0.058 | 11 |
|  | opt | 0.94 | <0.001 | 8 |  | opt | 0.76 | 0.026 | 8 |
| winogrande | falcon | 0.93 | 0.005 | 6 | gsm8k | falcon | 0.90 | 0.012 | 6 |
|  | mistral | 0.90 | <0.001 | 13 |  | mistral | 0.63 | 0.019 | 13 |
|  | qwen | 0.87 | <0.001 | 18 |  | qwen | 0.30 | 0.215 | 18 |
|  | gemma | 0.55 | 0.149 | 8 |  | gemma | 0.82 | 0.012 | 8 |
|  | gpt | 0.91 | 0.011 | 6 |  | gpt | 0.77 | 0.003 | 12 |
|  | all | 0.72 | <0.001 | 96 |  | all | 0.66 | <0.001 | 102 |

Table 5: **Statistics on benchmark prediction for the code set of 'genes'**. $\rho$ is the pearson correlation coefficient between the predicted labels and the true labels, p is the p-value for the student-test against 0 and N is the number of datapoint for this analysis.

| Benchmark | Family | $\rho$ | p | N | Benchmark | Family | $\rho$ | p | N |
|---|---|---|---|---|---|---|---|---|---|
| | llama | 0.39 | 0.076 | 21 | | llama | 0.70 | <0.001 | 21 |
| | bloom | 0.95 | 0.012 | 5 | | bloom | 0.94 | 0.016 | 5 |
| | pythia | 0.84 | 0.001 | 11 | | pythia | 0.52 | 0.095 | 11 |
| | opt | 0.88 | 0.003 | 8 | | opt | 0.93 | <0.001 | 8 |
| arc | falcon | 0.97 | 0.001 | 6 | hellaswag | falcon | 0.90 | 0.014 | 6 |
| | mistral | 0.35 | 0.210 | 14 | | mistral | 0.09 | 0.739 | 14 |
| | qwen | 0.87 | <0.001 | 18 | | qwen | 0.83 | <0.001 | 18 |
| | gemma | 0.18 | 0.655 | 8 | | gemma | 0.15 | 0.714 | 8 |
| | gpt | 0.84 | 0.034 | 6 | | gpt | 0.15 | 0.637 | 12 |
| | all | 0.76 | <0.001 | 97 | | all | 0.50 | <0.001 | 103 |

| Benchmark | Family | $\rho$ | p | N | Benchmark | Family | $\rho$ | p | N |
|---|---|---|---|---|---|---|---|---|---|
| | llama | 0.67 | <0.001 | 21 | | llama | 0.71 | <0.001 | 21 |
| | bloom | 0.84 | 0.072 | 5 | | bloom | 0.65 | 0.230 | 5 |
| | pythia | 0.08 | 0.794 | 11 | | pythia | 0.39 | 0.256 | 10 |
| | opt | 0.09 | 0.831 | 8 | | opt | 0.27 | 0.516 | 8 |
| mmlu | falcon | 0.88 | 0.020 | 6 | truthfulqa | falcon | 0.71 | 0.106 | 6 |
| | mistral | 0.49 | 0.068 | 14 | | mistral | 0.09 | 0.753 | 14 |
| | qwen | 0.93 | <0.001 | 18 | | qwen | 0.68 | 0.001 | 18 |
| | gemma | 0.50 | 0.206 | 8 | | gemma | 0.77 | 0.024 | 8 |
| | gpt | 0.95 | <0.001 | 12 | | gpt | 0.80 | 0.001 | 12 |
| | all | 0.79 | <0.001 | 103 | | all | 0.58 | <0.001 | 102 |

| Benchmark | Family | $\rho$ | p | N | Benchmark | Family | $\rho$ | p | N |
|---|---|---|---|---|---|---|---|---|---|
| | llama | 0.48 | 0.024 | 21 | | llama | 0.20 | 0.377 | 21 |
| | bloom | 0.87 | 0.052 | 5 | | bloom | 0.52 | 0.359 | 5 |
| | pythia | 0.71 | 0.013 | 11 | | pythia | 0.39 | 0.229 | 11 |
| | opt | 0.90 | 0.001 | 8 | | opt | 0.60 | 0.110 | 8 |
| winogrande | falcon | 0.88 | 0.020 | 6 | gsm8k | falcon | 0.83 | 0.040 | 6 |
| | mistral | 0.53 | 0.062 | 13 | | mistral | 0.44 | 0.127 | 13 |
| | qwen | 0.75 | <0.001 | 18 | | qwen | 0.39 | 0.103 | 18 |
| | gemma | 0.08 | 0.839 | 8 | | gemma | 0.58 | 0.124 | 8 |
| | gpt | 0.74 | 0.092 | 6 | | gpt | 0.87 | <0.001 | 12 |
| | all | 0.50 | <0.001 | 96 | | all | 0.70 | <0.001 | 102 |

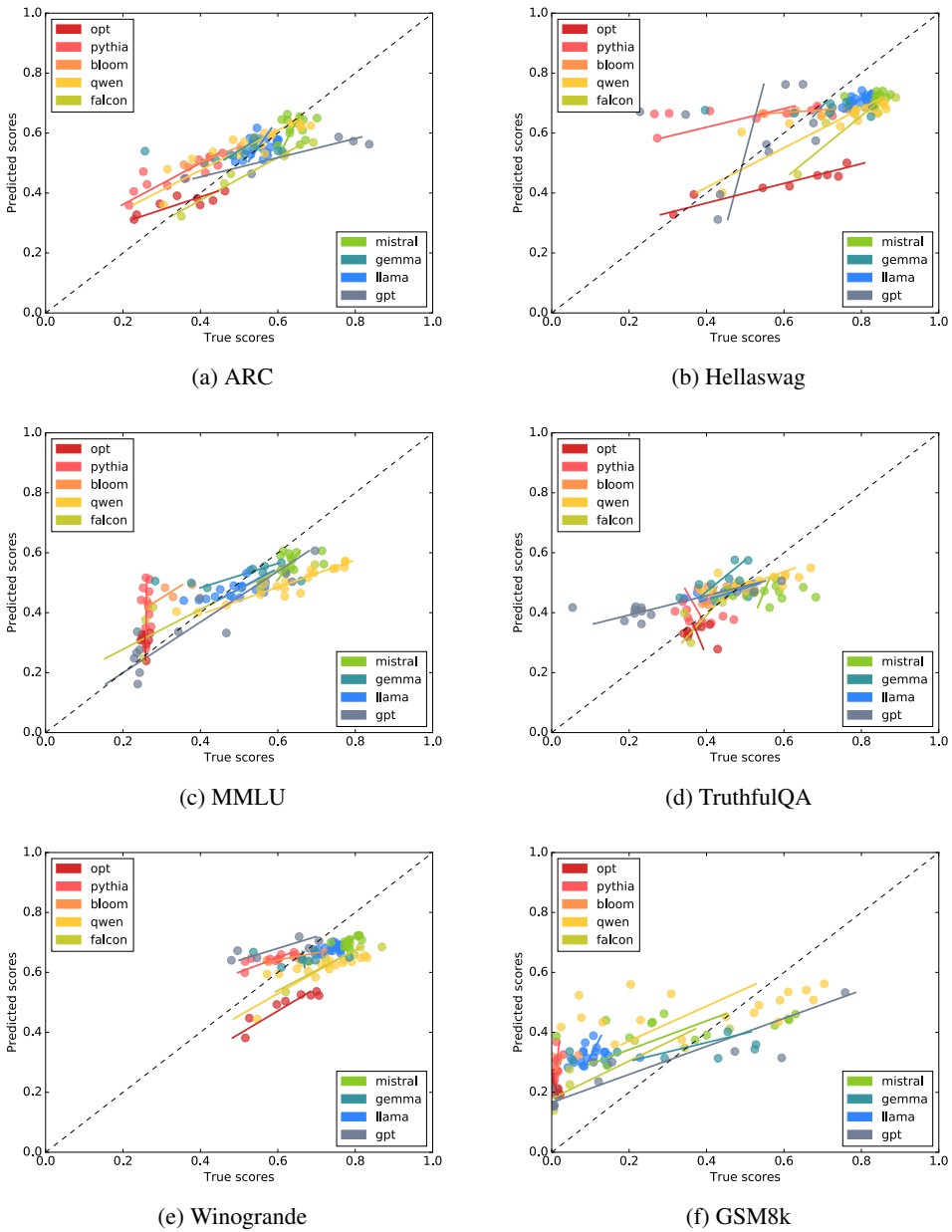

Figure 14: **Predictions of benchmark performances from the MLP compared to ground truth for every model(leave-one-out method) for the code set of genes** . Each point is the predicted score of the model when the regressor is trained on data from all other models but the given point. For the sake of clarity, names of models have been omitted and replace by the color of models from the dendrogram plot. Lines represent a linear regression between predicted and true labels within each family.

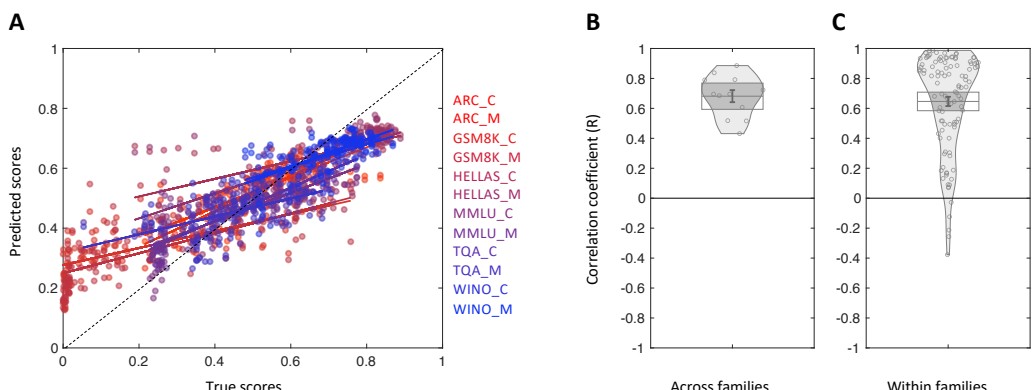

Figure 15: **Predictions from the logistic regression compared to ground truth for every model (leave one family out method)**. (a) Scatter plot of the correlation between the true scores in six benchmarks (names are given on the left of the plot) against the predicted scores using our methods based on the genetic distance calculated on either the "code" (*_C) or the "math" (*_M) set of "genes" (the plot presents individual LLMs as points and linear regression lines; colors of the panel correspond to the colors of the benchmarks' names). (b) Individual Pearson's correlation coefficients of the correlation between true and predicted scores across 6 benchmarks x 2 set of genes (leading to 12 combinations). (c) Individual Pearson's correlation coefficients of the correlation between true and predicted scores computed within each family and benchmarks (leading to 108 combinations). In (b) and (c) the horizontal line represents the mean, the dark vertical bar the standard error of the mean, the box delimitates the confidence interval and points are plotted within a probability density function.

## G GLOBAL FIGURES

In the main paper we claimed that separating completion and chat models is important as they do not return the allele to the same gene. Indeed, many private models add additional tokens to the prompt such as message markers for user and assistant that will be concatenated to the gene such that the allele returned doesn't correspond to the same gene as completion models. We still plotted the similarity matrices (see Figures 16 and 17) and dendrograms (see Figures 18 and 19) to show that, indeed, it doesn't capture the families very well.

As explained in the main text we can see that the Mistral 7B model prompted in a completion manner doesn't appear close to the same model from the Mistral API prompted in a chat format. A possible improvement to PhyloLM would be to prompt all completion models in a chat format to see whether it bridges the gap between completion and chat models. However depending on the prompting technique, some models are finetuned to interact with a specific message token whereas others might be more familiar with another prompting technique which could bias the comparison.

Looking at the figures (see Figure 16 for the math set of genes and Figure 17 for the code set of genes), one can see that, indeed, the similarity between completion and chat models (the separation starting at the second half of the gpt-family) is very low except for text-bison@001 and the first few gpt-3.5 and gpt-4 models for both sets of genes (maybe even clearer for the code set of genes). This can be seen further in the dendrograms as the chat models are all on the same branch and, despite having a lot of structure when plotting only the chat models dendrogram (see Figure 4 for the math set of genes and Figure 8 for the code set of genes) here not much is to be seen. Indeed, one has to bear in mind that the phylogenetic tree vizualisation plots in a 2D space information coming from a 150 dimensional space and is influenced by the variance in all the models to be plotted. As such, plotting models that are too different will lead to a vague representation while plotting a very specific set of models can lead to very precise discrimination in the history of the models (see Figure 3 for phylogenetic reconstruction from the math set of gene and Figure 7 from the code set of genes). That is why we splitted both types of models as the technical limitations linked to the interaction with some proprietary LLMs doesn't make it possible to extract precise genetic information from these models biasing the completion models from which we have precise genetic information.

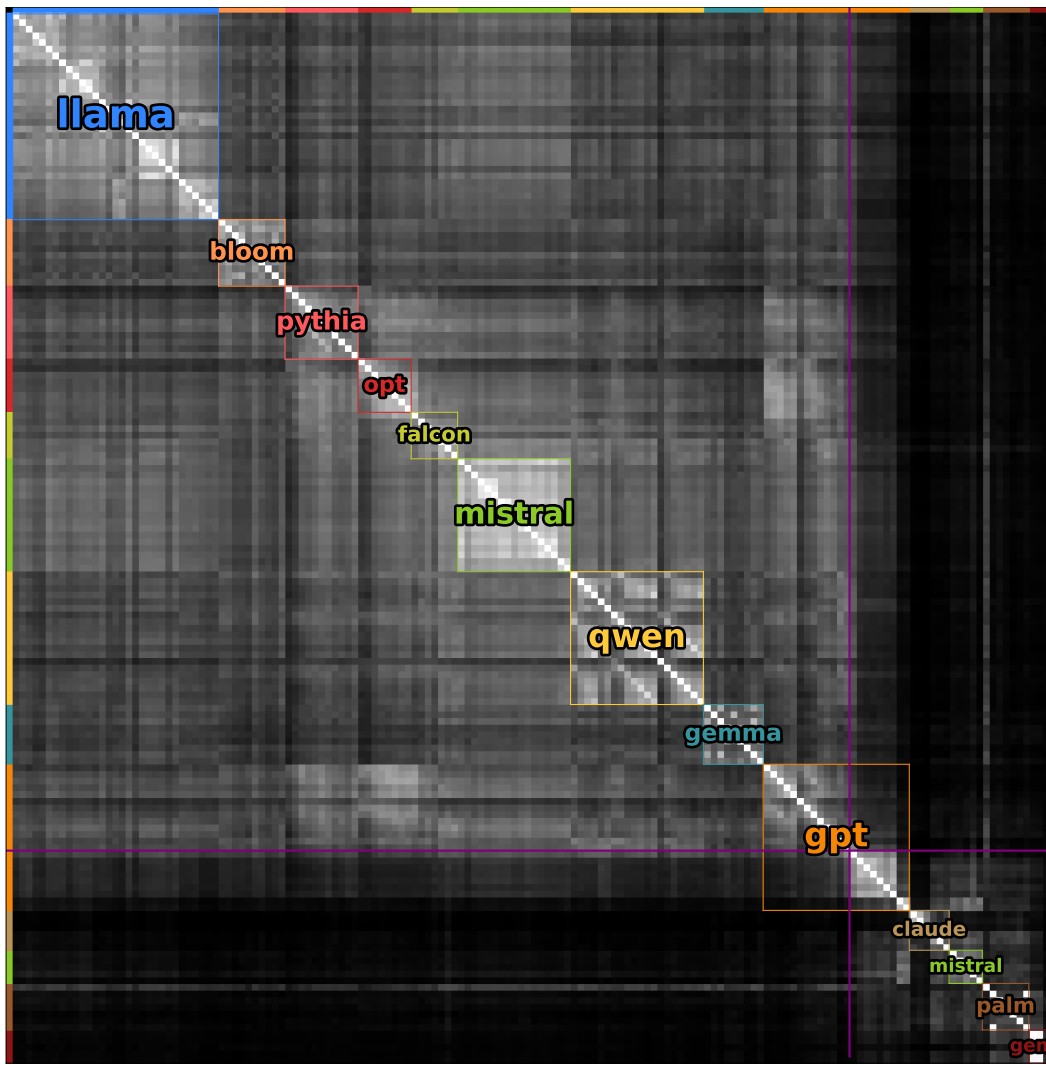

Figure 16: **Similarity matrix with all models (completion and chat) at once on the math set of genes**. All models included are shown in Appendix B. Purple lines split completion from chat models.

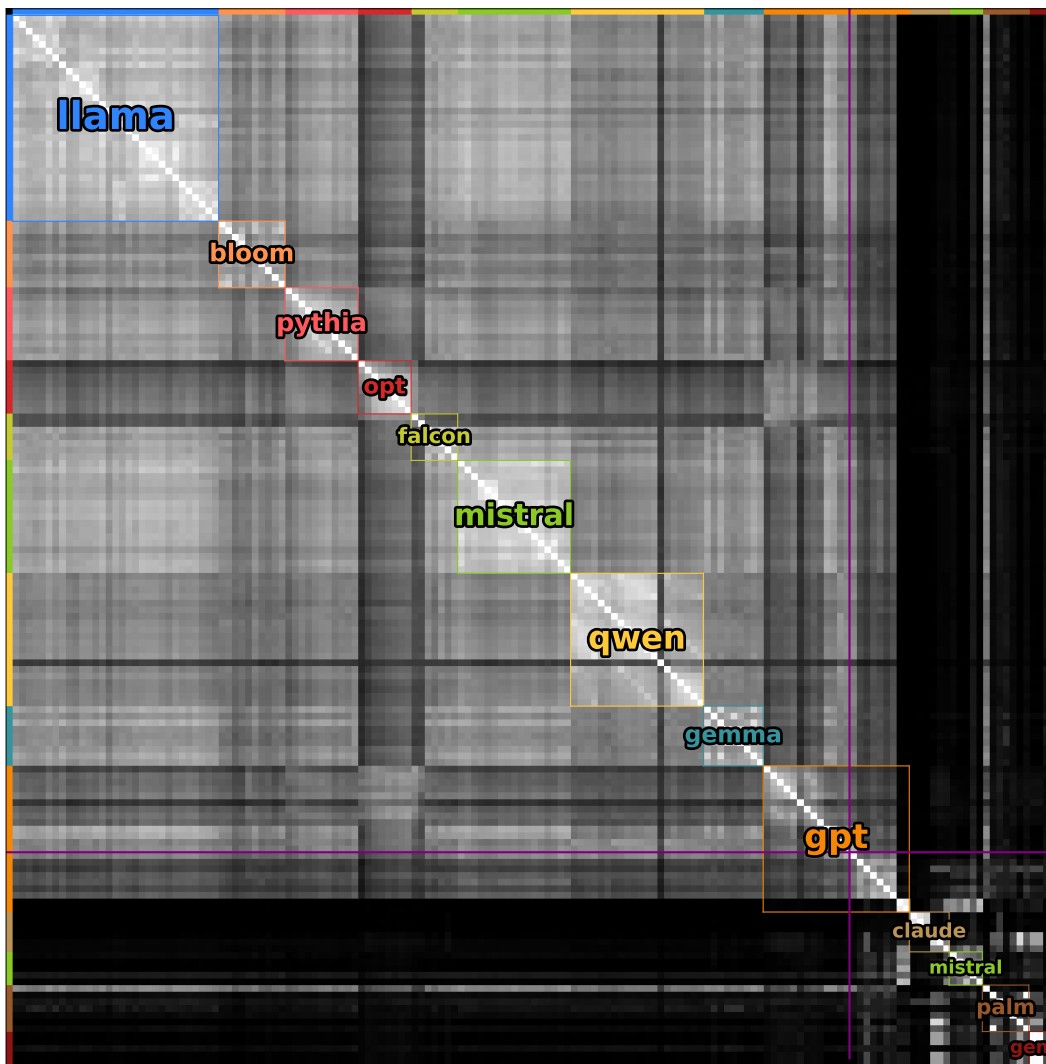

Figure 17: **Similarity matrix with all models (completion and chat) at once on the code set of genes**. All models included are shown in Appendix B. Purple lines split completion from chat models.

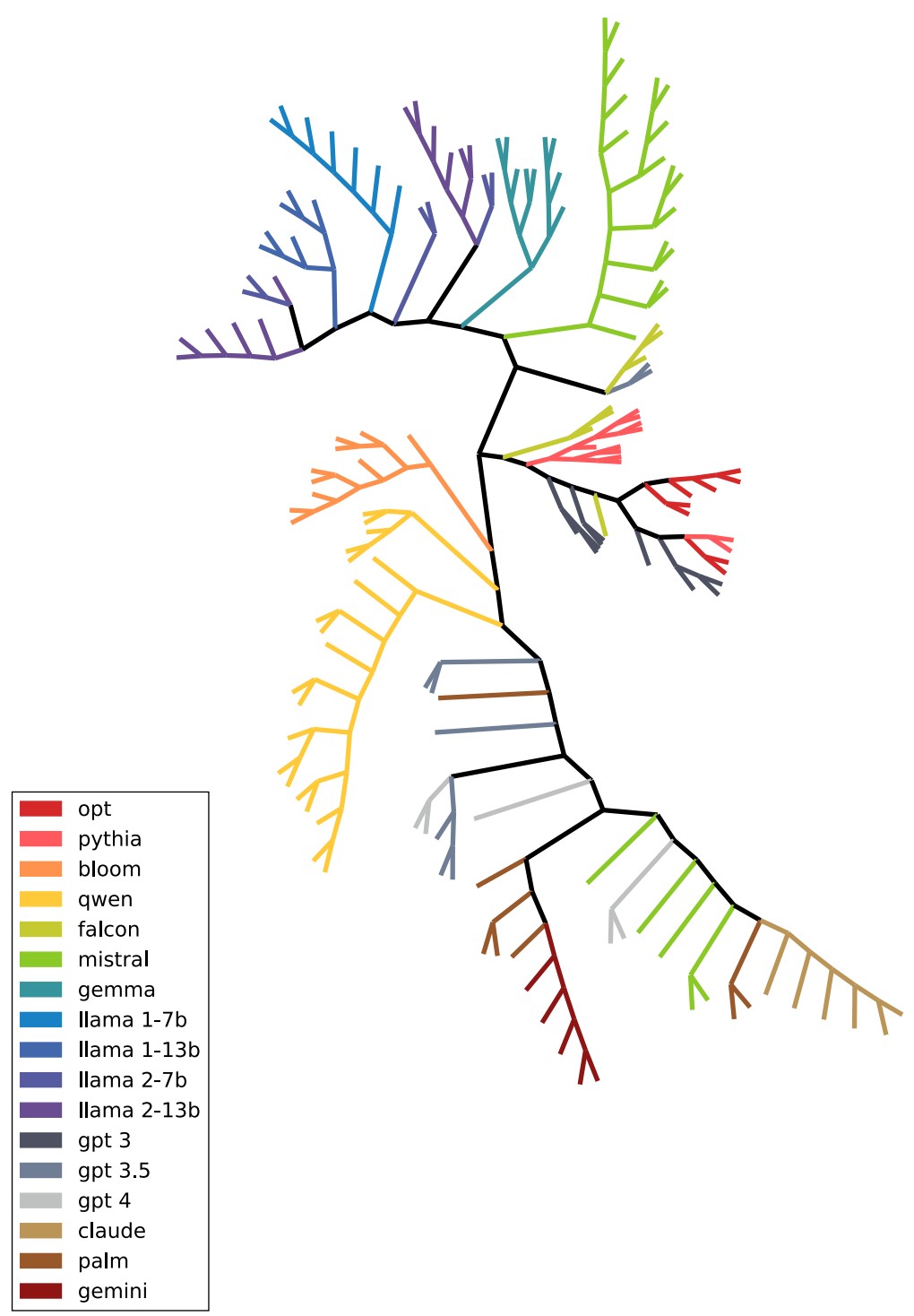

Figure 18: **Dendrogram with all models (completion and chat) at once on the math set of genes**. All models included are shown in Appendix B.

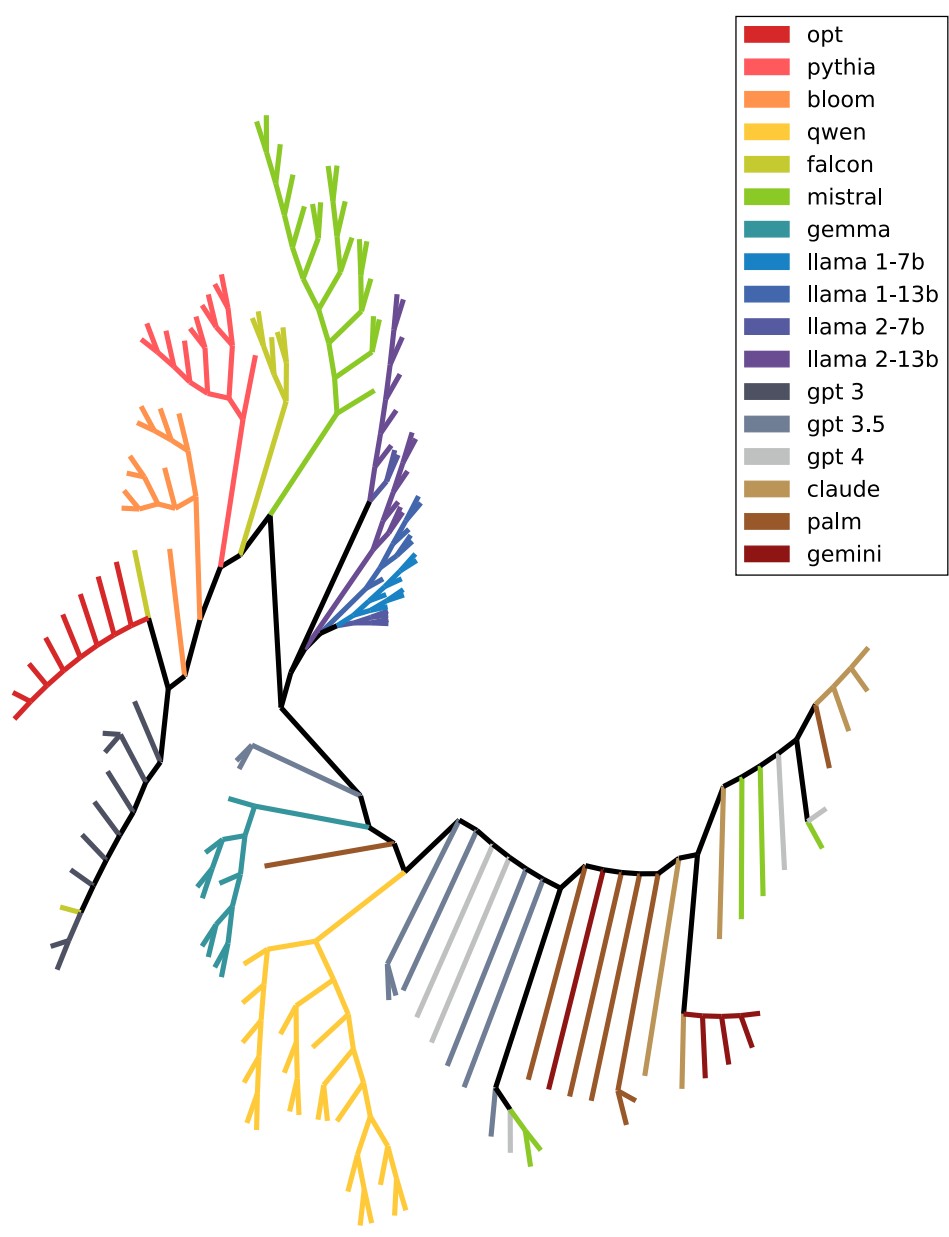

Figure 19: **Dendrogram with all models (completion and chat) at once on the code set of genes**. All models included are shown in Appendix B.

## H PHYLOLM AND MODEL QUANTIZATION

Quantization of models has become very common the in the LLM field as it makes it possible to run a very large model on a rather modest hardware. However this procedure relies on simplifying the inner computations of the model raising many concerns about whether the quantized version is as reliable as the original version. As such, we investigated the Qwen family of models with quantization. They provide a GPTQ 4 bit, GPTQ 8bit and AQW 4 bit version of all their chat models from the family 1.5 (Bai et al., 2023; Frantar et al., 2023; Lin et al., 2024). We computed the similarity matrix for all the models in the family (except the 32B version as all quantized models weren't online at the time of the study) and compared the similarity with respect to their 3 quantized versions. The similarity matrix between these models is shown in Figure 21 for the math genes and 22 for the code genes. The authors did not communicate on the hyperparameters used to quantize these models aside from the number of bits

Interestingly, in the Qwen 1.5 family release, the quantized version the closest to the original model is GPTQ 8bit followed closely by the 4bit and finally by the AWQ a lot farther. This observation stands for all the models in the family. However, intriguingly, the larger the model the less quantization seem to affect the distance to the original model (see Fig 20 - t(5)= 21.8 (p<0.001)).

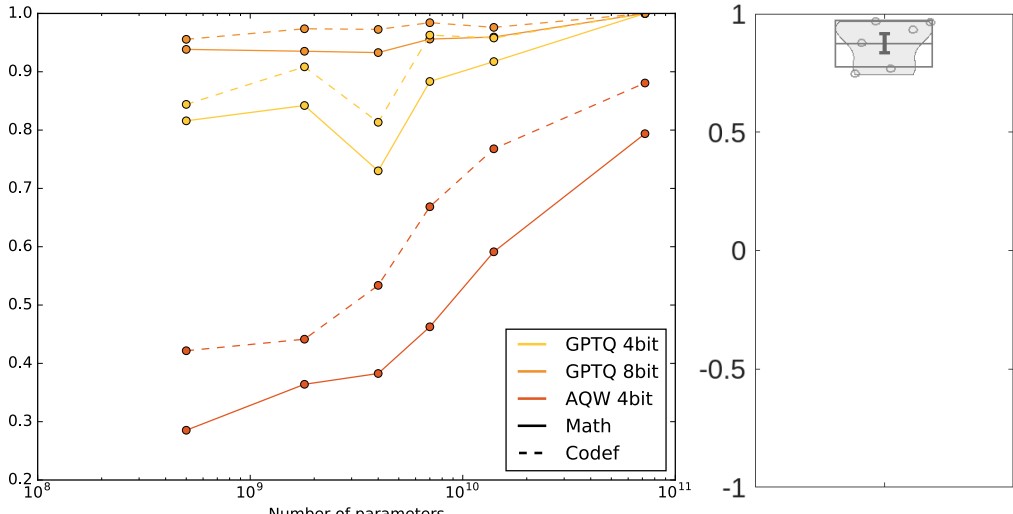

Figure 20: **Evolution of the similarity with the original model for all quantized Qwen1.5 Chat models.** Each curve represent the evolution of similarity with the original model for each quantization method depending on the size of the model. All models included come from the Qwen1.5 huggingface repository including the quantized versions (). Qwen1.5 32B Chat models are not included here as not all quantized versions were accessible at the time of the study.

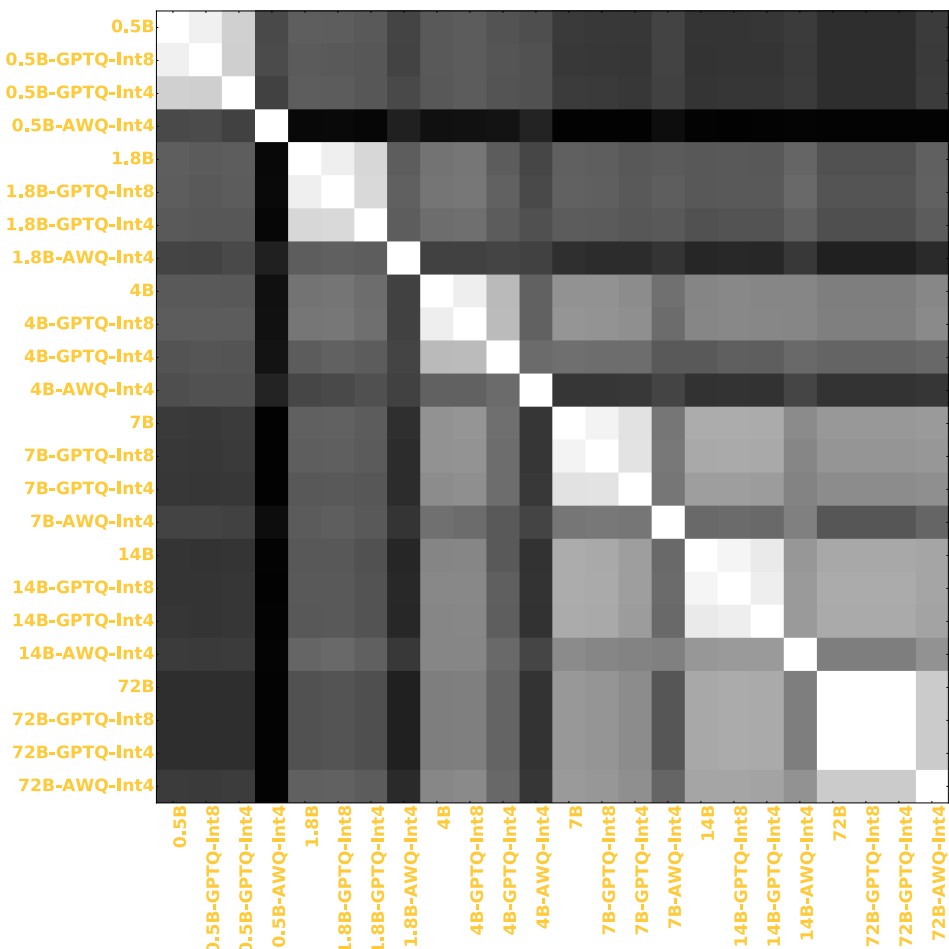

Figure 21: **Similarity Matrix between Qwen 1.5 Chat models and several quantized versions on the math set of genes..** For the sake of simplicity only the size of the models have been included in the legend for the original Qwen 1.5 Chat models and the quantization method as well as the size of the quantization for the quantized versions. All models included come from the Qwen1.5 huggingface repository including the quantized versions (). Qwen1.5 32B Chat models are not included here as not all quantized versions were accessible at the time of the study.

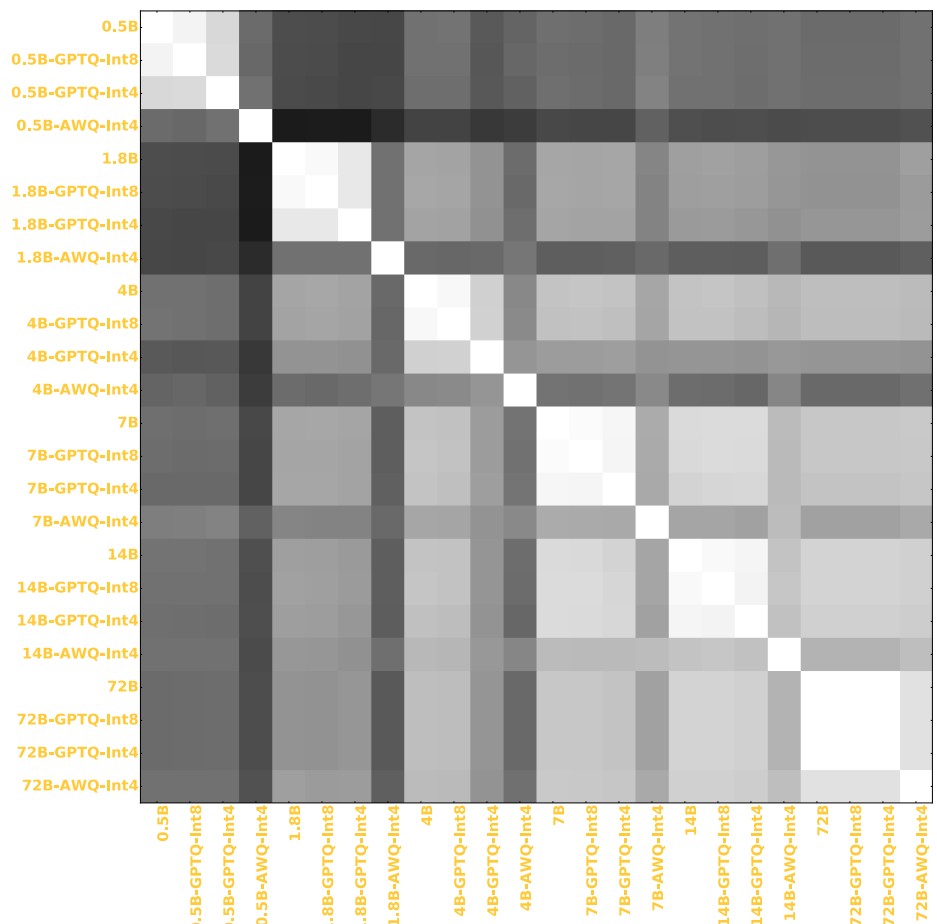

Figure 22: **Similarity Matrix between Qwen 1.5 Chat models and several quantized versions on the code set of genes.**. For the sake of simplicity only the size of the models have been included in the legend for the original Qwen 1.5 Chat models and the quantization method as well as the size of the quantization for the quantized versions. All models included come from the Qwen1.5 huggingface repository including the quantized versions (). Qwen1.5 32B Chat models are not included here as not all quantized versions were accessible at the time of the study.

# I BIG DENDROGRAMS

Dendrograms contained in the main text do not include model names for the sake of clarity as most of them would overlap. For the sake of clarity, we include here large vectorial figures of these corresponding dendrograms with the names so that readers can zoom in to see better where are the individual models in the dendrograms. Figure 23 shows the math based dendrogram while Figure 24 shows the code based dendrogram.

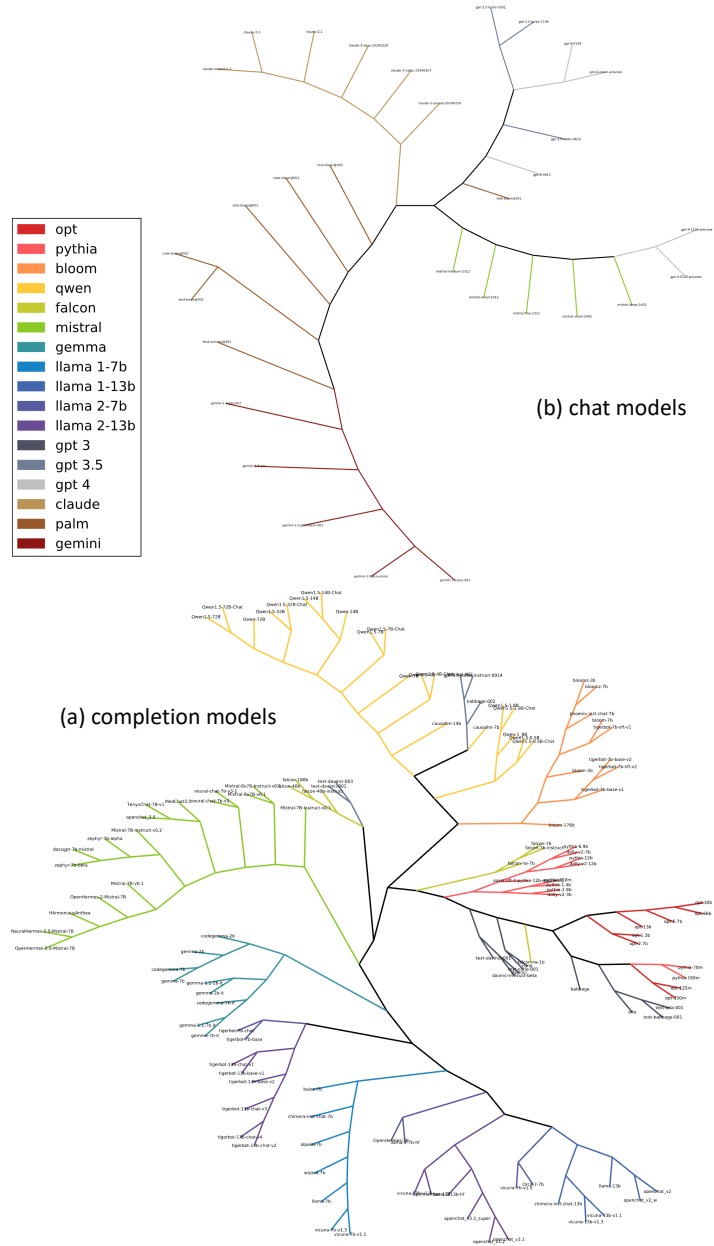

Figure 23: **Inferred phylogenetic tree of LLMs on the 'math' set of genes with the names**. (a) completion models inlcude all open source models included in our study and the 14 openai completion models (b) chat models include additional proprietary models. Completion and chat models were separated because they are not comparable due to additional prompting from the API. Llama models have been split by version of the pretrained model and the number of parameters. Names are very small but this is a vectorial figure : readers are encouraged to zoom in when reading from a computer.

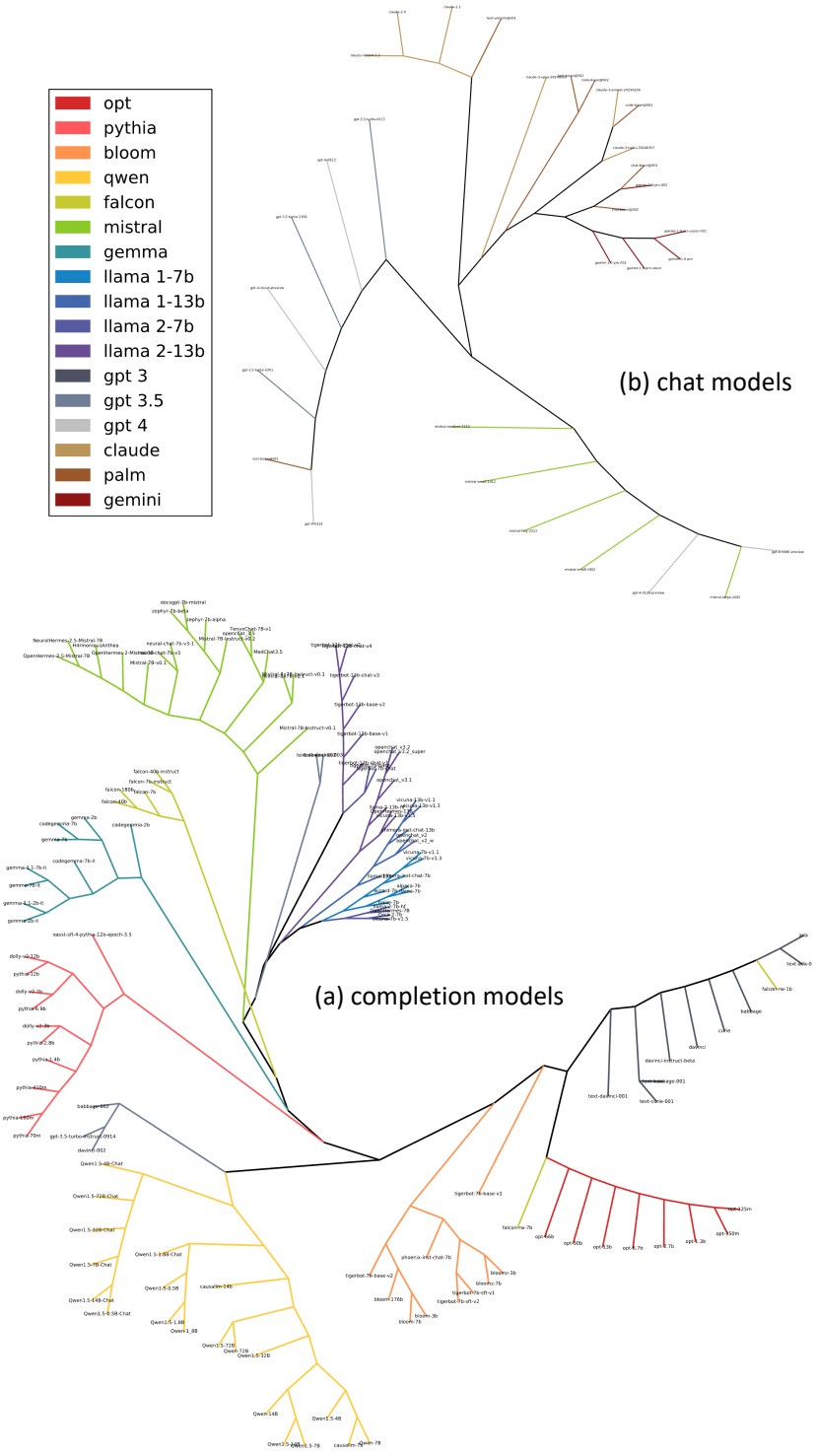

Figure 24: **Inferred phylogenetic tree of LLMs on the 'math' set of genes with the names**. (a) completion models inlcude all open source models included in our study and the 14 openai completion models (b) chat models include additional proprietary models. Completion and chat models were separated because they are not comparable due to additional prompting from the API. Llama models have been split by version of the pretrained model and the number of parameters. Names are very small but this is a vectorial figure : readers are encouraged to zoom in when reading from a computer.

## J   OTHER SETS OF GENES

The main text presents an extensive evaluation of 2 sets of genes : one based on reasoning and the other on code. Choosing a set of gene is asking a question to the group of LLMs studied. In the main text we wanted to investigate the phylogeny of language models so we tested them on skills that have evolved a lot recently. Here we include results on other sets of genes by asking other questions like language spoken, chat interaction familiarity and finally another gene set about more general knowledge.

### J.1   CHINESE POETRY GENES

We wanted to test the language spoken by various LLMs so we used a set of genes extracted from huggingface - erhwenkuo/poetry-chinese-zhtw using the same pipeline as presented in the main text in section 2.2 but cutting around the 5th character instead of 20-100 as chinese characters are much more informative than latin characters. The similarity matrix show a very little contrast between models from the same family for except Qwen and Bloom, known to have been trained on Chinese. Interestingly, Mistral also shows little contrast as well as tigerbot models finetuned from llama (that do not seem to know Chinese) appear to be quite close to bloom models and its own tigerbot finetunings (see Figure 25). When visualizing the matrix with dendrogram the major axis in the dendrogram looks very linked to the ability to write Chinese with the right models in the Figure 26 being the ones that show the most contrast in the matrix. The left models show very little contrast and are not known to speak chinese.

### J.2   CHAT-STYLE GENES

We also tested genes from the reasoning gene set but with additional chat prompting to see if we can spot models that have been finetuned with a chat format. New genes have directly been obtained from the original math genes discussed in the main text by adding additional prompting around them :

> User:[Reasoning gene]\n Assistant:

Results in Figure 27 show much variance within the families which appears to be a bimodal distribution. To better visualize it we plotted the dendrograms with colors of families but also from whether they are explicitly trained for chat-based interactions (see Fig 28). When looking at families we notice that are much more split in the dendrogram compared to previous results without the chat prompting (compare with Figure 4 - completion models). When looking at whether the models have been exposed to chat-based interactions during training we notice a more clear separation in the dendrogram. Indeed the bottom part of the plot only includes models not explicitly trained on chat based interactions while the top part is essentially composed of models trained for such interactions.

### J.3   WIKIPEDIA

Lastly we wanted to see if it was still possible to reconstruct the phylogeny of LLMs not using classical finetuning topics. We used a the wikimedia/wikipedia dataset from huggingface and reused the same pipeline as presented in the main text in section2.2 to extract 'genes' from the dataset. The similarity matrix in Figure 30 shows patterns very similar to what we observed on math or code in the main text. Similarly, the dendrogram (see Figure 29) splits extremely well the different families showing that even more sets of genes can be used to study the evolution of LLMs.

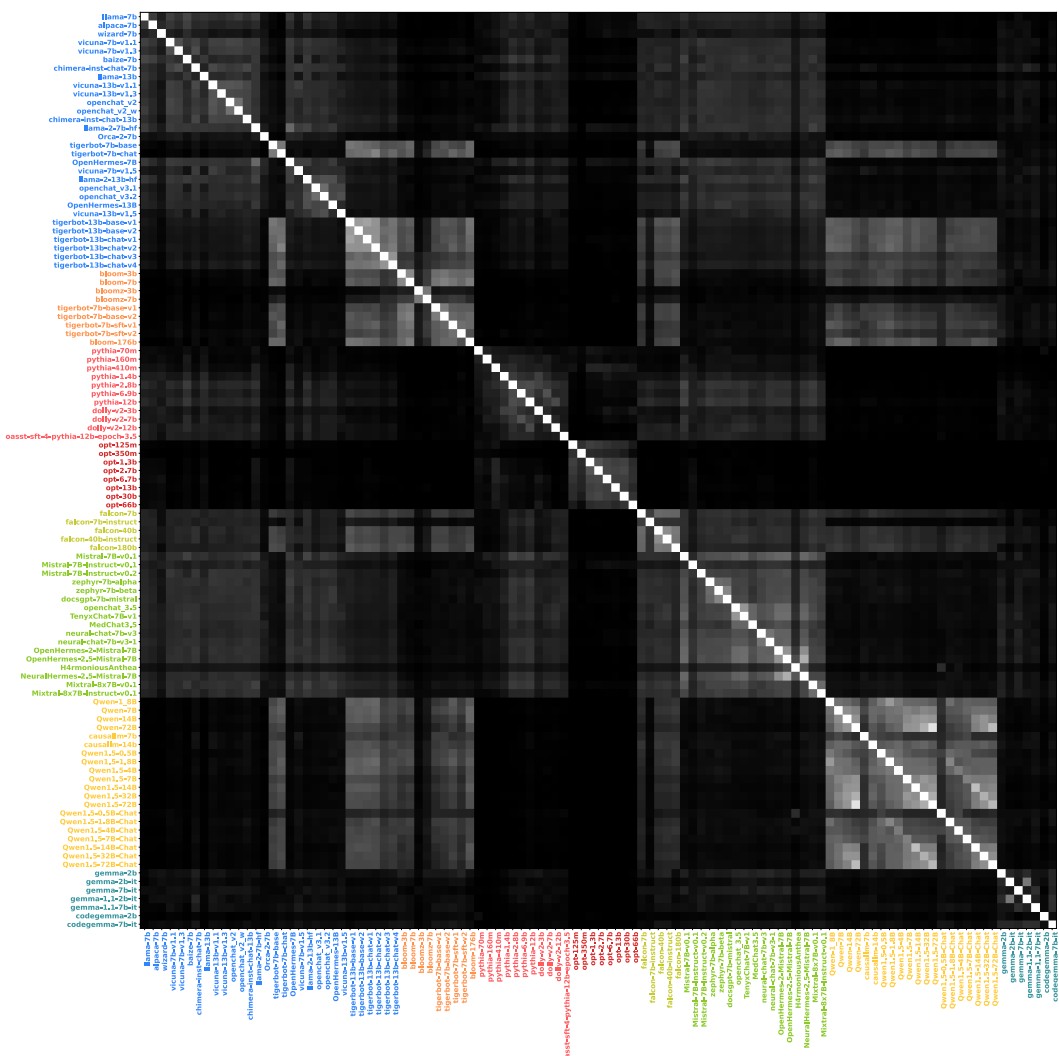

Figure 25: **Similarity matrix of some completion LLMs on the chinese poetry set of genes.**

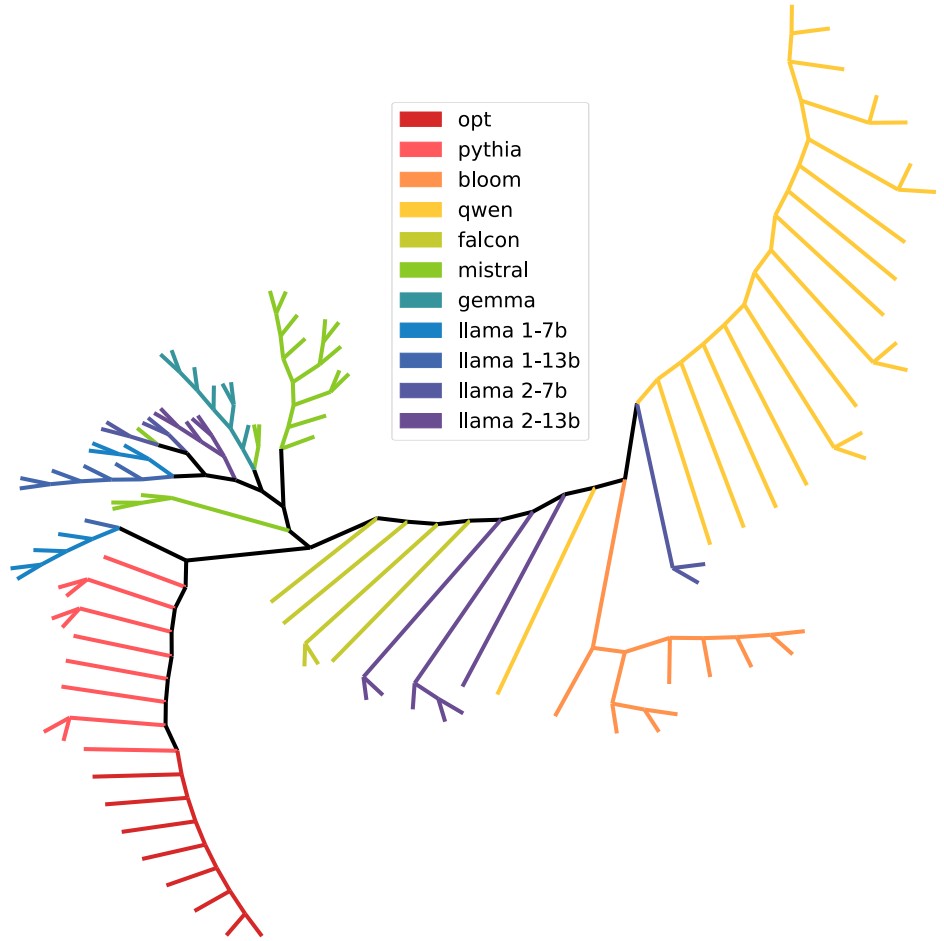

Figure 26: **Inferred phylogenetic tree of LLMs on the chinese poetry set of genes.** All models included are completion models. Llama models have been split by version of the pretrained model and the number of parameters.

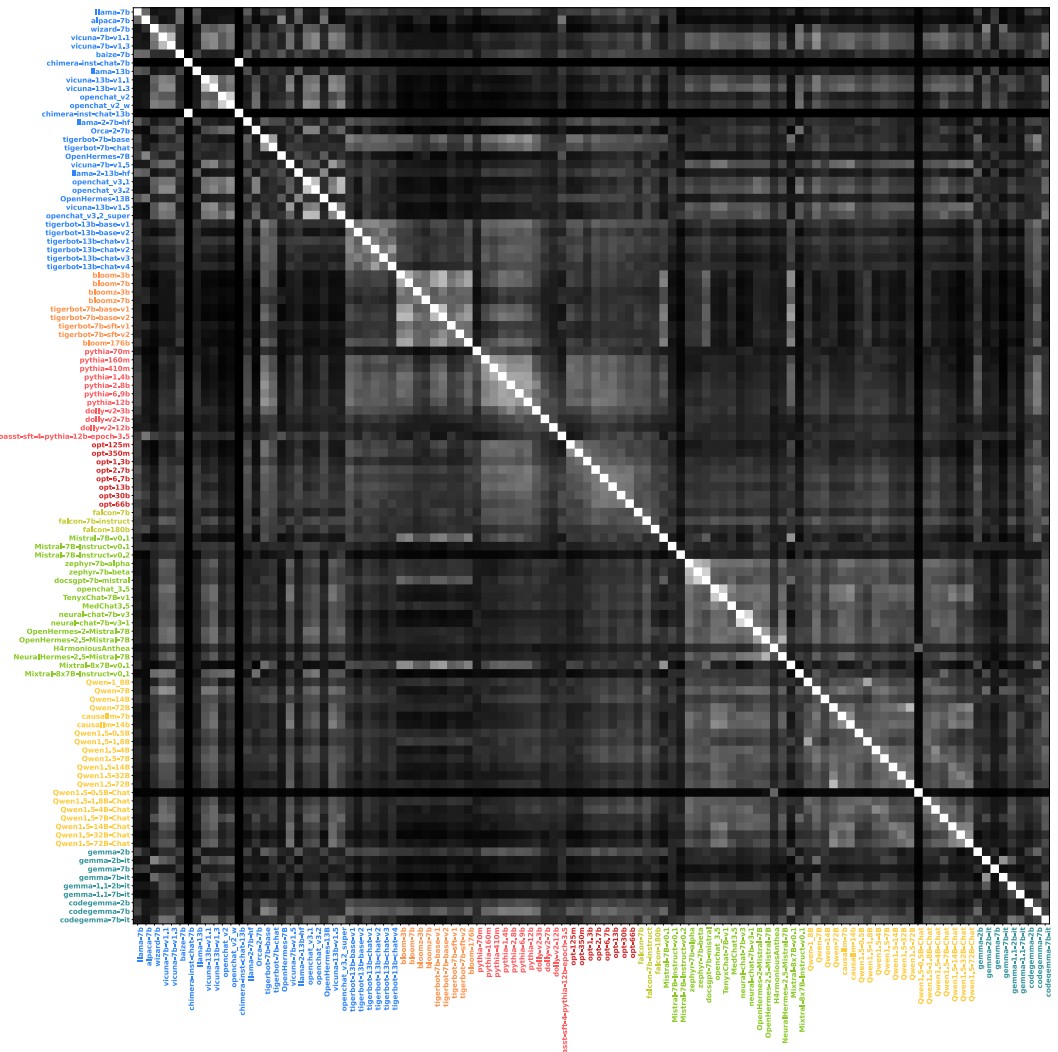

Figure 27: **Similarity matrix of some completion LLMs on the chat version of the math set of genes.**

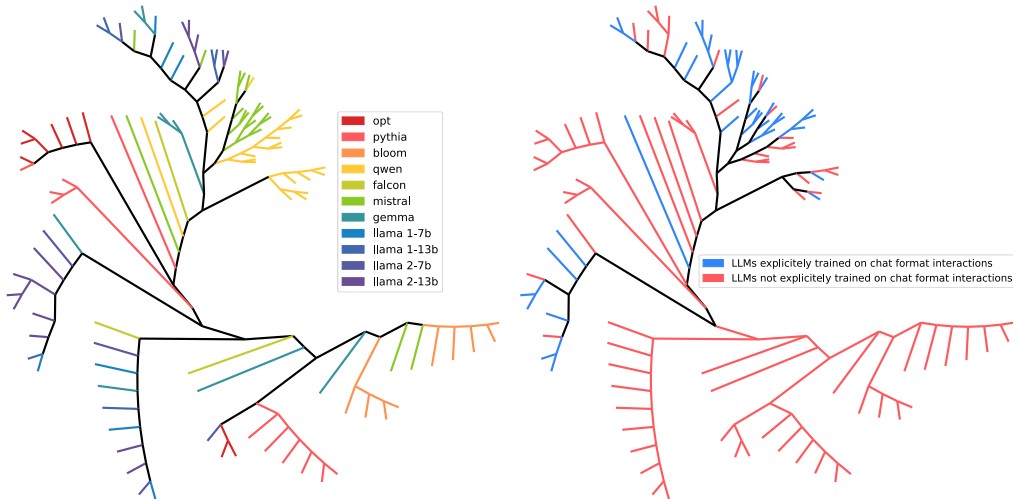

Figure 28: **Inferred phylogenetic tree of LLMs on the chat version of the math set of genes with coloring from families and chat interaction training.** Left shows the dendrogram colored with family colors and the right plot shows the same dendrogram but with colors indicating whether they have been explicitly trained to be familiar with chat interactions. All models included are completion models. Llama models have been split by version of the pretrained model and the number of parameters.

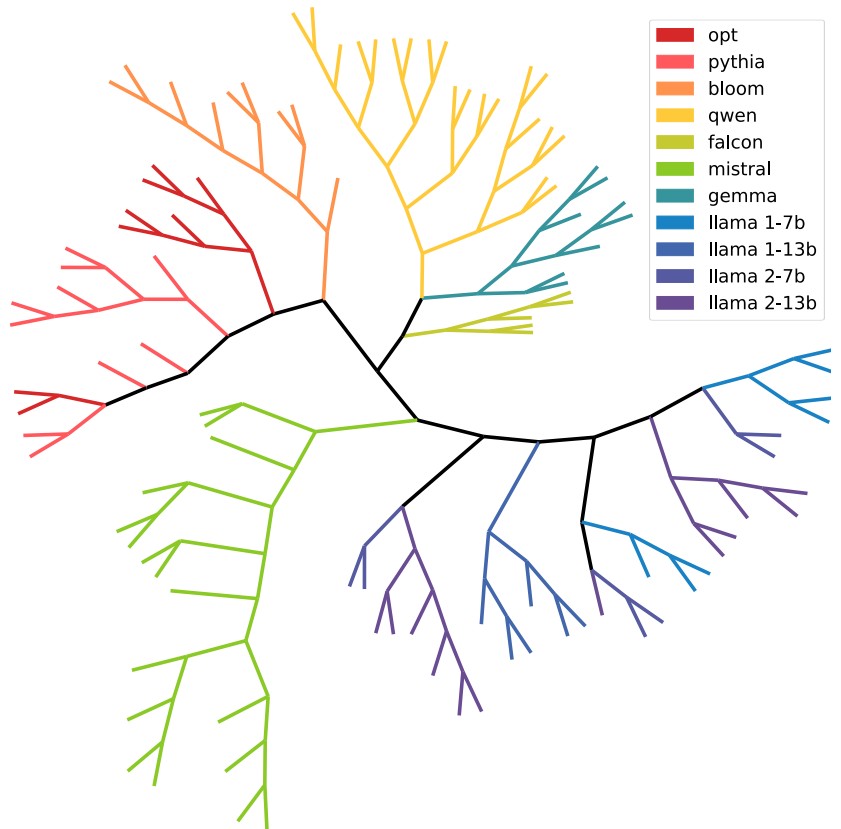

Figure 29: **Inferred phylogenetic tree of LLMs on the chinese poetry set of genes.** All models included are completion models. Llama models have been split by version of the pretrained model and the number of parameters.

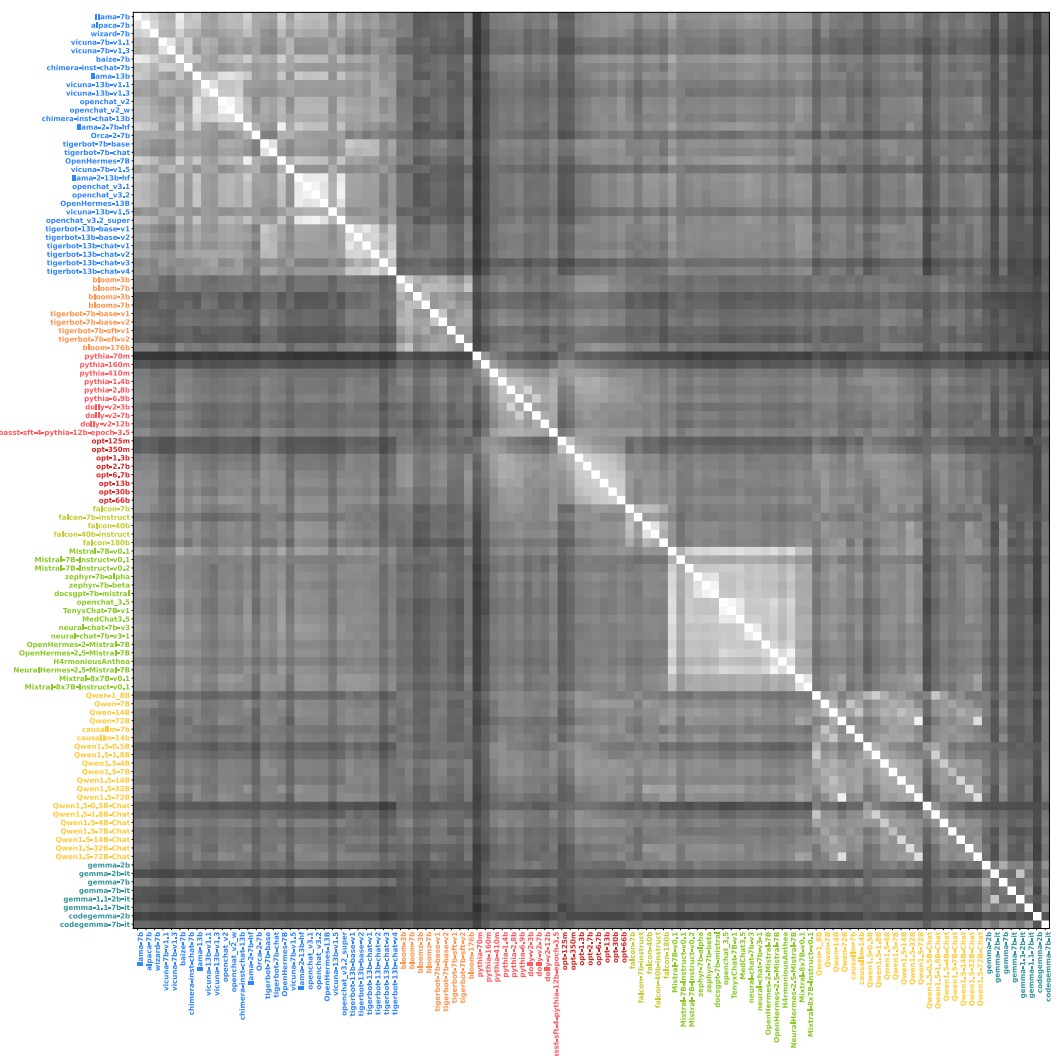

Figure 30: **Similarity matrix of some completion LLMs on the chinese poetry set of genes.**

## K ADDITIONAL ANALYSES OF GENE SETS' HYPERPARAMETERS

### K.1 GENE LENGTH

In section 2.2 we presented a pipeline to sample genes and explained that we truncated text coming from test benchmarks somewhere between the 20th and the 100th character. This choice is arbitrary and we provide here more precisions about the impact of the length of genes. We took a subset of LLM families included in the study and computed a gene set using the same pipeline as presented in the main text but with either only 5 characters, 200 characters or 1000 characters. We plot here the similarity matrices and phylogenetic trees (Figure 31) made from these genes. For short genes the matrix is a little darker than for longer genes. However, the similarity matrix doesn't seem to change very much with gene length from 20 to 100 characters long (main text configuration) and all 4 sets lead to good family clustering in the dendrograms. We conclude that the length of the gene doesn't impact very much the reconstruction and thus it is probably better to keep rather short genes in order to avoid unnecessary costs when running the algorithm.

### K.2 COMPLETION LENGTH

We also tested the impact of the number of characters to generate after each gene. Generating only 1 character will increase the likelihood for 2 models to generate the same thing (even if they meant something different but starting with the same character) thus the similarity matrix will be a lot brighter in general as models appear more similar. On the other hand, letting the LLM generate 20 characters is a lot more likely to produce generations that do not match leading to a similarity matrix a lot darker.

A similarity matrix with low contrast (too bright or too dark) makes all the differences between models in a very low numeric range. Knowing these matrices are a little noisy (see Figure 2b in section 2.2) it is better to have a contrast as high as possible in order to better visualize similarities between models.

Thus we computed the RMS contrast of the matrix (standard deviation of the similarity matrix including all the parameters but the diagonal) and plotted its value from 1 character long completion to 19 characters long completions. The results in Figure 32 show an optimal contrast for a completion length somewhere between 2, 3 and 4 characters.

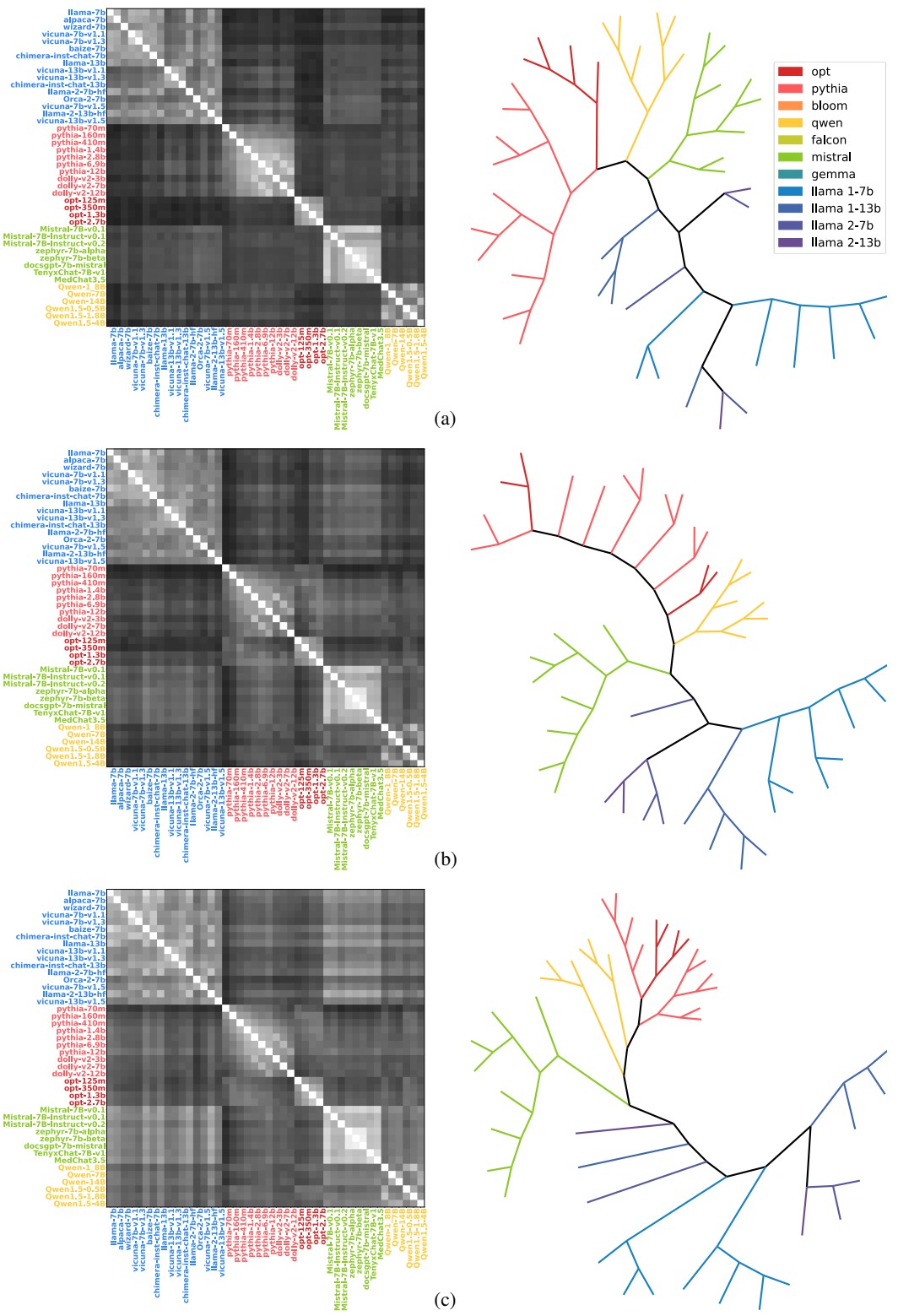

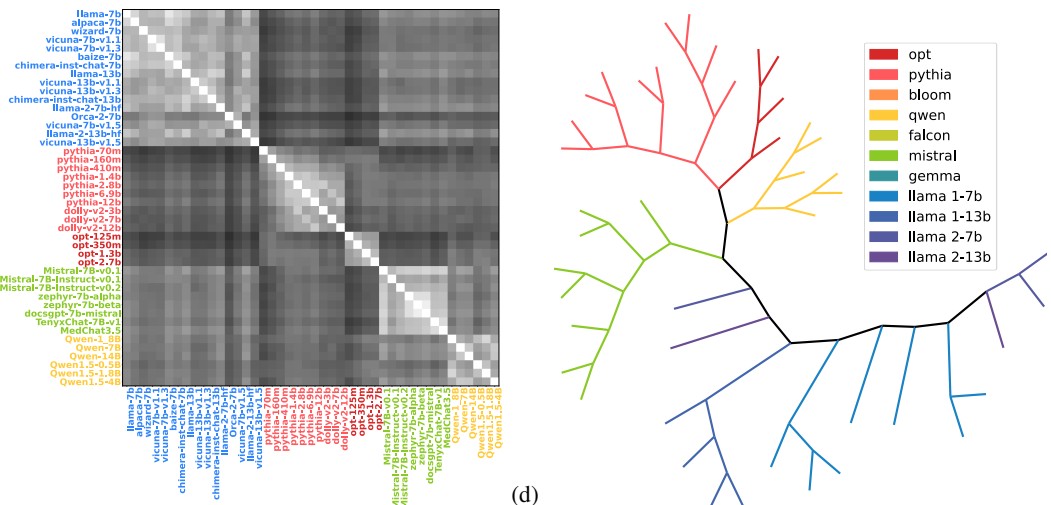

Figure 31: **Impact of the number of characters in the gene compared in each generation on similarity matrices** Similarity matrices only include Llama, Pythia, OPT, Mistral and Qwen families. (a) is the similarity matrix for only 5 character long genes (b) is for 20 to 100 characters long genes (like in the main text), (c) is for 200 characters long genes and finally (d) shows the similarity matrix for 1000 characters long genes.

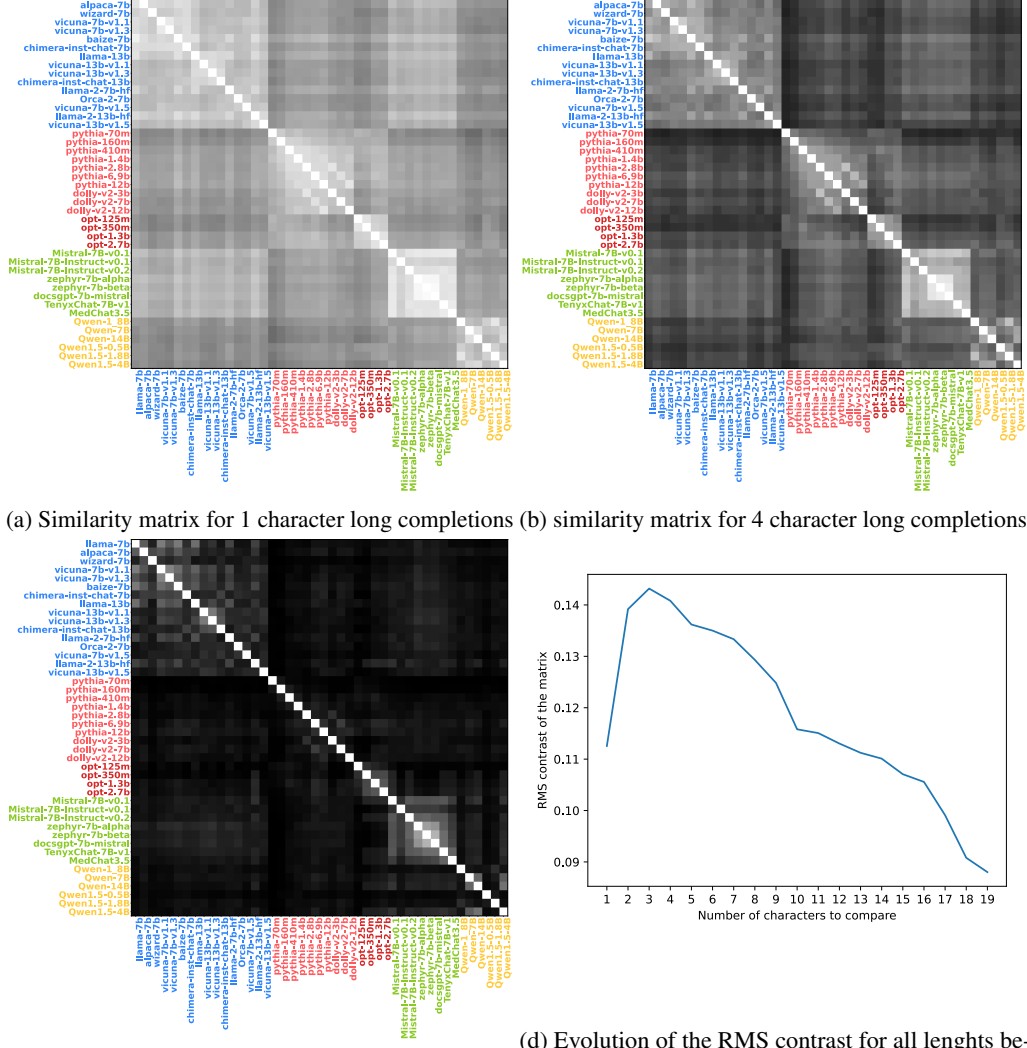

(a) Similarity matrix for 1 character long completions (b) similarity matrix for 4 character long completions

(c) similarity matrix for 19 character long completions

(d) Evolution of the RMS contrast for all lenghts between 1 and 19 characters.

Figure 32: **Impact of the number of characters compared in each generation on similarity matrices** Similarity matrices only include Llama, Pythia, OPT, Mistral and Qwen families.

