# OpenReview forum: "PhyloLM: Inferring the Phylogeny of Large Language Models and Predicting their Performances in Benchmarks"
_ICLR.cc/2025/Conference — ICLR 2025 Poster_

### Official Review · Reviewer_Lm8z · 2024-11-03

**Soundness:** 1
**Presentation:** 1
**Contribution:** 2
**Rating:** 3
**Confidence:** 4

**Summary:**

The paper utilizes a metric inspired by phylogenetic to measure similarities between LLMs, outputs dendrograms between them, and predicts performance. By putting LLMs into the evolution framework, the authors propsoes a new way to evaluate model performance in a cost effective way. For models where the training details are scarce, this new framework also helps in gaining insights into their training process.

**Strengths:**

1. Originality: the paper is quite original in its making connection between phylogenetics and LLMs. This might open a novel avenue for studying LLM lineage and evolution
2. Significance: the paper has the potential to have high impact as it proposes a new framework of understanding LLMs. Moreover, the use of the framework in evaluating LLMs with low resources is quite valuable considering the emergence of LLMs and the scarcity of compute resources in some research communities.

**Weaknesses:**

1. While the connection between phylogenetics and LLMs is novel, a not of the details are not worked out thoroughly in the paper. For example, the paper makes a connection between tokens used by LLMs and alleles in genetics. However, when comparing different LLMs, which usually use different tokenization schemes, the analogy breaks down. The authros tries to mitigate that using the 4-character generated by the model. Then an obvious drawback is that for models that do not exactly output a token of 4-character long, what the authors are measureing are no longer p(a|g). Why is 4 used here also seems arbitrary, and could be explained or motivated better. Have the authors done any experiments on changing this number and see if the results are different?
2. Again on the analogy between phylogenetics and LLMs, these kinds of analogies could serve as an inspiration and starting point, but relying entirely too much on it could be questionable. For example, a key component of the paper is the definition of the similarity metric between LLMs, which is adapted from a metric in phylogenetics. But why is this metric suited for the analysis of LLM's? Since what's being compared is P(a|g), i.e., distributions, why not use some more standard metrics for distributions like divergence?
3. The writing of the paper could be improved significantly. The clarity of the paper could benefit from some proof-reading. There are many places where the use of punctuations is wrong, e.g., ’gene’,

**Questions:**

The main questions are listed in the "weakness" section.

---

> ### Author Response · Authors · 2024-11-21
> **Rebuttal 1**
>
> We thank the reviewer for their interesting and very relevant feedback.  We are encouraged by the fact that the reviewer found the work “original”, that it “has the potential to have a high impact” and that it might “open a novel avenue for studying LLM lineage and evolution”. Here are our answers to their comments :
>
> **Probability of the token VS 4 first characters**
>
> We thank the reviewer for pointing out this notion of tokenizer that we didn’t expand on in the narrative of the paper. We explained the algorithm using P(a|g) as if all LLMs share the same tokenizer as it is a very intuitive way to present the analogy using classical notations from the LLM field that most people are familiar with. Later, we explain that, in practice, it is more complex due to the various tokenizers used by LLMs. In practice, while tokenizers will cut a text in different manners the text being generated remains the same. In this work, we are not interested in tokens but in the text being generated by LLMs that is independent from the way it is tokenized, and thus from the tokenizer used by the LLM. To do so we let the LLM generate several tokens until we get a 4 characters or longer generation that we truncate to the first 4 characters that we will compare between models. Put differently, we are re-tokenizing the texts generated by LLMs with a tokenizer with a vocabulary including all sequences of 4 characters and we are comparing completions P(t|c) using this independant tokenizer. We tried to make the paper easily understandable by a wide readership, therefore we introduced the analogy using intuitive notations and didn’t expand much on the technicalities. We updated section 2.4 to explain in more diverse terms (simple and more technical) how we implemented the re-tokenization. We are not really sure why the reviewer may think this is an issue in the method especially in the light of the good results that we have.
>
> **Why 4 characters ?**
>
> About the choice of using the 4 first characters, this is an excellent question, and we thank the reviewer for giving us the opportunity to justify our choice. We tested different values of the number of characters to generate and computed the standard deviation of the non diagonal coefficients of the similarity matrix as a measure of contrast (known as RMS contrast) for each value. Only comparing the 1st character will result in a lot of matches and a very bright similarity matrix with low contrast which is not very informative. Comparing the first 20 characters will result in much fewer matches and a darker low contrast similarity matrix. In short, the better the contrast the more precise the similarity matrix. The optimal value seems to be around 3 or 4 which justifies the choice in the main text (see new Appendix K).

---

> ### Author Response · Authors · 2024-11-21
> **Rebuttal 2**
>
> **Scientific background of using phylogenetic methods in cultural artifacts evolution**
>
> Using phylogeny algorithms to study the evolution of cultural artifacts is not a new idea. Dawkins proposes in his book to use these methods to study the evolution of human culture [7] and many other researchers have followed him in this regard. To cite a few of them, D’Huy used such algorithms to study the evolution of myths [1,4], Atkinson applied them to language evolution [2], Gray and Tehrani to the study of human culture evolution [3,5] and Le Bomin has shown their efficiency in studying the evolution of music and rhythms [6]. All these researchers have discovered valuable new insights applying these algorithms to their respective field that have, for some of them, been validated afterwards by archaeology. In these studies the important point for the analogy is to find objects (analogy to genes) that mutate with time and evolution in different forms (analogy to alleles). D’Huy for example uses features in stories such as the character gender, their equipment, family composition, … Similarly, in music, Le Bomin compares music using 3 components : repertoire (circumstances where they are played), performativ (instruments used, vocal techniques) and intrinsic (metrics, rhythms, melody, …). We apply this method to LLMs and select contexts as genes and tokens to be generated in these contexts as alleles as the probability to find an allele sampled after a given context is what changes with training, namely time and evolution.
>
> **Nei score VS KL divergence**
>
> We could compute divergences between probability distributions but it required having access to the full distribution which is not always the case especially for models accessed through an API. We wanted to make a method that can work for both open access models and API models so we cannot compute classical KL divergence in such a scenario. Therefore we found inspiration from biology where methods have been developed for decades comparing P(a|g) with not complete knowledge about the distribution. We do not claim that it is the optimal way to do it and the metric could be improved in future work. For a first paper about genetics of LLMs it makes sense to use tools that have been developed in genetics, and also applied to studying other cultural artifacts, for years.
>
> **Usage of punctuation for ‘genes’**
>
> While contexts are not genes in a biological sense, they play a comparable role in this work. To emphasize this analogy while clearly distinguishing contexts from actual genes, we intentionally referred to them as ‘genes’—using quotation marks to signal that the term is being used metaphorically rather than literally.
>
> We thank the reviewer for their valuable contribution in improving the paper. We hope our comments answer the reviewers questions and are grateful for the opportunity to refine our work and hope these improvements can lead to an even better evaluation of our work. We welcome any additional questions or feedback the receiver may have.
>
> **References**
>
> [1] d’Huy, Julien. (2013). Polyphemus (Aa. Th. 1137). A phylogenetic reconstruction of a prehistoric tale.Nouvelle Mythologie Comparée
>
> [2]Atkinson QD, Meade A, Venditti C, Greenhill SJ, Pagel M. (2008) Languages evolve in punctuational bursts. Science.
>
> [3]Gray RD, Bryant D, Greenhill SJ. (2010)  On the shape and fabric of human history. Philos Trans R Soc Lond B Biol Sci.
>
> [4]Tehrani, Jamie & d'Huy, Julien. (2017).  Phylogenetics Meets Folklore: Bioinformatics Approaches to the Study of International Folktales
>
> [5]Tehrani, Jamie & Collard, Mark. (2009). On the relationship between interindividual cultural transmission and population-level cultural diversity: a case study of weaving in Iranian tribal populations. Evolution & Human Behavior
>
> [6]Le Bomin S, Lecointre G, Heyer E (2016) The Evolution of Musical Diversity: The Key Role of Vertical Transmission. PLOS ONE
>
> [7] Dawkins, R. (2006). The Selfish Gene. Oxford University Press.

---

> > ### Author Response · Authors · 2024-11-26
> >
> > We sincerely thank the reviewer once again for their thorough and thoughtful review of our manuscript. We did our best to address the reviewer’s questions and have revised the paper in line with the suggestions provided by all reviewers.
> >
> > If there are any follow-up questions or additional points requiring clarification, please do not hesitate to let us know. We greatly value the reviewer’s insights and are fully prepared to provide any further information that may be helpful. Additionally, we kindly request the reviewer to consider updating their evaluation if they feel our responses have clarified and strengthened the contribution of our work.

---

> > > ### Author Response · Authors · 2024-12-02
> > >
> > > We kindly remind the reviewer that today is the final day of the discussion. We invite them to ask any remaining question and to update their evaluation if our responses have made the contribution clearer. We thank the reviewer again for their time and constructive feedback.

---

### Official Review · Reviewer_mtg8 · 2024-11-03

**Soundness:** 3
**Presentation:** 4
**Contribution:** 4
**Rating:** 10
**Confidence:** 4

**Summary:**

The authors transfer a well-known paradigm of phylogenetic similarity from computational biology to the study of LLMs. The authors leverage the responses of LLMs to prompts contained in common benchmarks as "genes' to evaluate the difference of responses to them as "alleles." They then apply Nei's standard genetic distance to the distance between tokens in the responses with additional character agglutination normalization on the first tokens for models with different tokenizers. The authors use their "genetic" LLM distance to build phylogenetic trees and predict the model performance on standardized benchmarks.

**Strengths:**

This work is outstanding.
- The clarity of writing is top-notch, and this paper should be readable not only by the ML and genetics community, but by the general public
- The problem the authors are trying to address - choice of most related LLMs and inference of performance of a previously untested LLM - is timely and important, as the proliferation of LLMs and tests means that for domains of interest, it is currently impossible to select the most appropriate LLM for a given application
- Authors have a deep understanding of bases of computational genetics and phylogenetics (reference genes, ancestral species inclusion) to accurately map the concepts from that field to LLMs
- Authors also have an excellent grasp of the inner workings of LLMs (e.g., the tokenization step), allowing them to
- Authors are able to choose appropriate statistical tools to perform tasks of interest rather than the most common ones (e.g., ICA instead of common but inappropriate PCA on L268).
- Authors perform extensive benchmarking, offering insights that are in themselves interesting and shed light on some of the long-standing questions in the LLM community, e.g., the performance of quantized vs. non-quantized models (L496-498) or synthetic dataset re-use in non-opensource models (Fig 4.)
- Supplementary data is extensive (full distance matrices), and all the necessary code to re-run the experiments is provided

**Weaknesses:**

- Fig 5A is overloaded and confused, and 5B is non-informative. Please consider splitting out samples to make the quality of correlation clear on a first view
- There is no discussion of neutral drift vs. selective pressure, which is particularly relevant, given that the former in genetics acts as an "evolutionary clock," generally used to estimate divergence in close populations. Authors, however, selected prompts from common benchmarks, for which models are frequently optimized to perform well, and that can be seen as genes highly selected for.
- The usage of exact token matching is a major weakness. LLMs are known for their semantic variability and ability to provide answers identical in meaning but with drastically different wording, which the author's method would consider as different. While the author's approach is arguably more relevant to identifying the similarity of training data for LLMs, overall performance is more likely to be predicted by a semantic similarity metric

**Questions:**

- L242-256, Is there a reason you did not use a 3rd-party tokenizer to calculate the distance between the models with different tokenizers? It would seem to improve the methodological consistency with the rest of the work.
- L046-L047, Is there a reason you did not cite Dawkin's "Selfish Gene", that is generally regarded as the source of the "meme" idea and hence the idea of applying population genetics to cultural artifacts?

---

> ### Author Response · Authors · 2024-11-21
> **Rebuttal Part 1**
>
> We are sincerely grateful to the reviewer for their enthusiasm and very thoughtful comments. We were very encouraged by the fact that they found the work “outstanding”, of “top notch” clarity and readable “by the general public” as we wanted to make it easily accessible. The comments of the reviewer were very relevant for our work. Here is how we addressed them :
>
> **Improvement of the clarity of Figure 5**
>
> We agree that Figure 5A feels overloaded, the main aim of this figure is to show that overall the prediction and true score are correlated. All the individual plots for each benchmark are in Appendix F (as explained in the paragraph right below). We switched the actual Figure 5 in appendix and put instead a methodology figure in 5A showing that we fitted a regression on all but one family to predict the score of the remaining family and in Figure 5B all the predictions for all families on ARC. This should make the process much more clear and we kept the link to the appendix so that the reader can see each of the 6 benchmarks individually as well as the previous Figure 5.
>
> **Selective pressure VS neutral drift in the light of the analogy**
>
> We haven’t discussed much neutral drift VS selective pressure because we wanted the narrative of the paper to be simple and easy to follow even for non specialists. However we took it into account when designing the sets of genes and are glad to discuss our choices.
> A key principle in genetics is selecting genes appropriate to the study's time scale. Some genes evolve too slowly to provide sufficient variance for reconstructing a phylogenetic tree, while others evolve so rapidly that the resulting variance is too high within the same branch to make it possible to trace species evolution accurately.
>
> Applying this principle to our work with PhyloLM, we decided not to use texts from the model’s training sets as they might be too much in the selective pressure direction, leading to a very large variance in outputs between models from the same family. To avoid this, we chose texts coming from datasets that are unlikely to be used as training data like benchmark test sets.
>
> At the same time, we were worried that selecting texts too distant from the training direction might result in a very low variance between models from the same family. Therefore, we aimed to strike a balance, choosing texts tangential to the training direction - outside the training sets but still close enough to exhibit moderate variance like texts about reasoning and coding that are unlikely to be found in the training sets.
>
> Nonetheless the insights of the reviewer were right, we tested a new gene set based on text extracted from wikipedia that is more orthogonal to the selective pressure and found very good results, maybe even better than with math and coding gene sets. We encourage the reviewer to give a look at Appendix J.3 and Figure 29. This could encourage future work in this direction to try to better understand what is the optimal variance to observe LLM evolution.
>
> **Comments on exact token matching**
>
> We thank the reviewer for this interesting comment on exact token matching. We believe this characteristic of PhyloLM can be seen as an issue or a relevant feature depending on what is the question we want to answer while running the algorithm. For example if we consider top end models like ChatGPT and Gemini, these models are trained to answer questions accurately. Thus if we query them with the text “The sum of 1+1 is” they will both answer 2. Thus we will not be able to notice the difference between the two models on this question except maybe in the way they present their answer. If one answers “2” and the other “1+1=2” we can make the difference between them in the framing of their answer. As such, we believe that if the question is to trace the evolution of LLMs, the syntax can be very important. However if we want to study the intelligence of these models, the syntax shouldn't matter. As such, for the main purpose of this paper that is to study the evolution of LLMs we believe the fact that a syntaxical match can be appropriate. However, the question of mapping the general intelligence of these models is also extremely relevant and, for this question, the exact token matching is a major flaw of PhyloLM. Future work in this direction could propose a more semantic based matching method.

---

> > ### Author Response · Authors · 2024-11-21
> > **Rebuttal Part 2**
> >
> > **Third party tokenizer**
> >
> > To address the reviewer’s questions, we indeed could use a 3rd-party tokenizer to calculate the distance between models which have different tokenizers. We just found it simpler to only compare the first 4 characters as it doesn’t require an arbitrary choice of third party tokenizer and it is very simple to implement. In a way it is similar to using a 3rd-party tokenizer with a vocabulary of words of 4 characters but indeed it could be a way to explore in future work.
> >
> > We thank the reviewer for the additional reference and added it in the paper.
> >
> > We hope that this discussion clarifies our work. We would be glad to discuss any remaining or new questions.

---

> > > ### Comment · Reviewer_mtg8 · 2024-11-26
> > > **Reviewer's response to the rebuttal**
> > >
> > > Thank you for the extensive response.
> > >
> > > Figure 5 is now significantly more clear, as well as the usage of tokenizer are fully addressed in the revised manuscript. I also applaud the decision of authors to expand and revise the background literature on L48-49.
> > >
> > > Due to the confusion surrounding the natural selection and role played by the neutral drift, I believe that this point is worth discussion, even if in an appendix, but I understand the authors' decision to opt for better clarity for general public.

---

### Official Review · Reviewer_DQM3 · 2024-11-08

**Soundness:** 2
**Presentation:** 3
**Contribution:** 2
**Rating:** 5
**Confidence:** 3

**Summary:**

The paper proposes a biology-inspired paradigm for categorizing LLMs - an "evolutionary tree" of LLMs can be constructed by sampling from each LLM using a shared prompt dataset, and then comparing all outputs. The specific term used in the paper is "phylogenetic tree". The paper proposes an algorithm for constructing such a tree. The paper also discusses results, and the correlation of such generated trees with the ground truth and model performance.

**Strengths:**

Originality:
- Classifying LLMs into an evolutionary tree is a novel idea

Quality:
- The paper provides an algorithm, sample prompts used on LLMs, an extensive list of tested LLMs, and a re-constructed evolutionary tree on L324, vs. the "ground truth" evolutionary tree.

Clarity:
- The paper is generally clearly written, and the authors' thought process, various analogies to biology (e.g. a prompt as an "allele") are well-explained.

**Weaknesses:**

While the connection to biology is interesting and well-described, a case was not well-made for *why* the analogies should be made as the paper describes.

For example, the paper proposes the analogy of "gene & allele" and "prompt & completion": this is a bit hand-wavy; genes & alleles suggest the idea of some inherent characteristic, but how a prompt is completed may be more a function of the training dataset, rather than some model characteristic as implied by the paper: the idea that models are "evolved", and thus classifiable into an evolutionary tree. The separate contributions of model architecture vs. training dataset to a model was not explored.

Further, the prompt datasets used in the paper appeared to be fairly trivial, and only a very short completion was evaluated. This would be like asking college applicants to write essays, but limited to 3 words each: why comparing such short completions would still be meaningful was not explored.

**Questions:**

Suggestion: It may be more informative to apply the same technique on more well-defined and challenging tasks, such as writing an essay, rather than compare next-few-token completions on what are essentially random string prompts.

Question: What might be the relative contributions of model architecture vs. pre-training/instruction-tuning datasets when using this approach to classify models?

Question: The paper suggests that this analysis can provide an estimate of model capabilities; but why would this analysis provide any more insight than simply inspecting the model size, and directly evaluating the model on well-defined tasks?

Question: The paper appears to use a string similarity measure of sampled outputs to draw conclusions about model relationships. Could we compare sampled outputs in any "deeper" way, such as by comparing some notion of intent, or "tendencies" of particular models?

---

> ### Author Response · Authors · 2024-11-21
> **Rebuttal Part 1**
>
> We sincerely thank the reviewer for their thoughtful feedback and constructive suggestions. We are encouraged by the fact that the reviewer found the idea “novel”, “clearly written” and of good quality. Below, we address the main points raised:
>
> **More details on why the analogy makes sense**
>
> We acknowledge the reviewer’s concern about the analogy between "gene & allele" and "prompt & completion." Our work builds on prior studies that successfully applied phylogenetic methods to cultural artifacts even though there was not a strict equivalence between the domains. The effectiveness of these methods lies not in a direct equivalence between genetics and cultural artifacts, but in the fact that they evolve in a similar way [7]. As such, applying algorithms designed to study the evolution of living species to cultural artifacts can prove efficient in reconstructing their evolution as well. For instance, D’Huy used these techniques to analyze myths and texts, demonstrating their effectiveness in tracing the evolution of mythological narratives over time [1,4], and uncovering new insights that have been validated by archeological studies later. Similarly, Atkinson applied these methods to study language evolution [2], while Grey and Tehrani used them to reconstruct human cultural history [3,5]. Importantly, this approach is not limited to texts; Le Bomin demonstrated its applicability to music and rhythms [6], generating valuable insights previously unavailable in the field. Despite the correspondence between alleles and variants of cultural objects (be them myths or songs) being metaphorical, the translation of genetic concepts and tools has proven useful in all these domaines and delivered important insights and predictions that has been tested and verified empirically.
>
> In our analogy, LLMs are cultural artifacts (as they are built by humans based on what they learned from other humans, and trained on huge amounts of human-generated data) that evolve through more or less gradual changes in architecture, training corpora and fine-tuning. Their evolution reflects in changing their “behavior”, which is essentially their prompt-specific token selection probabilities. While prompts and completions are not strictly equivalent to genes and alleles, both reflect evolutionary mechanisms : genes change through mutations over reproduction - phylogenetic algorithms use these small variations from specie to specie to reconstruct their evolutionary history; token probabilities change through modifications of the characteristics of the LLMs  -phylogenetic algorithms could use these small variations from LLM to LLM to reconstruct their history. We hope this clarifies the reason why phylogenetic algorithms are used in cultural artifact evolution reconstruction and also why it makes sense to use them in LLMs as well.
>
> In addition, we discussed above why the analogy makes sense on a theoretical level (ex ante) but the results also show that it works (ex post). PhyloLM was able to successfully reconstruct the evolution of LLMs within a family as well as classifying models in various families using this methodology. The results we show in the paper also validate the efficiency of the method.
>
> **Short VS Long Completion**
>
> The reviewer questioned the use of short completions and suggested exploring longer contexts or more complex tasks. We focused on short contexts to ensure computational efficiency and simplicity and, a key contribution of our work is demonstrating that short contexts can reveal substantial model-specific information. To address this concern more in details, we performed additional experiments with contexts of various sizes (5, 200 and 1000 characters) using the same methodology. These results provided in Appendix K in the revised version of the paper show that the size of the genes doesn’t matter very much in the phylogeny reconstruction of families of language models.
>
> Generating essays or longer completions, while informative, introduces high computational cost and complexity in comparison. Our approach emphasizes generating many short completions to maximize information while minimizing resource use and the complexity of the algorithm.
>
> **Model architecture**
>
> We share the reviewer’s interest in exploring the respective contributions of model architecture and training datasets to model behavior. However, identifying suitable model families for such an analysis is challenging, as there are few cases where models with diverse architectures are trained on the same dataset. This scarcity makes it difficult to conduct a large-scale study robust enough to draw meaningful conclusions in the current landscape of LLMs.
>
> Critically, seen from an evolutionary perspective, the fact that most LLMs are based on the transformers architecture shows the architecture doesn’t seem to be the main determinant of LLMs evolution as most teams have converged to a similar architecture for now.

---

> > ### Author Response · Authors · 2024-11-21
> > **Rebuttal Part 2**
> >
> > **PhyloLM VS Direct Benchmarking**
> >
> > The reviewer questioned the advantage of our method over direct benchmarking. We emphasize that traditional benchmarks can be prohibitively expensive, particularly when evaluating the thousands of LLMs now available. PhyloLM offers a cost-effective alternative by estimating performance across numerous models based on data from just a few benchmarks. For instance, researchers seeking the best LLM for a specific task can use PhyloLM to pre-select promising candidates from hundreds of models at minimal cost to run the full benchmark on, significantly reducing the number of models requiring full benchmarking. While PhyloLM predictions are not flawless, they provide valuable guidance for efficiently navigating the large landscape of models.
> >
> > Additionally, we speculate that PhyloLM can extend beyond standard benchmarks. Some capabilities, such as artistic skill or explanation conciseness, are challenging to quantify in a benchmark and often require human evaluation—an expensive and time-consuming process. For example, if we are interested in finding LLMs that are able to act very much like humans in a Turing test game it may be very difficult to create a benchmark that is able to rank LLMs efficiently. The best metric is probably whether humans are able to distinguish them from a real human in a Turing test. We can ask humans to grade a few LLMs on such a task (which is expensive to run on many models) and then run PhyloLM to guess the score of hundreds of models in order to find good candidates for the task making it possible to explore a lot of models at a low cost. PhyloLM can help identify models likely to excel in these abstract skills, serving as a valuable tool for prioritizing candidates for further, more detailed evaluation. We view this idea as a very interesting follow up work.
> >
> > **Deeper comparison between LLM generations**
> >
> > We appreciate the suggestion to explore deeper comparisons, such as model "intent" or "tendencies." It is true that PhyloLM focuses on token-level changes, which may provide limited insight into higher-level cognitive traits of LLMs. However, as this is the first study to apply phylogenetic algorithms to LLMs, our aim was to begin with a straightforward and foundational approach. While it is well-established that training alters token probabilities–making it feasible to trace model evolution–it is less clear how or if higher-level traits like intent are shaped during pretraining or finetuning. This is an exciting direction for future research, but we believe it is out of the scope of this first paper introducing this novel idea to the field.
> >
> > We hope this response clarifies our contributions, addresses the reviewer’s concerns and hope our answers can lead to a better evaluation of our work. We are grateful for the opportunity to refine our work further and welcome any additional questions or feedback.
> >
> > **References**
> >
> > [1] d’Huy, Julien. (2013). Polyphemus (Aa. Th. 1137). A phylogenetic reconstruction of a prehistoric tale. Nouvelle Mythologie Comparée, 1, 3–18.
> >
> > [2]Atkinson QD, Meade A, Venditti C, Greenhill SJ, Pagel M. Languages evolve in punctuational bursts. Science. 2008 Feb 1;319(5863):588. doi: 10.1126/science.1149683. PMID: 18239118.
> >
> > [3]Gray RD, Bryant D, Greenhill SJ. On the shape and fabric of human history. Philos Trans R Soc Lond B Biol Sci. 2010 Dec 12;365(1559):3923-33. doi: 10.1098/rstb.2010.0162. PMID: 21041216; PMCID: PMC2981918.
> >
> > [4]Tehrani, Jamie & d'Huy, Julien. (2017). 2017. Phylogenetics Meets Folklore: Bioinformatics Approaches to the Study of International Folktales. 10.1007/978-3-319-39445-9_6.
> >
> > [5]Tehrani, Jamie & Collard, Mark. (2009). On the relationship between interindividual cultural transmission and population-level cultural diversity: a case study of weaving in Iranian tribal populations. Evolution & Human Behavior, 30 (4). pp. 229-304. ISSN 10905138. 30. 10.1016/j.evolhumbehav.2009.03.002.
> >
> > [6]Le Bomin S, Lecointre G, Heyer E (2016) The Evolution of Musical Diversity: The Key Role of Vertical Transmission. PLOS ONE 11(3): e0151570. https://doi.org/10.1371/journal.pone.0151570
> >
> > [7] Dawkins, R. (2006). The Selfish Gene. Oxford University Press.

---

> > > ### Author Response · Authors · 2024-11-26
> > >
> > > We sincerely thank the reviewer once again for their thorough and thoughtful review of our manuscript. We did our best to address the reviewer’s questions and have revised the paper in line with the suggestions provided by all reviewers.
> > >
> > > If there are any follow-up questions or additional points requiring clarification, please do not hesitate to let us know. We greatly value the reviewer’s insights and are fully prepared to provide any further information that may be helpful. Additionally, we kindly request the reviewer to consider updating their evaluation if they feel our responses have clarified and strengthened the contribution of our work.

---

> > > > ### Author Response · Authors · 2024-12-02
> > > >
> > > > We kindly remind the reviewer that today is the final day of the discussion. We invite them to ask any remaining question and to update their evaluation if our responses have made the contribution clearer. We thank the reviewer again for their time and constructive feedback.

---

### Official Review · Reviewer_Gfne · 2024-11-08

**Soundness:** 3
**Presentation:** 3
**Contribution:** 2
**Rating:** 6
**Confidence:** 4

**Summary:**

The paper introduces PhyloLM, a method adapting phylogenetic algorithms to Large Language Models (LLMs) to explore whether and how they relate to each other and to predict their performance characteristics. More concretely, the authors make an analogy between phylogeny and LLM, with token context corresponding to the genes and generated tokens corresponding to the alleles. Then by applying Nei Generic distance to the empirically sampled character distribution conditional on some context, the authors first compute the similarities between models and then use the NJ algorithm to construct the phylogenetic tree. Experimentally, the authors have evaluated the choice of hyperparameters of the phylogenetic algorithms, the correctness of the recovered trees, and the quality of prediction when using the model similarities to infer their eval performances. They found that the recovered trees correspond to the know relationships of the different model (classes) and the performance prediction correlate well with the actual performance.

**Strengths:**

- The presentation of the paper is clear and easy-to-read and the methodology seems reasonable and sound.
- The idea of applying phylogenetic algorithms to study the evolution of language models, to the best of my knowledge, is a novel and interesting idea. This could inspire new works along this direction.
- The observation that using the conditional empirical distribution (with N=32) of the conditional generation of 4 subsequent letters could already provide a lot of information about the identify of the language model is an interesting (and to me surprising) observation which can be further explored by future works.

**Weaknesses:**

- **[Practical usefulness of the phylogenetic tree]** In the current paper, the phylogenetics of Language models identified by running the NJ algorithm is verified using existing knowledge of the open and closed source models. Here, despite not knowing the exact training set/algorithm for some of the close models, their high-level relationships are still known to their end users (such as what company produces these models, which generation they are from, how they relate to previous models of the same company, etc). So it’s not clear to me whether any additional information can be gained by constructing the similarities and the phylogenetic tree that is not already know to the public. I believe this work still has its value purely for exploring this academically interesting direction, but from reading it, I haven’t got a sense whether the authors believe their methods could provide new insights about model evolution/relationships that’s not already publicly known.
- **[How to think about the benefits of performance prediction using similarities]** In terms of the contribution of using the similarity score to predict the downstream eval benchmarks’ performance, the authors have mentioned in line 312 - 315 that their estimation could use fewer queries than actually evaluating on the actual benchmarks such as MMLU. However, I think the authors could use more space to discuss the type of tradeoffs involved in actually using the similarity matrix-based performance estimators. For example, although the authors’ method use fewer inference compute for eval, they also are only correlated with the actual values but not very accurate, which makes me think some further discussion on what scenarios the authors could imagine their methods as practically useful for performance prediction/comparison.
- Minor:
    - There is a small amount of content redundancy between line 211 - 214 on  page 4 and line 297 - 305 on page 6 discussing the very expensive distance matrix.
    - The *completion models* on line 384 could be italicized.

**Questions:**

Beyond the two questions above, I’m also curious whether the authors have experimented with any prompt ($c$) distribution other than math and coding and whether there are any distributions where the recovered phylogenetic trees do not correspond to our prior knowledge of how the models relate.

---

> ### Author Response · Authors · 2024-11-21
> **Rebuttal**
>
> We sincerely thank the reviewer for their insightful feedback. We are glad that the reviewer found the idea “novel”, the paper “clear and that the reviewer was surprised and interested by some of our findings. We address the reviewer comments below.
>
> **Practical usefulness of the phylogenetic tree**
>
> The reviewer raised an important concern about the utility of reconstructing phylogenetic trees, particularly since some open-access models disclose their lineage in model cards. However, many major models from companies like Google and OpenAI often lack transparency regarding training data or lineage, posing challenges for researchers. Even among open-access models like Gemma, Llama, Mistral, and Qwen, full training details are frequently absent (training data and sometimes lineage). PhyloLM addresses this gap by adapting genetic concepts to trace model lineages, especially for completion-based models.
>
> PhyloLM has successfully reconstructed some of the Gemma and Gemini family’s lineage and identified potential training data sources for earlier OpenAI models. While such efforts may be less critical for fully transparent models, PhyloLM can be a great tool to better understand proprietary models, supporting transparency and aiding researchers in understanding the evolution of large language models. Moreover, even for transparent models, the exponential growth of open-access models makes it increasingly important to develop tools to visualize differences between hundreds or even thousands of models.Furthermore, PhyloLM doesn’t only return the phylogeny (that may already been known) but also the distance between a parent and a child that is also quite important and not disclosed in model card. In short, PhyloLM can serve as a powerful tool for large scale LLM visualization, and can provide very valuable information even when lineages are disclosed.
>
> **How to think about the benefits of performance prediction using similarities**
>
> Regarding performance prediction, let us first clarify that we are not proposing PhyloLM as a replacement for benchmarks. Instead, it functions as a complementary mapping tool. Its predictions are approximate but computationally inexpensive, helping researchers efficiently narrow down potential model candidates for specific tasks. These candidates can then undergo thorough testing on more expensive benchmarks, offering a cost-effective strategy for navigating the expanding LLM landscape rather than testing every LLM one by one on each benchmark.
>
> Finally, beyond performance prediction, we speculate that PhyloLM can also address tasks that are difficult to benchmark, such as assessing “creative” skills or abstract abilities of LLMs. For instance, given a few models identified as strong in an abstract task (e.g., through human feedback), PhyloLM could predict other models likely to excel in the same task–a capability that benchmarking alone cannot provide. For example, humans could grade several LLMs on how they liked their interaction with a chat bot and PhyloLM could feed on this information to try to find new chatbot that users could like as well.
>
> **Other sets of genes**
>
> Lastly, the reviewer is right, pointing out that the selection of the gene set can influence the reconstructed trees, as it is already the case in studies in biology (or other artifacts). Our initial focus was on genes based on assessment of reasoning and coding skills to help follow recent trends in LLM development and increase our chance to  show interesting differences between models and their finetuned versions.
>
> However, as the Reviewer raised the point, we have since tested additional gene sets aiming at showing that the choice of genes can help visualize more than just the evolution of LLMs. For example, a Chinese poetry gene set (added in Appendix J.1) has helped visualize models that understand Chinese (Qwen and Bloom in our study). Similarly, chat-style genes (added in Appendix J.2) have proven effective for identifying models fine-tuned for conversational interactions. Finally we also added a gene set extracted from Wikipedia showing that our results are still consistent with a third general gene set (see Appendix J.3). These new results will be included in the Appendix, and future work will explore even more nuanced and underreported skills absent from model cards such as skills at chess.
>
> We hope this discussion provides clarity about our work, addresses the reviewer’s concerns, and facilitates a more comprehensive evaluation of our contributions. We would be glad to discuss any additional questions or feedback the reviewer may have.

---

> > ### Author Response · Authors · 2024-11-26
> >
> > We sincerely thank the reviewer once again for their thorough and thoughtful review of our manuscript. We did our best to address the reviewer’s questions and have revised the paper in line with the suggestions provided by all reviewers.
> >
> > If there are any follow-up questions or additional points requiring clarification, please do not hesitate to let us know. We greatly value the reviewer’s insights and are fully prepared to provide any further information that may be helpful. Additionally, we kindly request the reviewer to consider updating their evaluation if they feel our responses have clarified and strengthened the contribution of our work.

---

> > > ### Author Response · Authors · 2024-12-02
> > >
> > > We kindly remind the reviewer that today is the final day of the discussion. We invite them to ask any remaining question and to update their evaluation if our responses have made the contribution clearer. We thank the reviewer again for their time and constructive feedback.

---

### Author Response · Authors · 2024-11-27
**Global Rebuttal**

We would like to start by thanking all the reviewers for their very valuable feedback. We found it very encouraging that the reviewers found the work “novel” (Gfne08), “interesting” (Gfne08) and “original” (DQM308,Lm8z03). We are glad most of them found it very clear and easy to read/well explained (DQM308,mtg803,Gfne08). We are particularly happy that reviewers found the work outstanding (mtg803), potential high impact (Lm8z03) that it could inspire new works in the same direction (Lm8z, Gfne08).

The field of LLMs is evolving extremely fast with more than 600 new open access LLMs released everyday on the huggingface hub and proprietary models of always increasing intelligence and complexity while being less and less transparent. To differentiate between models people use benchmarks to assess their capabilities on specific tasks such as reasoning, general knowledge or coding. However these benchmarks are extremely expensive to compute and more than 90% of LLMs are not benchmarked. How can we reliably evaluate and select the most suitable LLM for a specific task amidst the abundance of unbenchmarked models? In a landscape where powerful proprietary models are increasingly opaque, how do we effectively study and understand their capabilities and limitations?
It is in this light that we propose PhyloLM, an algorithm inspired from genetics that is able to produce distance matrices between models at a very low cost making it possible to draw phylogenetic trees of language models but also to estimate benchmark capabilities of many models very cheaply.

The use of phylogenetic algorithms to trace the evolution of cultural artifacts has repeatedly demonstrated its effectiveness across various domains [1-7]. Although cultural artifacts differ fundamentally from genes, they share a similar evolutionary process: myths gradually refine over time, allowing their evolutionary history to be reconstructed by analyzing similarities among versions [1,4]. Similarly, musical practices evolve through gradual changes in techniques and instruments, enabling the reconstruction of their histories [6]. Richard Dawkins also likened the evolution of memes to genetic evolution [7].

PhyloLM shows very good results in recovering the families of models among 17 different families and more than 150 LLMs but also the evolution within a family. These results validate the methodology, the evolutionary analogy with LLMs and the use of phylogenetic tools in LLMs paving a new way in the field of model analysis and understanding.

We sincerely thank all the 4 reviewers for their very valuable feedback that really helped improve the paper. Here is a list of the main points raised during the discussion and how we addressed them. All changes are written in green in the revised paper.
- Reviewers Gfne08 and mtg803 enquired about whether some gene sets could be designed to not reflect the phylogeny but other qualities of LLMs. We updated the paper with few more appendices starting with adding more sets of genes to show the impact of gene choice in appendix J. We show that using a gene set based on chinese poetry doesn’t lead to good phylogenetic trees but makes it very clear which LLM speak chinese. Using a chat-formatted gene set doesn’t reconstruct a good evolutionary tree either but the tree reflects which models where trained on chat interactions. These results show that, while we focused on gene sets to reconstruct the phylogenetic tree in the main text, many more questions can be answered with a well tailored gene set. This would be a very interesting direction to delve into for future work.
- Reviewers DQM3 and Lm8z questioned the arbitrary values about the gene length and completion length. We tested various values and show that the ones from the main text are nearly optimal in terms of contrast of the similarity matrix and phylogenetic tree accuracy. Results are shown in appendix K.
- Reviewers Lm8z03 questioned the validity of cutting 4 first characters of LLM completions in order to compare them in a similar referential rather than using tokens and mtg803 proposed to use a 3rd party tokenizer to do this comparison. We answered that cutting the 4 first characters of a LLM completion is akin to using a universal 3rd party tokenizer for which tokens are 4 characters long.
- Finally we reviewer mtg803 suggested to change Figure 5 to increase its clarity. We swapped Figure 5 from the main text with figures from the appendix F for better clarity, furthermore, we put additional very relevant references and corrected minor text formatting issues as pointed out by reviewers mtg803 Gfne08.

In addition to this global rebuttal we address each reviewer question and comment in individual rebuttal. Overall we believe these suggestions have helped improve the clarity of the paper both in terms form and content. Code and data used to generate these additional figures will be made available publicly as well.

---

> ### Author Response · Authors · 2024-11-27
> **Global Rebuttal References**
>
> **References**
>
> [1] d’Huy, Julien. (2013). Polyphemus (Aa. Th. 1137). A phylogenetic reconstruction of a prehistoric tale.Nouvelle Mythologie Comparée
>
> [2]Atkinson QD, Meade A, Venditti C, Greenhill SJ, Pagel M. (2008) Languages evolve in punctuational bursts. Science.
>
> [3]Gray RD, Bryant D, Greenhill SJ. (2010)  On the shape and fabric of human history. Philos Trans R Soc Lond B Biol Sci.
>
> [4]Tehrani, Jamie & d'Huy, Julien. (2017).  Phylogenetics Meets Folklore: Bioinformatics Approaches to the Study of International Folktales
>
> [5]Tehrani, Jamie & Collard, Mark. (2009). On the relationship between interindividual cultural transmission and population-level cultural diversity: a case study of weaving in Iranian tribal populations. Evolution & Human Behavior
>
> [6]Le Bomin S, Lecointre G, Heyer E (2016) The Evolution of Musical Diversity: The Key Role of Vertical Transmission. PLOS ONE
>
> [7] Dawkins, R. (2006). The Selfish Gene. Oxford University Press.

---

### Meta-Review · Area_Chair_xe9y · 2024-12-08

**Metareview:**

The paper:
* Proposes similarity score between any two LLMs, which is, averaged over a huge set of prompts, the (normalized) dot product between string-response distributions.
* Uses this similarity score to construct a phylogenetic tree of 100+ LLMs, showing ancestral similarities (e.g. OpenAI models belong on one subtree, same with Gemini, etc.)
* Similarity matrix can be used to predict the performance of held-out LLMs on ARC benchmark.

## Strengths
* The use of the string-response distribution metric is well-motivated, and the empirical results do show that they do have some predictive power.
* The research is timely and important, especially for introducing data-analysis techniques for performance prediction on evaluation metrics.

## Weaknesses
* The biological analogies seem more distracting than helpful. For instance, in my summary of the paper, I was able to use purely mathematical notation to explain the contributions, without needing to use any evolutionary terminology.
  * The phylogenetic tree isn't particular either - it can be seen as just one type of "clustering", but there are others that could've worked just as well (e.g. spectral/k-means)
* Some efforts would have been better spent providing more fundamental analysis. For instance, since the similarity score is simply the dot product of the normalized text output distributions, each LLM can be represented as its own feature vector. This feature vector could perhaps be very high dimensional but informative, if applied over more prompt datasets than just math and code.
  * For the performance prediction, additional modeling techniques (different kernel / similarity definition) could have also made things more effective.
  * The mechanics of the string output distribution could've been studied further (e.g. why is 4 response characters optimal?).

**Additional Comments On Reviewer Discussion:**

The variance scores on this paper are very high (3, 5, 6, 10), which made me read the paper very carefully myself, hence I gave my own careful review and assessment.

Main benefits raised by reviewers:
* Connections to evolution and biology is novel (All reviewers)
* Paper has potential to be high impact for mainly raising the use of string output distributions (Reviewer Lm8z, mtg8, Gfne)

The main issues raised by reviewers:
* Does the paper really need to discuss so much into genetics and evolution? (Reviewer DQM3, Lm8z)
* Mechanics of the string output distribution: why is 4 characters optimal? (Reviewer Lm8z)

I think there's general agreement that this paper focuses on an important topic, and got interesting results, around representing LLMs by their string-output distributions. But I personally don't think the paper should've went heavy on the evolutionary analogies, given my explanation of the exact mathematics that the paper is performing - they could've spent more time studying the mechanics of the method (e.g. why 4 characters only) and doing more rigorous data science.

Therefore, I propose to accept (poster), but don't believe it should be given higher recommendation because there were a lot of missed opportunities to understand string output distributions more rigorously.

---

### Decision · Program_Chairs · 2025-01-22

Accept (Poster)